# Rational design of a hospital-specific phage cocktail to treat *Enterobacter cloacae* complex infections

Dinesh Subedi [1,2,3] ✉, Fernando Gordillo Altamirano[1,2,4], Rylee Deehan[1,5], Avindya Perera[1], Ruzeen Patwa[1,4], Xenia Kostoulias [2,4,6], Denis Korneev [7], Luke Blakeway[4], Nenad Macesic[2,4], Anton Y. Peleg [2,4,6,8] & Jeremy J. Barr [1,2,8] ✉

The Alfred Hospital in Melbourne, Australia, has reported an ongoing outbreak of infections caused by multidrug-resistant *Enterobacter cloacae* complex (ECC). Phage therapy is a promising strategy to treat antimicrobial-resistant infections. Utilizing the hospital's isolate collection, built over the past decade, we established an initial 3-phage cocktail with 54% ECC coverage. We then iteratively improved this product by enhancing phage killing efficiency using phage adaptation and expanded host range through targeted phage isolation against low-coverage ECC isolates. This optimization yielded Entelli-02, containing five well-characterized virulent phages that target clinical ECC isolates via distinct bacterial cell surface receptors. Entelli-02 exhibits 88% host coverage against The Alfred Hospital's ECC isolate collection ($n = 206$), confirmed by plaque formation and reduced bacterial load in septicaemic mice by >99%. We produced this cocktail as a therapeutic-grade product, ready for clinical use. Entelli-02 represents a hospital-specific phage cocktail with frontline efficacy and on-demand availability.

The emergence and spread of antimicrobial resistance (AMR) poses a serious threat to global health and calls for alternative strategies to combat bacterial infections[1]. Phage therapy, which involves the use of virulent bacteriophages (phages) that can infect and kill bacteria, has garnered renewed attention as one potential solution to the AMR crisis[2]. Phage therapy offers several advantages over conventional antibiotics, including high specificity, self-replication, low toxicity and adaptability to changing bacterial pathogens[3]. Combinations of multiple phages, which are colloquially known as phage cocktails, are often used to broaden the antimicrobial spectrum. Phage cocktails can

be produced a priori against a target pathogen or group of bacteria[4]. Despite these benefits, clinical trials employing phage cocktails have shown limited clinical efficacy[5–12]. Treatment failure has been attributed to target strain divergence, phage stability issues and the mismatch between phage infectivity against the bacterial isolates the cocktails were constructed upon versus the clinical strains that were eventually treated. In contrast, personalized phage therapy involves the identification and use of a phage with demonstrated activity against a patient's bacterial infection[13,14]. This approach is often followed by adapting the phages (also known as 'phage training') to improve their effectiveness,

[1]School of Biological Sciences, Monash University, Clayton, Victoria, Australia. [2]Centre to Impact AMR, Monash University, Clayton, Victoria, Australia. [3]School of Optometry and Vision Science, UNSW Medicine, University of New South Wales NSW, Kensington, New South Wales, Australia. [4]Department of Infectious Diseases, The Alfred Hospital and School of Translational Medicine, Monash University, Melbourne, Victoria, Australia. [5]Australian Centre for Ecogenomics, School of Chemistry and Molecular Biosciences, University of Queensland, Saint Lucia, Queensland, Australia. [6]Infection Program, Monash Biomedicine Discovery Institute, Department of Microbiology, Monash University, Clayton, Victoria, Australia. [7]Ramaciotti Centre for Cryo-Electron Microscopy, Monash University, Clayton, Victoria, Australia. [8]These authors contributed equally: Anton Y. Peleg, Jeremy J. Barr. ✉e-mail: dinesh.subedi@monash.edu; jeremy.barr@monash.edu

followed by small-scale production and bespoke treatment. While personalized phage therapy approaches have shown promising results with >70% treatment efficacy[15], translating these approaches into the clinic presents additional complexities, requiring rapid phage characterization, safety and efficacy testing, and administration of the product, all within a clinically relevant timeline. A combination of both approaches, with the initial use of a predefined broad-spectrum phage cocktail, while transitioning to personalized approaches as needed, is an appealing strategy for rapid treatment of antibiotic-resistant bacterial infections.

To bridge the divide between broad-spectrum phage cocktails and personalized therapy, we designed a phage product that was targeted towards a high-risk, multiclonal outbreak of a nosocomial pathogen at a local hospital. We targeted *Enterobacter cloacae* complex (ECC), which is an emerging group of nosocomial pathogens that pose a substantial threat to human health due to their acquisition of virulence and AMR determinants. Classified among the ESKAPE (*Enterococcus faecium*, *Staphyllococcus aureus*, *Klebsiella pneumoniae*, *Acinetobacter baumannii*, *Pseudomonas aeruginosa* and *Enterobacter* species) pathogens, this complex encompasses the clinically relevant species *E. cloacae*, *E. asburiae*, *E. hormaechei*, *E. kobei* and *E. ludwigii*, which can cause diverse infections, such as pneumonia, urinary tract infections, intraabdominal infection and bacteraemia[16,17]. Notably, these pathogens have considerable epidemic potential, having contributed to a global surge in carbapenem-resistant and extended-spectrum beta-lactamase-producing phenotypes[18], and were responsible for numerous clonal hospital outbreaks with limited treatment options[19–21]. As a result, ECC is increasingly associated with severe AMR infections in healthcare settings and was associated with >200,000 deaths globally in 2019 alone[22–27].

The Alfred Hospital, a tertiary referral centre in Melbourne, Australia, has reported an ongoing outbreak of carbapenemase-producing Enterobacterales infections over the past decade[23,28,29]. A retrospective analysis of bloodstream infections at this centre noted that ECC contributed ~20 cases per year, with a recent surge in ECC infections from 2018–2021[29]. Given this clinical burden, there was an urgent need to develop alternative treatments against this endemic nosocomial AMR pathogen. To address this, we developed a tailored phage product that is suitable for frontline use against this ECC outbreak at The Alfred Hospital. We developed a standardized approach that combines academic phage research with clinical insights backed by an extensive collection of 206 clinical ECC isolates over the past decade. This enabled the creation of Entelli-02, an institution-specific phage cocktail that not only demonstrates frontline efficacy but also ensures rapid availability of an effective antimicrobial. Our approach bridges personalized phage therapy with broad-spectrum phage products that are tailored towards a given hospital's (that is, institution) pathogen profile and addresses the urgent need for effective treatment options against AMR pathogens in healthcare settings.

## Results

### Isolation and characterization of phages against *Enterobacter* isolates from The Alfred Hospital

The Alfred Hospital in Melbourne, Australia, has reported an ongoing outbreak of nosocomial infections with limited treatment options and high mortality rates[29]. Of particular concern is the ECC, which is an emerging AMR threat with considerable epidemic potential[22]. Here we explored phages as a frontline antimicrobial solution for The Alfred Hospital's ECC outbreak. Using clinical ECC isolates collected over a 10-year period, we selected a subset of 36 ECC isolates that represent the sequence type (ST) diversity of the full isolate collection (Supplementary Data 1). We then constructed an initial phage library against these isolates. After isolating and purifying 21 phages, we screened their plaque-forming capacity against our isolate subset to obtain a host-range map (Fig. 1a). Of the 36 isolates, 34 were susceptible to

at least one phage. However, considering the time and cost associated with characterizing and producing each therapeutic phage, we aimed to build a phage combination that offered the broadest host range with the fewest possible phage combinations. On the basis of the complementary broad-spectrum host range, we selected øEnA02, øEnC07 and øEnC15 to produce the initial ECC phage combination (cocktail-V1), which provided 61% coverage against the 36 sub-selected ECC hosts (Fig. 1a). It should be noted that these phages were selected on the basis of host-range coverage, and their phage receptor targets were unknown at the time. All phages in the cocktail were sequenced and their genomes were analysed to ensure that they did not carry genes known to allow lysogeny or virulence, including integrases, recombinases, mobile genetic elements or genes encoding antibiotic resistance or toxicity. On the basis of genome similarity, we classified the phages at the genus level (Fig. 1b, and Supplementary Figs. 1 and 2). Electron microscopy imaging (Fig. 1c–e) revealed icosahedral capsids and sheathed contractile tails with lateral tail fibres, consistent with the *Tevenvirinae* subfamily within the *Straboviridae* family[30]. Importantly, therapeutically relevant bacteriophages must be capable of efficient replication and effectively suppress the growth of target strains. To this end, we evaluated the replicative characteristics of the three phages on their host of isolation using the one-step growth curve, revealing that øEnA02 (host: Eaero) had a burst size of 86 per infected cell with a latency of 20 min (Fig. 1f), øEnC07 (host: APO57) had a burst size of 40 with a latency of 20 min (Fig. 1g), and øEnC15 (host: CPO093) had a burst size of 349 with a latency of 15 min (Fig. 1h). Next, we verified the killing efficacy of the 3-phage cocktail as assessed by efficiency of plating (EOP) to evaluate how well the cocktail phages could infect other permissive hosts[31]. The results demonstrated at least one phage within the cocktail with an EOP between 0.1 and >1 for each host except for two instances: øEnA02 against CPO239, and øEnC15 against CPO053 (Fig. 1i).

With our goal to produce a frontline antimicrobial preparation, phage stability under variable storage conditions is paramount[32,33]. We examined the stability of each phage within the cocktail. Cleaned phage lysates in LB media were stored individually in polypropylene tubes under storage conditions at 4 °C, room temperature (~22 °C) and 37 °C for 30 months, with phage titrations performed at different intervals. Within the initial 6 months, no noticeable drop in titre occurred in lysates stored at 4 °C and room temperature, yet over the following 30 months, 4 °C proved the most stable storage condition. Phages stored at 37 °C were the least stable, losing therapeutic titre within 2 months (Fig. 1j). Collectively, these data suggest that these three phages have broad host range and high antimicrobial efficacy against ECC isolates, and demonstrate excellent stability, making them promising candidates for further development and evaluation.

### Identification of phage receptors

Bacteria can quickly evolve resistance against phage predation, which typically manifests via loss-of-function mutations in the surface-associated structures that phages adsorb to. The simultaneous use of diverse phages targeting different bacterial surface structures has been reported to minimize or delay the evolution of phage resistance[34,35]. To identify the receptors involved in phage–host interaction, we generated phage-resistant mutants which were sequenced to examine loss-of-function mutations, focusing on lipopolysaccharide (LPS) and other cell surface related genes[36]. We identified putative phage receptors by mapping the raw sequencing reads of phage-resistant mutants to the wild-type (WT) host genome. To validate these phage receptors, we complemented the candidate wild-type gene back to its respective phage-resistant mutant and performed an adsorption assay to determine whether phage infectivity was restored. We found that each of the three phages recognized a different component of the LPS structure, which was used to mediate adsorption and subsequent infection (Fig. 2a). For the phage-resistant mutant øEnA02, we identified two loss-of-function mutations: the first in an O-antigen-associated

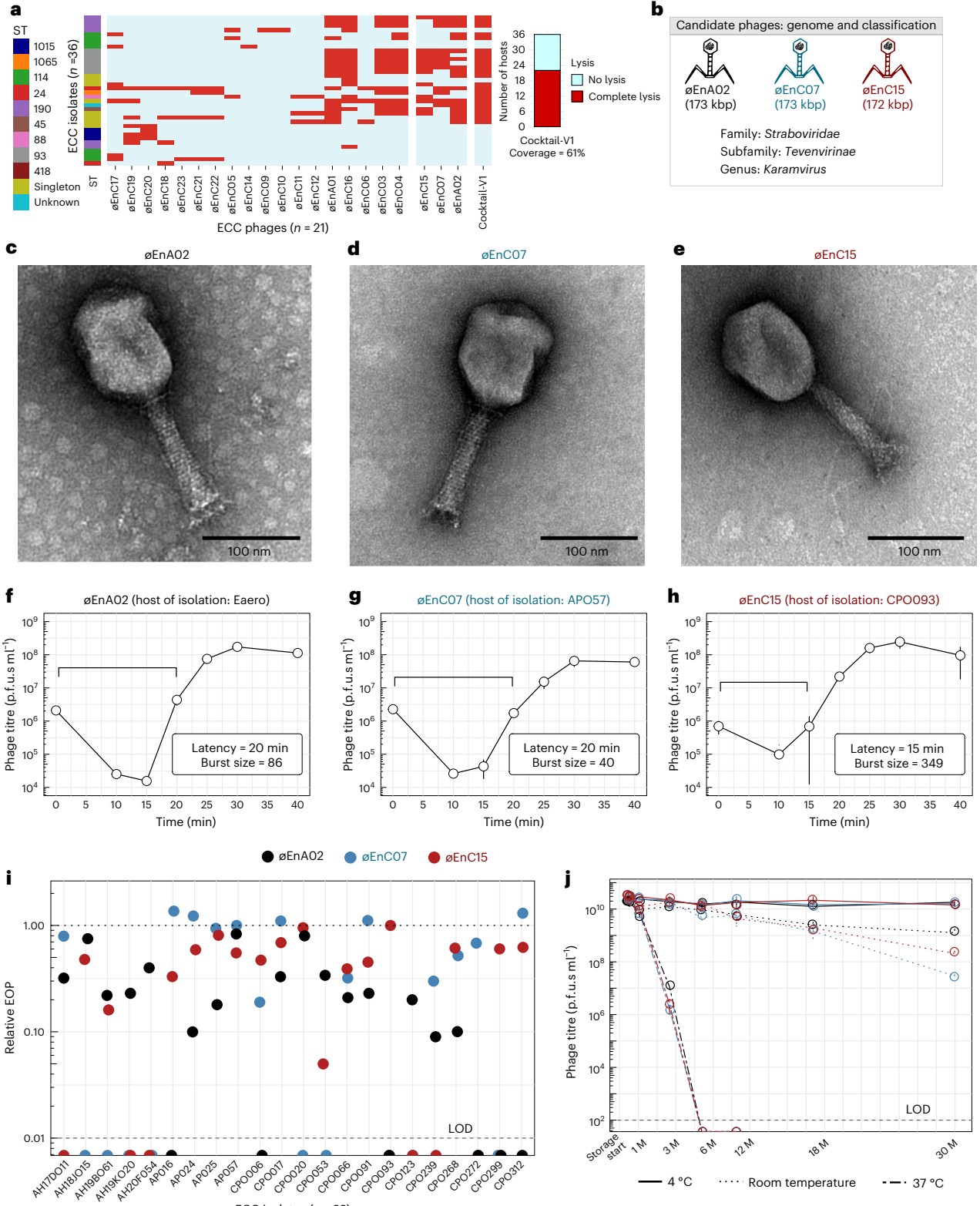

**Fig. 1 | Formulation and characterization of a 3-phage combination. a**, Host-range map of isolated phages (columns, $n = 21$) and cocktail-V1 tested against a subset of 36 ECC isolates. Each row represents an ECC isolate, classified on the basis of ST and colour coded accordingly. Singletons are STs that only occurred once. **b**, Representations of phage genome size and classification of the three phages (øEnA02, øEnC07 and øEnC15) of cocktail-V1. **c–e**, TEM images of the three phages (øEnA02 (**c**), øEnC07 (**d**) and øEnC15 (**e**)). **f–h**, One-step growth curve of the three phages (øEnA02 (**f**), øEnC07 (**g**) and øEnC15 (**h**)) in the cocktail propagated on their host of isolation. Data represent mean ± s.e.m

of 3 biological replicates. The latency time (in minutes) and burst size (in p.f.u.s per infection) were calculated for each phage. **i**, Relative EOP values for three phages, represented by different coloured dots ($n = 1$) across ECC isolates. The upper dotted line represents the EOPs on the host of isolation, which are set as 1. The lower dashed line represents the limit of detection (LOD) of the assay. **j**, Stability of the three phages (øEnA02, øEnC07 and øEnC15) at 4 °C, room temperature and 37 °C storage conditions. Data represent mean ± s.e.m of 2 biological replicates. M, months.

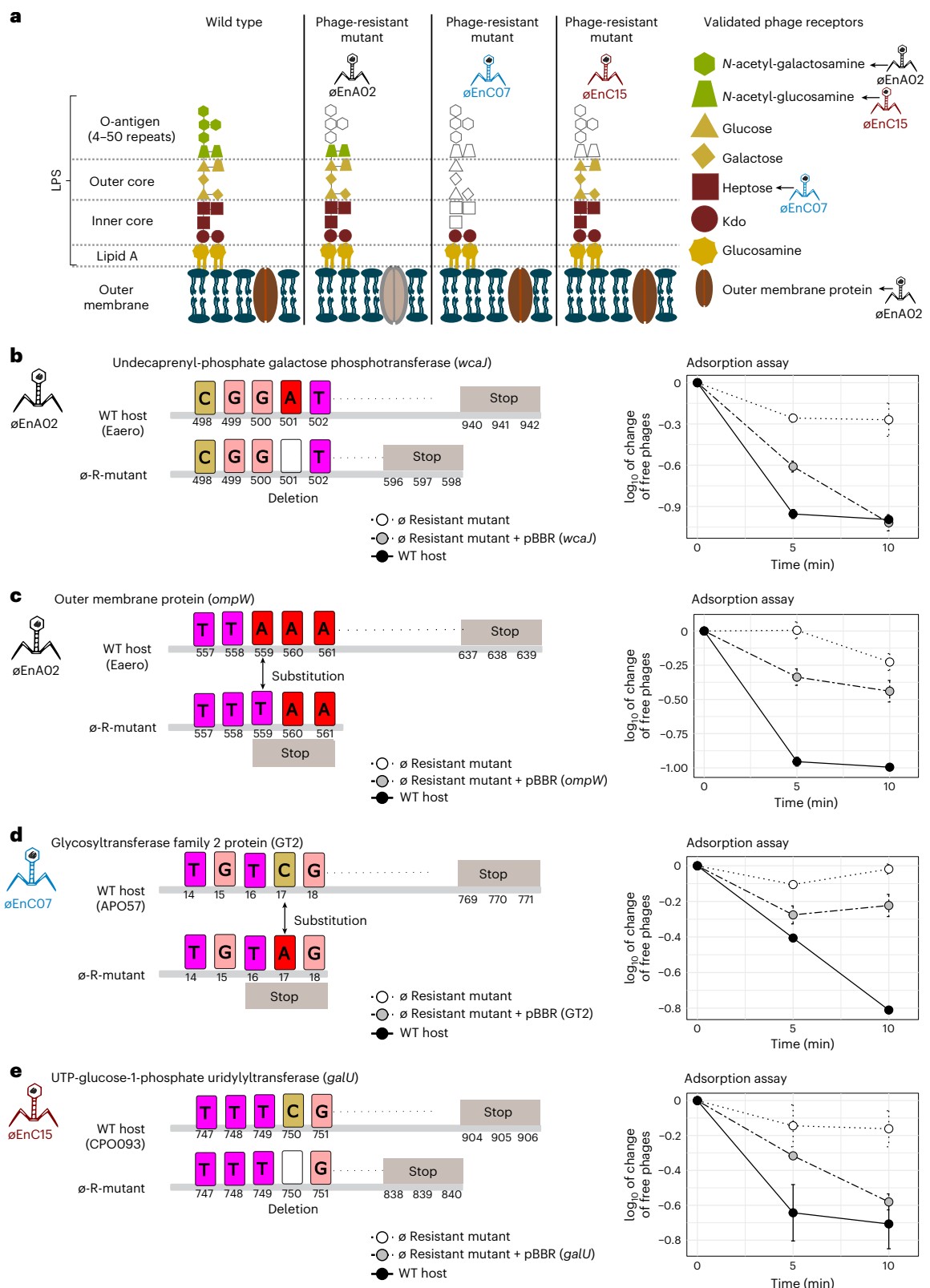

**Fig. 2 | Receptors of three phages of cocktail-V1. a**, Schematic representation of the *Enterobacter* LPS and membrane structure, with validated phage receptors indicated by arrows. Colours are for illustrative purposes only and the faded section represents the potential loss of surface-associated structure in phage-resistant mutants. Figure drawn on the basis of ref. 96 using Inkscape. **b**, SNPs identified in the genome of øEnA02-resistant mutant (ø-R-mutant) and predicted effect on *wcaJ*. The graph on the right shows phage adsorption assay testing the ability of phages to adsorb to its corresponding WT, phage-resistant mutant and complemented host over 10 min. **c**, SNPs identified in the genome of øEnA02-resistant mutant and predicted effect on *ompW* with phage adsorption assay as shown in the graph on the right. **d**, SNPs identified in the genome of øEnC07-resistant mutant and predicted effect on GT2 with phage adsorption assays as shown in the graph on the right. **e**, SNPs identified in the genome of øEnC15-resistant mutant and predicted effect on *galU* with phage adsorption assay as shown in the graph on the right. WT indicates wild-type host, ø-R-mutant indicates phage-resistant mutant, and numbers represent nucleotide positions for the respective genes. Data represent mean ± s.e.m of 3 biological replicates.

gene (*wcaJ*) (Fig. 2b)[37] and the second in an outer membrane protein (*ompW*) (Fig. 2c), with both mutations resulting in the gain of an early stop codon. O-antigen is a serogroup-specific sugar-based component of the outer LPS layer of Gram-negative bacteria and is a well-characterized phage receptor[38–40]. OmpW has been shown to be a phage receptor for *Vibrio cholerae* phage[41–43], having important roles in virulence and acting as a colicin receptor in *E. coli*, suggesting potential fitness trade-offs associated with phage resistance[42,43]. Following complementation, we quantified øEnA02 adsorption to the WT host, each of the two phage-resistant mutants and each mutant complemented with the respective WT gene (Fig. 2b,c). The WT host adsorbed ~1 log of phage within 10 min, while both phage-resistant mutants showed minimal adsorption. Comparatively, the complemented mutant restored phage adsorption, albeit only partially for the OmpW mutant (Fig. 2c), suggesting that øEnA02 uses OmpW as a secondary receptor. The øEnC07-resistant mutant had a loss-of-function mutation in a glycosyltransferase gene (Fig. 2d), which is part of the *rfa* operon and associated with LPS core synthesis[44,45]. Complementing the mutant with the WT gene restored phage infectivity and increased phage adsorption by 0.2 log. Finally, øEnC15-resistant mutant had a stop codon in the UTP-glucose-1-phosphate uridylyltransferase gene (*galU*) (Fig. 2e), which encodes an enzyme for UDP-glucose synthesis, which in turn is a precursor for O-antigen synthesis[46]. Complementing *galU* into the phage-resistant mutant restored phage adsorption, indicated by a 0.6 log increase.

The identification of phage receptors provides mechanistic insights into phage infectivity and the emergence of resistance. While all our phages broadly targeted the LPS, this was mediated through recognition of distinct LPS subunits, which may reduce the emergence of phage resistance when used in combination (Supplementary Fig. 3a). We noted that phage activity was not fully restored in phage-resistant mutants after complementation, which may have been due to altered gene expression post complementation. In addition, phage-resistant mutants harboured several single nucleotide polymorphisms (SNPs) in non-target genes with either known or hypothetical functions (Supplementary Data 2), and we cannot exclude the possibility that these mutations contributed to partial restoration or altered adsorption. This remains an area for future investigation.

## In vivo effectiveness of the preliminary phage cocktail-V1

After establishing the host range, infectivity, genomes, stability and receptors for our preliminary 3-phage combination (cocktail-V1), we evaluated its efficacy in reducing ECC burden in vivo using a murine model. For this model, we selected ECC isolate APO57, which was susceptible to all three phages at an EOP of ~1. We optimized the ECC inoculum dose to induce severe septicaemia in 8-week-old BALB/c mice, reaching an ethical endpoint within 12 h (Supplementary Fig. 3b,c). Mice were intraperitoneally injected with 200 µl of an optimized dose of $5 \times 10^6$ colony-forming units (c.f.u.s) ml$^{-1}$ of isolate APO57. At 1 h post infection (hpi), mice were treated with a single dose of cocktail-V1, which contained $10^8$ plaque-forming units (p.f.u.s) ml$^{-1}$ of each of the three phages in 200 µl, while control mice received an equivalent volume of sterile PBS. The experiment concluded at 12 hpi and vital organs were collected to assess the bacterial and phage load (Extended Data Fig. 1a). The phage-cocktail treatment resulted in >3 log reduction (>99.9%) in the bacterial burden in the infected mouse's blood, kidney, liver and spleen compared with the PBS control group ($p < 0.05$) (Extended Data Fig. 1b). Notably, bacterial load in the blood was cleared below our limit of detection in four out of six mice. Phage load assessment revealed propagation of all three phages in vivo, indicating successful replication and dissemination throughout the mice (Extended Data Fig. 1c). We further observed that the liver and spleen exhibited higher phage concentrations compared with the blood and kidney ($p < 0.05$). However, there was no significant difference in phage concentrations when comparing the blood to the kidney or the liver to the spleen. Our phage cocktail development was an iterative process, and this in vivo experiment served as a key checkpoint to assess efficacy. Due to its capacity to reduce bacterial burden and ability to replicate effectively in mouse organs, we advanced the 3-phage combination to the next stage of clinical evaluation.

## Efficacy of cocktail-V1 against the wider collection of clinical ECC isolates

At this stage, we sought to determine the effectiveness of our phage cocktail-V1 against the broader clinical collection of ECC isolates from The Alfred Hospital. We conducted high-throughput screening of the lytic activity of cocktail-V1 and its individual components via a spot assay against 120 clinical ECC isolates (Extended Data Fig. 1d). At the time, this represented the entire ECC collection at The Alfred Hospital, encompassing a range of STs that contributed to the nosocomial outbreak. Two researchers who were blinded to the experiment visually scored the zones on the bacterial lawn as complete lysis, partial lysis, or no lysis (Supplementary Fig. 3d). The 3-phage cocktail lysed 54.2% of the 120 ECC isolates, which was a decrease of ~7% in lysis when compared against the initial 36-isolate collection cocktail-V1 was built upon (Fig. 1a). Instances of partial lysis were difficult to interpret and may represent low efficiency infections, emerging phage resistance, or bacterial lysis resulting from high phage doses without active propagation (killing from without)[47]. We observed an association between phage susceptibility and ST of the ECC isolates, with STs 114, 190 and 1015 being the least susceptible to the cocktail. The modest host-range coverage of 54.2%, along with limited activity against specific STs underscored the need for a more tailored approach to phage isolation and expansion of phage infectivity to ensure comprehensive coverage across broader STs encompassing the ECC nosocomial outbreak.

## Phage adaptation to improve infectivity

A fundamental difference between phages and antibiotics is that phages can evolve and adapt to changes in their hosts. This can be harnessed to improve the fitness and antimicrobial efficacy of select phages[48–50]. Using an experimental evolution approach, we set out to improve the lytic capacity of the three phages from cocktail-V1. Our first goal was to determine the optimum phage adaptation (that is, training) duration that would result in improved lytic capacity. For this, we selected ECC isolate AH17D011 and øEnC07, a phage–host pair with an EOP of 0.79 (Fig. 1i). This phage–host pair was propagated for 24 h, followed by purification of the population of adapted phages, which were used to infect the naive ECC host, with the entire process being repeated for 10 days. From day 5 onwards, we performed growth kinetics assays to compare bacterial growth in the presence and absence of adapted phages to evaluate changes in their lytic efficacy, which are reported as phage scores (Extended Data Fig. 2)[51,52]. Phage scores take growth kinetics data and integrate these into a single value from 0 to 1 (Methods), with higher values representing greater phage fitness and infectivity. We observed that the phage score increased sharply from day 5 of adaptation and plateaued afterwards. On the basis of these observations, we selected 7 days as a sufficient phage adaptation period for further experiments. Importantly, following phage adaptation, evolved phages were twice plaque purified to ensure that a single phage genotype was taken for downstream characterization, including lytic activity via EOPs, growth curve and genomic changes (Fig. 3a).

We then examined whether our phage adaptation protocol could enhance lytic activity against hosts that had lower EOPs and were less responsive to a specific phage. As such, we adapted the remaining two phages from our cocktail: øEnA02 with host AH19K020 (EOP 0.2), and øEnC15 with a non-permissive host CPO390 (EOP below the limit of detection). Following phage adaptation, EOPs improved by 4,000-fold for øEnA02 ($p < 0.001$) and by 8.7-fold for øEnC07 ($p = 0.033$) compared with their ancestor (Fig. 3b,c). Intriguingly, øEnC15, which did

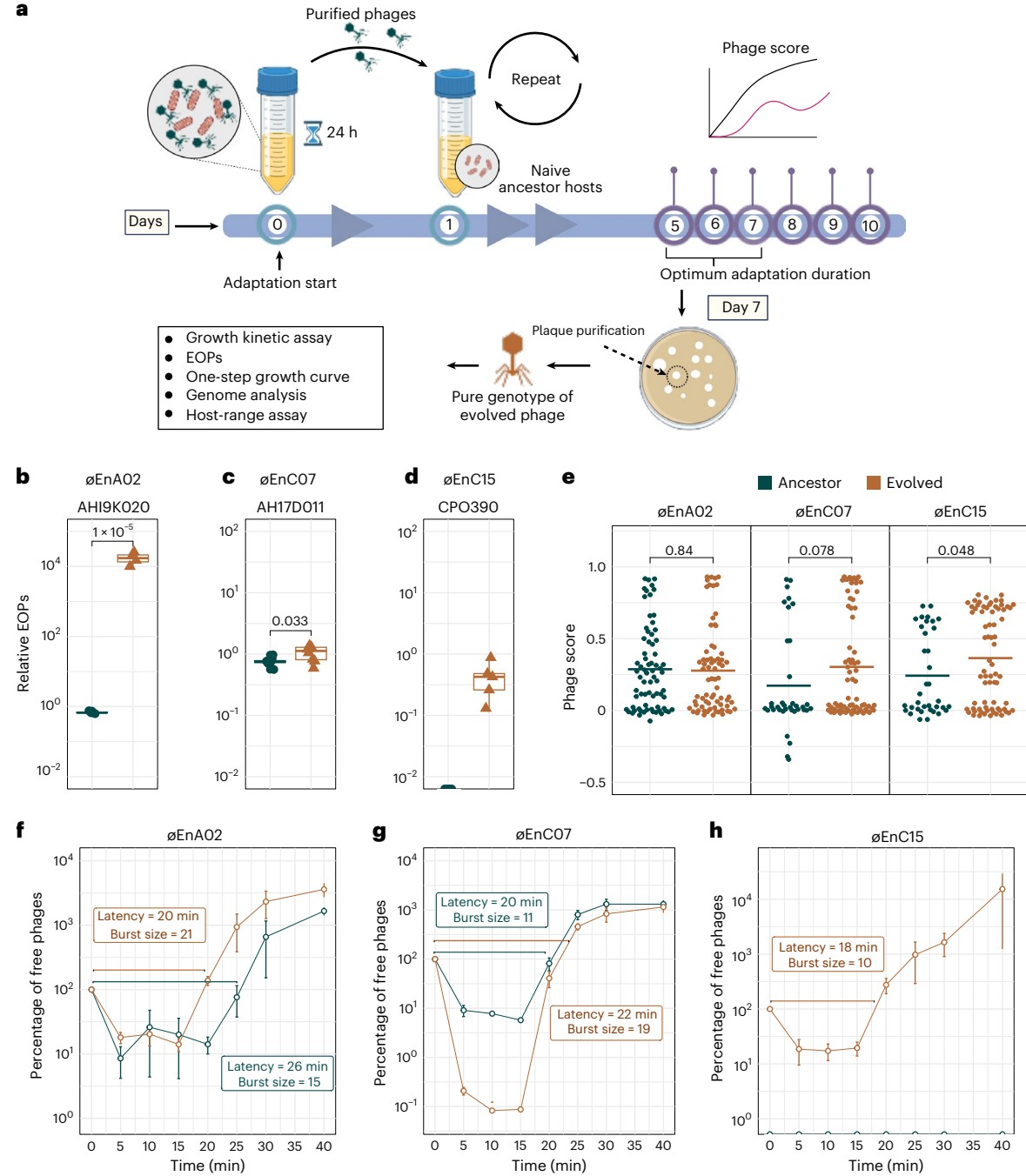

**Fig. 3 | Phage adaptation and its effect on phage infectivity. a**, Experimental protocol for phage adaptation. Ancestral phage was propagated with target hosts for 24 h. Subsequently, the purified phage population was repropagated on a naive population of the same host, with this cycle repeated for 10 days. After 5 days, daily measurements of phage growth curves were conducted using the evolved population to find the optimum adaptation duration. The final evolved phages (that is, day 7) were isolated via two rounds of single-plaque purification to obtain an individual genotype of evolved phage mutants. Then, the growth characteristics of these single-genotype evolved phages were assessed. **b**–**d**, The relative EOPs of evolved phages (øEnA02 (**b**), øE2nC07 (**c**) and øEnC15 (**d**)) compared to their ancestral counterparts. The labels on top of the boxes represent the adaptation ECC host. Each point (*n* = 6) represents data from

independent biological replicates. *p* values were calculated using a two-tailed *t*-test with Welch's correction. Box bounds indicate 25th and 75th percentiles. **e**, Phage score of ancestor phages compared to their respective evolved phages, tested against a subset of 36 ECC isolates. Each dot represents a phage score (*n* = 36) and horizontal lines show means. *p* values were calculated using a two-tailed *t*-test with Welch's correction. **f**–**h**, One-step growth curve of evolved phages (øEnA02 (**f**), øE2nC07 (**g**) and øEnC15 (**h**)) compared to their ancestral counterparts, propagated on their respective adaptation hosts. Data represent mean ± s.e.m of *n* = 3 biological replicates. *Y* axis represents phage titre, normalized to 100% at time 0. Data are displayed on a logarithmic scale (base 10) for visualization. The latency time (in minutes) and burst size (in p.f.u.s per infection) were calculated for each phage. Panel **a** created with BioRender.com.

not produce plaques or show lytic activity via growth assays with host CPO390, was found to infect and propagate at an EOP of 0.4 post adaptation (Fig. 3d). This unexpected finding suggests that phage adaptation may have overcome barriers, such as host defence mechanisms or

surface structures that initially block infection[53,54], warranting further investigation into these underlying factors.

To evaluate the broader fitness of adapted phages and any potential trade-offs, we performed growth kinetics assays to compare the

lytic activity of ancestral phages to their evolved counterparts across our 36 ECC isolate subset, with data reported as phage score (Fig. 3e and Extended Data Fig. 3). The evolved phages showed an increased average phage score for øEnC07 ($p$ = 0.078) and øEnC15 ($p$ = 0.048), while øEnA02 showed minimal change ($p$ = 0.84). To determine whether the observed improvements in the killing efficiency of the evolved phages were accompanied by changes in their life cycles, we conducted one-step kill curves on their adaptation hosts. Compared with its ancestor, the evolved øEnA02 had a faster replicative cycle as demonstrated by a shorter latency period (20 min vs 26 min) and released more progeny virions (burst size; 21 vs 15) (Fig. 3f). Evolved øEnC07 showed an enhanced adsorption rate, as indicated by the steeper decline of free phage percentage in the first 15 min of infection, with a comparable latency (22 min vs 20 min) and larger burst size (19 vs 11) (Fig. 3g). As for the øEnC15, we lacked an ancestral phage that could infect the host for the comparison, so we evaluated the lifecycle parameters of the evolved generation, revealing a latent period of ~18 min and a burst size of 10 (Fig. 3h).

To explore the molecular mechanisms underlying the improved phage efficacy, we conducted genome sequencing and comparative analysis between the evolved phages and their ancestors (Extended Data Fig. 4). We identified several SNPs in the evolved phages, with the majority located in the tail region of the genome, including the tail fibre and receptor-recognizing proteins. These proteins are critical for the initial adsorption of phages to host bacteria. Interestingly, we found that some of the SNPs were unique to certain phage–host pairs, indicating the specificity of the evolutionary process. On the basis of our observation, we hypothesized that the improved phage efficacy resulted from selection and accumulation of beneficial mutations during the adaptation process, which enabled the phages to better recognize and bind to their hosts. These phenotypic improvements and genotypic changes emerged within 7 days. However, this timeline and outcome may vary depending on the specific phage–host combination and should not be generalized. Collectively, these results demonstrate that our phage adaptation approach successfully improved phage infectivity against inefficient or even non-permissive hosts, via mutations in phage tail structures, impacting divergent variables of the phage lifecycle and enhancing killing efficiency.

### Targeted phage isolation against problematic STs

Thus far, we focused on characterizing three phages of our cocktail-V1 and demonstrated their in vitro and in vivo efficacy, followed by phage adaptation to improve their lytic capacity. However, this 3-phage combination had limited effectiveness against specific STs from the ECC collection (Extended Data Fig. 1d). To expand coverage, we selected additional ECC isolates, including the problematic STs 190, 114 and 1015, which lacked sufficient phage coverage, as hosts for targeted phage isolation. Five additional phages were isolated and their activity tested via spot assay (Extended Data Fig. 5a). On the basis of the host-range map and complementary spectrum of activity provided by cocktail-V1, we selected two candidate phages (øNando and øTaquito) for further characterization (Fig. 4a). Genomic and taxonomic classification revealed that both phages belonged to the genus *Pseudotevenvirus* under the *Straboviridae* family (Fig. 4b). Transmission electron microscopy (TEM) revealed phages with an icosahedral head and a long contractile tail, consistent with the characteristics of the *Straboviridae* family (Fig. 4c,d). The genomes of both phages lack transposase or integrase genes, antimicrobial resistance markers and virulence genes, indicating that they are not temperate and are suitable candidates for therapeutic use.

Identification of receptors revealed that both phages targeted different bacterial LPS surface structures to mediate infection (Fig. 4e). For the øNando-resistant host (mutated from isolate CPO448), we found a loss-of-function mutation within *wzzB*, which encodes a protein involved in the biosynthesis of O-antigen and helps determine the

chain length[55]. In addition, for øTaquito-resistant host (isolation host CPO165), we discovered a loss-of-function mutation within the *rfaQ* gene. RfaQ is involved in LPS biosynthesis as it transfers the first two heptose residues in the inner core of LPS, and its loss leads to a severely truncated LPS alongside pleiotropic effects on bacterial cells[44]. We next evaluated one-step growth curves for both phages, with øNando (host: CPO448) demonstrating a latency of 15 min and a burst size of 88 (Fig. 4f) and øTaquito (host: CPO165) demonstrating a latency of 18 min and a burst size of 13 (Fig. 4g). Regarding stability, cleaned phage lysates stored in LB media at 4 °C maintained stability for up to 18 months (Fig. 4h), while room temperature storage supported sufficient phage stability for at least 6 months. These findings, combined with our initial phage results, warranted the inclusion of these two additional phages to our cocktail in an effort to expand host-range coverage against problematic STs that cocktail-V1 did not target.

### Improved 5-phage cocktail achieved broad coverage against clinical ECC isolates

Our first-generation, 3-phage cocktail (cocktail-V1) achieved a modest ~54.2% host-range coverage against the ECC isolates endemic to The Alfred Hospital. To improve its coverage and efficacy, we adapted the original three phages to enhance their lytic activity against selected hosts, followed by a targeted isolation of two phages against problematic STs. Our focus then shifted to identifying the optimal combination of these phages for the development of an improved cocktail. To this end, we prepared eight combinations of phages, each containing either the three ancestral or three evolved phages, along with one or both of the two newly isolated phages (øNando and øTaquito), resulting in 3-, 4- or 5-phage formulations (Extended Data Fig. 5b). We tested these cocktails and a PBS control (9 samples labelled A-I) using a spot assay against an expanded collection of 156 clinical ECC isolates, which included our previous panel ($n$ = 120) (Extended Data Fig. 1d) plus 36 additional ECC isolates that caused infections at The Alfred Hospital during the timeline of this study. Both of our 5-phage combinations, utilizing evolved phages (Combination G: ev_øEnA02, ev_øEnC07, ev_øEnC15, øNando and øTaquito) and ancestral phages (Combination I: øEnA02, øEnC07, øEnC15, øNando and øTaquito) produced complete lysis via spot assays against 65% and 75% of the collection, respectively. This was a marked improvement over cocktail-V1's 43.9% spot lysis on the expanded collection, which included problematic and untargeted STs 190, 114 and 1015 (Fig. 5a and Extended Data Fig. 5b). Next, to resolve whether partial lysis results were indicative of productive infection, we used growth kinetics assays and calculated phage scores. We selected 67 hosts for analysis, including all hosts that produced partial lysis results, along with several hosts showing complete lysis as controls. To distinguish productive infections, an optimum cut-off value of 0.28 in phage score was determined using the maximum-likelihood estimation[56]. Most hosts with partial lysis zones on spot assays yielded productive infections, except for 12 hosts in Combination G and 9 in Combination I. All instances of complete lysis spots were confirmed to be productive, except for one host in Combination G. In addition, no difference in the phage scores was observed between Combinations G and I (Fig. 5b). Combining these results with the spot assay data, we deduced that both Combinations G and I achieved productive infection rates of at least 92% (144 out of 156 isolates) (Fig. 5c). Given our previous demonstration of enhanced fitness and lytic replication in the adapted phages compared with their ancestral counterparts (Fig. 3e), Combination G was selected for manufacturing stable, therapeutic-grade phage product.

### Production of a therapeutic-grade phage cocktail Entelli-02

We manufactured a therapeutic-grade phage product that we named 'Entelli-02' from the 5-phage combination (Combination G) at our in-house facility, the Monash Phage Foundry (Fig. 5d). Importantly, we define a therapeutic-grade phage product as having been produced under institutionally approved guidelines with endpoint quality control

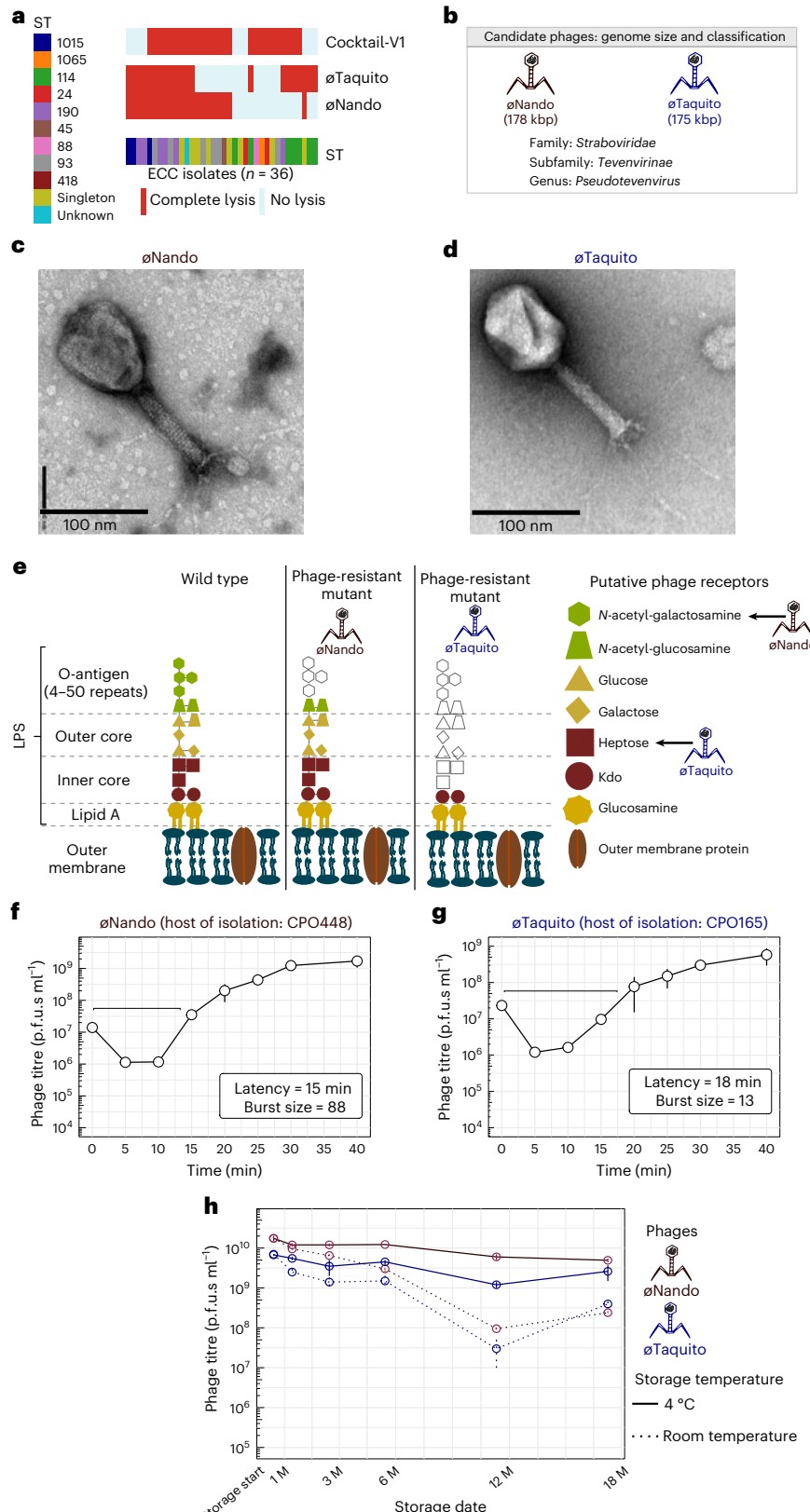

**Fig. 4 | Targeted phage isolation and expansion of host range. a**, Host-range map of isolated phages and cocktail-V1 tested against the initial subset of 36 ECC isolates. Each row represents a phage. STs are colour coded. Singletons are STs that occurred once. Red boxes show complete lysis and light blue boxes show no lysis. **b**, Representations of phage genome size and classification of the phages øNando and øTaquito. **c,d**, TEM images of the øNando (**c**) and øTaquito (**d**) phages. **e**, Schematic representation of the *Enterobacter* LPS and membrane structure, with putative phage receptors indicated by arrows. Colours are for illustrative purposes only and the faded section represents the potential loss of LPS structure in phage-resistant mutants. Figure drawn on the basis of ref. 96 using Inkscape. **f,g**, One-step growth curves of øNando (**f**) and øTaquito (**g**), propagated on their host of isolation. Data represent mean ± s.e.m. of 3 biological replicates. The latency time (in minutes) and burst size (in p.f.u.s per infection) were calculated for each phage. **h**, Stability tracking of øNando and øTaquito over 18 months (M), with storage at 4 °C and room temperature. Data represent mean ± s.e.m. of 2 biological replicates.

measures for sterility, endotoxins, phage activity and phage purity, and being suitable for intravenous administration to patients under Australia's Therapeutic Goods Administration (TGA) Special Access Scheme (Category A). For the first stage of production, each phage was amplified individually through overnight propagation with their respective ECC hosts of isolation. The resulting phage lysates were processed using a two-stage sequential depth filtration (0.5–15 μm and 0.2–3.5 μm retention ratings) to reduce bacterial biomass, followed by sterilizing-grade (0.2 μm) filtration into sealed glass containers. Bacteria-free lysates (~1 l each) were then transferred to a clean-room facility for further processing. Phage lysates were diluted ~10-fold in 1× PBS, followed by buffer exchange and concentration using tangential flow filtration (TFF), resulting in a recovery of 130–160 ml of washed and concentrated lysate from each production run. We observed recovery efficiencies ranging from 24% to 76%, with a p.f.u.s ml$^{-1}$ count exceeding $10^{10}$ p.f.u.s ml$^{-1}$ for each phage (Fig. 5d and Extended Data Table 1).

For stage two of production, we mixed and, if necessary, diluted the five concentrated lysates to achieve a uniform phage product with a titre of >$10^9$ p.f.u.s ml$^{-1}$ per phage in a total of 75 ml (Fig. 5e). We then depleted endotoxin using both EndotrapHD and 1-octanol treatments, resulting in nearly 10-fold reduction in endotoxin levels (Fig. 5f). Finally, we diluted the cocktail 10-fold in 1× PBS supplemented with 1 mM CaCl$_2$ to obtain our final product consisting of an average titre of ~$5 \times 10^8$ p.f.u.s ml$^{-1}$ phage$^{-1}$ in a total volume of 500 ml. The cocktail was twice filter sterilized using sterilizing-grade filters (0.2 μm), followed by quality control validation for phage titre and endotoxin levels (Methods). The cocktail was then packaged into syringe-accessible glass vials, each containing 35 ml of Entelli-02, which is suitable for a 2-week treatment course with an effective dose of 1 ml administered twice daily (b.i.d.). The final packed product underwent external validation for sterility and endotoxin, according to the USP71 and USP85 guidelines, respectively, with >10% of the production batch sent for validation. The final product contained no visible growth of microorganisms and had an endotoxin concentration of 1,575 endotoxin units (EU) ml$^{-1}$, which according to FDA guidelines (5 EU × kg × h) would be safe for intravenous administration b.i.d. to a patient >30 kg (ref. 57). Electron microscopy images confirmed the visual integrity and cleanliness on Entelli-02 (Fig. 5g). Furthermore, considering that these phages were amplified on clinical isolates that contained potential prophages, we performed whole-genome sequencing of the final product followed by read mapping to the individual Entelli-02 phage genomes, the host bacterial genomes and predicted prophage regions. Sequence analysis showed that >99.2% of reads mapped to the Entelli-02 phage genomes, 0.27% to the bacterial genome, and ~0.09% to predicted prophage regions within the clinical production strains (Extended Data Table 2). To further examine the prophage presence, we determined the normalized coverage per million reads (relative abundance) of Entelli-02 phages and predicted prophages. Considering the average titre of ~$5 \times 10^8$ p.f.u.s ml$^{-1}$ phage$^{-1}$ in our product and on the basis of the relative abundance of prophage genomes (~0.01%) (Fig. 5h), we inferred that the level of prophage contamination in Entelli-02 is <$5 \times 10^4$ p.f.u.s ml$^{-1}$. Finally, an important aspect of the chemistry, manufacturing and control process is maintaining the stability of the individual components over time. We measured titres of the individual phages using selective plating within our final Entelli-02 product and found no major loss of titre over 18 months of storage at 4 °C in 1× PBS supplemented with 1 mM CaCl$_2$ (Fig. 5i), except for øEnA02, which showed ~1 log reduction.

## Host range, phage resistance and antibiotic synergy of Entelli-02

We produced a therapeutic-grade Entelli-02 product, which demonstrated broad host coverage against ECC isolates from The Alfred Hospital. To evaluate its clinical relevance, we performed a final host-range screen of our Entelli-02 product against the entire ECC collection from The Alfred Hospital, which had increased to 206 clinical isolates during

the timeline of this study. We conducted spot assays at three concentrations ($10^7$, $10^5$ and $10^3$ p.f.u.s ml$^{-1}$) and assessed relative EOPs. We found that the majority of isolates (34%) were infected by a single phage within Entelli-02, ~25% of isolates were susceptible to two to three phages, 5% to four phages, and just 2% of isolates were susceptible to all five phages, while Entelli-02 failed to infect 12% of isolates (Fig. 6a). EOP heat maps indicated that most infections occurred between 0.1–1, with øEnA02 showing the lowest relative EOP of the five phages (Fig. 6b). Overall, Entelli-02 was able to infect 180 out of 206 ECC isolates at The Alfred Hospital.

Next, we investigated emergence of phage resistance against Entelli-02. To mimic clinical use, we conducted a 5-day in vitro evolution experiment using the five hosts of isolation serially passaged with daily doses of Entelli-02. We measured phage scores of each host with Entelli-02, with lower scores correlating to reduced infectivity against the cocktail, probably due to the emergence of phage resistance (Fig. 6c and Extended Data Fig. 6a). By day 1, four of the hosts had phage scores between 0.7 and 0.95 suggesting that they were still susceptible to the Entelli-02 cocktail, with minimal emergence of phage resistance. Comparatively, host Eaero had a phage score of 0.33 by day 1, reflecting its low-level infectivity by phages in Entelli-02, compared with host APO57, which was highly susceptible to all five phages (Fig. 6d). Over the following 5 days, phage scores remained above 0.6 in four isolates, suggesting limited emergence of phage resistance with Entelli-02 (Fig. 6c).

To further investigate phage resistance impacts on bacterial infectivity and antibiotic interactions, we repeated our phage-resistance evolution experiments to isolate three new and independent phage-resistant mutants, for each component phage from Entelli-02, using their respective hosts of isolation (total of 15 mutants). These phage-resistant mutants were then screened for phage infectivity, which was assessed via phage score against each individual phage and the full Entelli-02 cocktail (Fig. 6d and Extended Data Fig. 6b). Broadly comparing the activity of Entelli-02 across the wild-type and phage-resistant mutants, our data suggest that when multiple phages exhibit strong lytic activity, Entelli-02 maintains efficacy despite the emergence of phage resistance. However, in cases where only one phage dominates the lytic activity (for example, øEnA02, øEnC15 or øTaquito), cross-resistance can reduce Entelli-02 effectiveness. This suggests that relying on one dominant phage increases the risk that resistance to it may compromise the entire cocktail. Other interesting observations include mutants resistant to øEnC07, whose APO57 host was initially sensitive to all five phages, lost sensitivity to øEnC15, øNando and øTaquito, but retained sensitivity to øEnA02. In contrast, øTaquito-resistant mutants were the least sensitive overall. Since øTaquito's isolation host (CPO165) had low sensitivity to other Entelli-02 phages and its resistant mutants exhibited inner core LPS loss (Fig. 4e), this probably impaired receptor availability for subsequent phage infections. Interestingly, øEnA02 showed the highest activity against other phage-resistant mutants probably due to its dual receptors (Fig. 2b,c). These findings highlight the importance of designing phage cocktails with multiple active agents per host and where possible, including phages with diverse receptors to maximize therapeutic robustness and minimize cross-resistance. The use of receptor-diverse phages may offer additional benefits, such as reduced bacterial pathogenesis[58] or increased sensitivity to certain antibiotic classes[59]. Finally, we screened whether our phage-resistant mutants incurred fitness costs[60,61] through comparative growth curves, but did not observe any significant growth defects for phage-resistant mutants under standard growth in LB (Extended Data Fig. 6c).

Next, we evaluated the interaction between Entelli-02 and seven clinically relevant antibiotics (Fig. 6e), including β-lactams (meropenem, imipenem, cefepime, ceftazidime), an aminoglycoside (amikacin), a polymyxin (colistin) and a fluoroquinolone (ciprofloxacin) against the five isolation hosts and 10 phage-resistant mutants (selected from Fig. 6d) using growth kinetics assays. Each condition

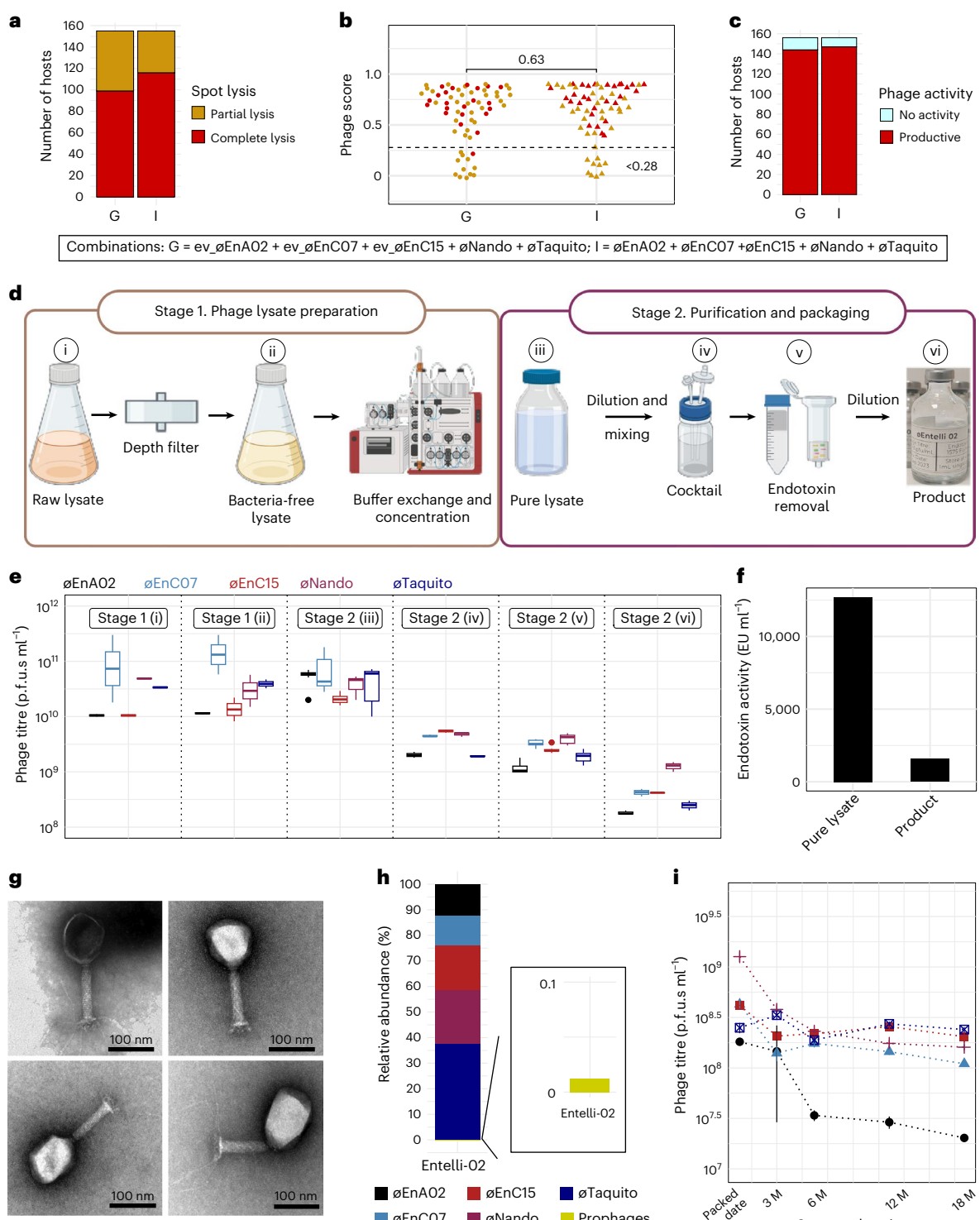

**Fig. 5 | Phage combination screening and manufacturing, and quality control of Entelli-02 phage product. a**, Host-range coverage of phage combinations (G and I) tested against 156 ECC isolates on the basis of spot assay data. **b**, Comparison of phage scores of combinations G and I against ECC isolates. Each dot represents a phage score (*n* = 67). *p* values were calculated using a two-tailed *t*-test with Welch's correction. Dashed line (<0.28) indicates the threshold for characterizing a phage activity as productive. **c**, Host-range coverage (*n* = 156) of phage combinations (G and I), based on combined phage score and spot assay data. **d**, Schematic representation of the two-stage therapeutic-grade phage production protocol developed by the Monash Phage Foundry. **e**, Phage titre at each production stage. Each data point represents results from three biological replicates. Each boxplot color represents a specific phage shown at the top of the graph. Whiskers indicate maximum and minimum values, individual points beyond whiskers are outliers, box bounds indicate 25th and 75th percentiles, and centre line indicates the median. **f**, Endotoxin activity as EU per ml of pure lysate [stage 2 (iii)] and the final product [stage 2 (vi)]. **g**, TEM images of phage product Entelli-02 showing intact virions and no contaminants. **h**, Normalized relative coverage (relative abundance) of sequencing reads from Entelli-02 against each component phage and predicted prophages from the clinical ECC production strains. **i**, Phage titre stability of the packaged Entelli-02 product with storage at 4 °C in 1× PBS supplemented with 1 mM CaCl$_2$ over 18 months (M). Data represent mean ± s.e.m of 2 biological replicates. Panel **d** created with BioRender.com.

included: antibiotic alone (at 0.5× minimum inhibitory concentration (MIC)), Entelli-02 alone (at a multiplicity of infection (MOI) of 0.1), and combinations of both, totalling 315 interactions (Extended Data Fig. 7). Positive interactions indicate that the combination resulted in greater bacterial suppression than either treatment alone. No interaction meant that the combination performed similarly to the single treatments, while negative interactions indicated reduced suppression compared with the single treatments[15]. A limitation of the experiment was that when Entelli-02 achieved near-complete suppression (for example, APO57), it masked potential positive effects from combination treatments (Extended Data Fig. 7). Our analysis revealed that wild-type isolates broadly exhibited more frequent positive interactions with antibiotics than did the phage-resistant mutants. Notably, two wild-type isolates showed positive interactions with all antibiotics tested (Fig. 6e) In contrast, the phage-resistant mutants exhibited variable antibiotic interaction profiles, but among them, β-lactams demonstrated most positive interaction: 6 of 10 mutants interacted positively to ceftazidime, 4 to meropenem, and 3 each to imipenem and cefepime. This suggests that β-lactams could be promising partners for combination therapy with Entelli-02. In contrast, negative interactions with colistin were observed in 5/10 mutants. Colistin acts by electrostatically interacting with negatively charged LPS molecules, suggesting that our phage-resistant mutants, which have mutations in LPS-associated genes, probably impaired colistin's efficacy[62], further suggesting that colistin should be used cautiously in combination with phages, as has been previously described[63,64]. In addition, two negative interactions were observed with ciprofloxacin and phage-resistant mutants, while no interactions were found with amikacin despite positive effects seen in wild-type isolates, which may be associated with changes in cell permeability[65]. In summary, our Entelli-02 product broadly slowed the emergence of phage resistance and demonstrated positive interactions with clinically relevant antibiotics, particularly β-lactams, with some phage-resistant mutants exhibiting potential antibiotic resensitization events that could be exploited further clinically.

### Preclinical evaluation of Entelli-02

After establishing the in vitro efficacy of Entelli-02 and characterizing its phage-resistance dynamics and antibiotic synergies, we proceeded to examine its efficacy in a murine infection model. We first replicated our previously established infection model with isolate APO57, which is sensitive to all five phages, by comparing the original 3-phage cocktail (cocktail-V1) with Entelli-02. Consistent with previous results, both cocktails reduced bacterial burden by ~3-logs in the blood, kidney, liver and spleen compared with the PBS control group ($p < 0.05$), with no differences between the two ($p > 0.05$) (Fig. 6f). Phage load analysis revealed that øEnA02 was the dominant phage in vivo, mirroring in vitro growth kinetics data where øEnA02 demonstrated the highest growth inhibition against APO57 (Extended Data Fig. 8a,b). Next, we used a contemporary clinical isolate AALF22D176 from The Alfred Hospital

that was naïve to both cocktails, meaning none of the phages in cocktails were isolated or screened against it. This isolate represents a common sequence type (ST190) in the Alfred Hospital's collection and was susceptible to all five phages in Entelli-02, although at different inhibition rates compared with APO57 (Extended Data Fig. 8d,e). While both cocktail-V1 and Entelli-02 suppressed bacterial load compared with the control ($p < 0.05$), Entelli-02 showed superior efficacy compared with cocktail-V1 ($p < 0.05$) (Fig. 6g). Phage replication also differed between the cocktails, with øEnC15 dominating in cocktail-V1-treated mice ($p < 0.05$), while øNando expanded the most in Entelli-02-treated mice ($p < 0.05$), which was consistent with in vitro growth kinetics (Extended Data Fig. 8d,e). These results demonstrate that the 5-phage Entelli-02 cocktail is comparable to the original 3-phage combination against host APO57 and offers improved bacterial suppression and phage propagation capacity in vivo against contemporary isolates that more accurately reflect clinical scenarios.

## Discussion

Currently, the lack of standardized and readily available phage products for use against nosocomial multidrug-resistant pathogens is a major impediment to administering effective phage therapy in healthcare settings[66]. Here we employed a unique approach that leverages the utility of broad-spectrum phage cocktails with the increased efficacy and specificity found in personalized phage therapy. We define this approach as an 'institutional cocktail' that has been designed upon a high-quality and representative collection of nosocomial pathogens, and that can be employed locally and rapidly as a frontline therapeutic with high probability of antimicrobial activity. Our integrated academic-clinical approach was personalized towards the endemic pathogen profile at the institutional level (that is, The Alfred Hospital) and aimed to improve both the treatment efficacy and response time for future phage therapy cases. Through this, we developed an institutional cocktail, which we named Entelli-02, to serve as a frontline therapeutic against AMR ECC infections at The Alfred Hospital, in Melbourne, Australia. We envision this approach being employed to treat other nosocomial pathogens, expanded across additional institutions, and iterated upon over time to improve product efficacy towards an everchanging pathogen population.

Personalized phage therapy approaches have reported clinical improvement in ~75% of patients, bacterial eradication in ~60–80% of patients, and adverse events in ~15% of patients[8,15]. However, personalized phage therapy applications can be time consuming, complex to administer and difficult to scale, which limits the number of patients treated. Conversely, broad-spectrum phage cocktails designed to treat entire pathogen complexes as stand-alone therapies have been produced under current pharmaceutical Good Manufacturing Practices (GMP) guidelines and can be made readily available. Yet the performance of these defined cocktails in randomized controlled trials completed so far has been underwhelming[2,8–12], hinting at a disconnect between the strains

---

**Fig. 6 | Preclinical evaluation of Entelli-02. a**, Percentage host coverage of the 5 individual Entelli-02 phage components against 206 ECC isolates determined by plaque formation. **b**, EOP heat map for Entelli-02 phage components. Black bar indicates inferred overall host coverage (any detectable EOP) by Entelli-02. EOP scale divided at 0.001–0.1 and 0.1–1 for visual clarity. **c**, Comparison of phage scores following 5 days of serial passage of Entelli-02 on each host of isolation. Points represent mean phage scores ($n = 6$) from each 24-h kinetic growth measurements ($OD_{600}$). **d**, Heat map comparing cross-resistance patterns of wild-type and 3 phage-resistant clones (R1–R3) against individual phages and Entelli-02. The right panel indicates the original phage and their host of isolation that was used for generation of resistance mutants. Dotted boxes highlight instances of complete cross-resistance (phage score <0.1). **e**, Interaction profiles between phages and antibiotics for wild-type and phage-resistant mutants (two clones; R1 and R2), tested against seven antibiotics from four different classes. Interactions are categorized as positive, negative or none on the basis

of combined effects on bacterial growth. **f,g**, Bacterial burden in tissues of mice infected with ECC isolates APO57 (**f**) or AALF22D176 (**g**), comparing untreated controls versus cocktail-V1 or Entelli-02 treatment. Points represent individual mice ($n = 6$ per group). Boxplots show median (centre line), interquartile range (box edges: 25th–75th percentiles), the most extreme values within 1.5× interquartile range (whiskers), and outliers as individual points beyond whiskers. The total represents all samples combined ($n = 24$). Statistical analysis was performed using Mann–Whitney $U$-test (two-sided) with exact $p$ values to compare medians of bacterial count between groups. The dashed line indicates the LOD. Right: phage propagation in tissues of treated mice. Each point represents phage titres from individual tissue samples ($n = 24$ total) of 6 mice, displayed as boxplots with medians and interquartile ranges as above. $p$ values shown for phages with significant propagation; complete statistical comparisons in Extended Data Fig. 8c,f.

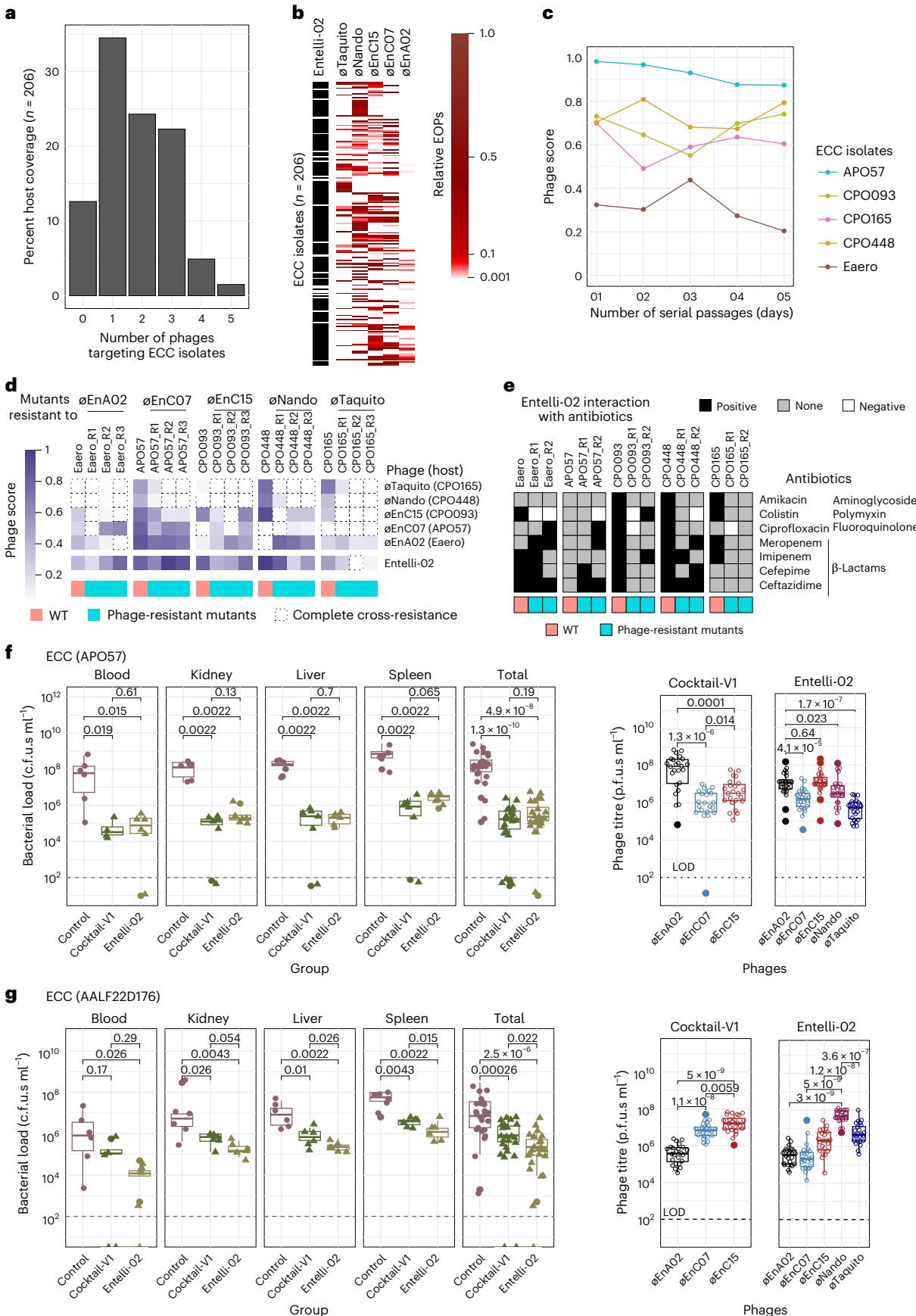

used to develop these products and the clinical strains being treated in patient cohorts. While the mixing of phages that target multiple strains or species is a traditional and widely used method, several factors can compromise cocktail efficacy[8]. The use of limited isolate collections or clinically irrelevant hosts may result in poor coverage and inefficient killing, and bacteria can quickly evolve phage resistance. Importantly, few broad-spectrum cocktails identify the phage receptors their constituents use[67]. Thus, it is important to implement a rational approach for the selection of phages in a cocktail, ensuring a broad host range, high efficacy and minimal risk of resistance development[67–70].

For the iterative development of Entelli-02, we began by isolating phages against 36 genetically diverse isolates of ECC, representative of the ECC outbreak at The Alfred Hospital. We formulated a first-generation 3-phage combination (cocktail-V1) on the basis of orthogonal host-range coverage, which showed 3 $\log_{10}$ reduction of ECC in our in vivo mouse model. However, screening of cocktail-V1 against 120 clinical ECC isolates revealed only 54% host coverage, with low killing efficiency against a few particular STs. To improve our phage cocktail, we applied two approaches. First, we conducted phage adaptation[48–50] to improve the infectivity of phages against target hosts[71–74], followed by isolation of single phage genotypes, which were characterized on the basis of improvements in phage score[75,76]. Second, we employed targeted phage isolation against problematic STs that were not covered by cocktail-V1, adding two additional phages, both of which recognized distinct components of the LPS[77,78]. Both of these approaches enhanced our cocktail's ability to target and infect ECC isolates, resulting in a widened host range for Entelli-02, covering the majority of the STs from The Alfred Hospital.

In preparation for clinical administration, we manufactured a therapeutic-grade Entelli-02 cocktail and ensured final product stability. The conclusive assessment of Entelli-02's therapeutic potential was supported by its plaque-forming capacity in 88% of clinically, genetically and phenotypically diverse ECC isolates. Supporting its clinical utility, Entelli-02 demonstrated potent efficacy in a murine infection model, where it significantly reduced bacterial colonization of a newly isolated representative ECC (ST190) at The Alfred Hospital. Notably, this isolate had not been used during the cocktail's development or testing, simulating a realistic therapeutic challenge and confirming the cocktail's frontline capacity. Moreover, Entelli-02 remained effective even in the context of evolving bacterial resistance and demonstrated positive interactions with β-lactam antibiotics. While individual phages in the cocktail may face stability issues[11], they remained stable as part of the Entelli-02 formulation for up to 18 months, further supporting the feasibility of Entelli-02 as a frontline clinical treatment.

In summary, our study represents a comprehensive and multifaceted approach to generating an 'institutional cocktail', Entelli-02, that bridges the gap between personalized phage therapy and broad-spectrum phage cocktails. Developed in response to a local hospital's outbreak and on the basis of a clinically curated isolate collection, Entelli-02 is designed to combat an endemic AMR pathogen in a nosocomial setting as a frontline therapeutic. Importantly, this 'institutional cocktail' offers a streamlined alternative to the extensive processes typical of personalized phage therapies, and leverages advantages from both single-patient and broad-spectrum phage cocktails, setting the blueprint for establishing 'institutional cocktails' against other AMR pathogens across different hospitals and institutions. Although our phage cocktail has demonstrated effectiveness against isolates obtained from a single health centre, it is important to acknowledge that its efficacy may vary across geographical and clinical settings. Therefore, pre-treatment assessment of phage activity against individual patient isolates remains essential to avoid ineffective use. The ~12% of isolates not covered by Entelli-02 underscore the need for continued isolate surveillance and cocktail refinement. Alternatively, a personalized phage therapy approach, which utilizes individual phages that are pre-produced in quality-controlled batches, would

enable modular treatment strategies with similar benefits to Entelli-02 in their speed and precision for therapeutic applications. Further work is needed to determine the breadth and activity of our ECC phage cocktail against both national and international ECC isolates, including conducting clinical trials to evaluate its efficacy and safety in treating AMR ECC infections at The Alfred Hospital. Finally, this approach can be iterated and adapted to the changing genetic background of ECC clinical isolates at The Alfred Hospital, allowing us to produce future improved versions of the Entelli-02 cocktail.

## Methods

### ECC clinical isolates
This study used clinical isolates of ECC from The Alfred Hospital, Melbourne, Australia, collected between 2010 and 2024. In total, the repository at The Alfred Hospital contained 206 bacterial isolates. For initial phage isolation and analysis, we selected a subset of 36 isolates to represent the ST diversity of the whole collection set. To further the scope of our investigation, the study was expanded to include up to a total of 206 isolates to evaluate phage and cocktail activities. No patient data were collected or used in these experiments and all isolates were retrieved without identifiable patient data. Isolate details are listed as Supplementary Data 1.

### Bacterial strains, plasmids and growth conditions
We used lysogeny broth (LB) (10 g l$^{-1}$ tryptone (Oxoid), 5 g l$^{-1}$ yeast extract (Merck), 10 g l$^{-1}$ NaCl (Merck)) to grow bacteria, unless otherwise mentioned. Agar (Merck) at a concentration of 15 g l$^{-1}$ was employed for agar plates and 7.5 g l$^{-1}$ for the creation of a soft agar double-layer (top agar). All bacterial cultures were grown aerobically at 37 °C overnight. We used Omnipur phosphate-buffered saline (PBS) (Merck) for all experiments unless otherwise mentioned. In gene transformation experiments, we used vector pBBRMCS[79] and E. coli DH5-α competent cells (C2987H, New England BioLabs (NEB)) as a host.

### Phage isolation and purification
Phages were isolated from environmental sources through an enrichment process[80]. Briefly, 3–4 batches of raw sewage were combined in a flask, totalling up to 100 ml, followed by addition of 10 ml of 10× LB along with supplements of 1 mM CaCl$_2$ and MgCl$_2$. Next, 1 ml overnight cultures of up to three targeted bacterial host isolates were added into the mixture and incubated at 37 °C overnight. The resulting lysates were purified by centrifugation, filtration and chloroform treatment[81]. Phage activity was assessed by spotting 10 μl of lysates onto bacterial lawns on soft agar. When bacterial lysis was observed, we plated serial dilutions of the lysate with the host bacteria using the soft overlay (top) agar method and examined for clear plaques on the bacterial lawn after overnight incubation. For two successive rounds, plaques were individually picked from each plate, resuspended in 500 μl of PBS by vigorous vortexing, and reamplified. Lysates prepared from the purified plaques were quantified in plaque-forming units (p.f.u.s) per ml and stored at 4 °C.

### Determination of host range, efficiency of plating, one-step growth curve and adsorption assay
Host range was examined by spot assays. We applied 10 μl of phage lysate (≥10$^8$ p.f.u.s ml$^{-1}$) onto a bacterial lawn prepared by mixing 1 ml of bacterial culture with 3 ml of soft agar and spreading over an LB agar plate. For the initial host-range assessment on the host subset ($n = 36$), spot assay results were further confirmed for their ability to form plaques, verifying productive infection. For screening in broader sets ($n = 120$ and $n = 156$), the resulting zones were recorded as complete (phage clearing bacteria), partial (bacteria partially lysed with hazy lysis zone) or no lysis (no activity). Three researchers assessed these results blindly to each other and to the phage combinations used (Extended Data Fig. 3d). EOP was examined by comparing p.f.u.s ml$^{-1}$ on the host of interest to p.f.u.s ml$^{-1}$ on the original isolation host. Phage lysates were serially diluted, mixed with each host, and plated using top agar to determine

the p.f.u.s ml$^{-1}$. For broad host screening ($n$ = 206), phage lysates with an initial titre of $10^8$ p.f.u.s ml$^{-1}$ (on host of isolation) were serially diluted to $10^{-1}$, $10^{-3}$ and $10^{-5}$, and spotted onto lawns of each bacterial isolate. EOP values were calculated relative to the phage titre $10^8$ p.f.u.s ml$^{-1}$. For one-step growth curves, bacteria from overnight broth cultures were diluted 1:50 in LB and grown until an optical density at 600 nm ($OD_{600}$) of 0.2 was reached (corresponding to ~$10^8$ c.f.u.s ml$^{-1}$) (~2 h). The culture was then infected with phage at an MOI of 0.1. Phage was allowed to adsorb for 5 min at 37 °C with orbital agitation at 190 r.p.m. The mixture was then pelleted (4,000$g$, 2 min, room temperature), resuspended in fresh pre-warmed LB broth, and incubated. Samples (100 µl) were collected every 5 or 10 min for 1 h, serially diluted in PBS, and plated for phage quantification. The experiment was repeated in at least three different occasions. Using these data, we calculated the latent period and burst size as described previously[82]. Adsorption assays were performed on the basis of the principles of a one-step growth curve, with some modifications. Phages were mixed with their bacterial hosts at an MOI of 0.1 and samples were taken at 0, 5 and 10 min. Each sample was immediately diluted 1:10 in PBS and centrifuged to remove adsorbed phages; the supernatant was plated for phage quantification and calculation of the changes in free phage concentration.

### Isolation of phage-resistant mutants
To generate phage-resistant mutants, we followed the broth culture method[35]. First, we incubated phages with their respective isolation host at an MOI of 1 in a 96-well microtitre plate, with a total volume of 200 µl each, in 9 replicates. After an overnight incubation, we transferred 20 µl of the resulting growth to fresh LB containing $10^6$ p.f.u.s ml$^{-1}$ of phage particles. We repeated this process for 3 consecutive days.

After 3 days, we purified the supposedly resistant bacterial colonies through two rounds of single-colony isolation and confirmed phage resistance by top agar assays using the phage-resistant mutants and their respective phages, expecting an absence of plaques after incubation. Subsequently, we performed adsorption assays to further confirm phage resistance. Stocks of the verified mutants were stored at −80 °C until further experimentation.

### TEM
For examination through TEM using negative staining, phage lysates were purified using Vivaspin 6 centrifugal concentrators (MWCO 1,000,000 kDa, Merck). The lysate was washed several times with SM buffer (100 mM NaCl, 8 mM MgSO4·7H2O, 50 mM Tris-Cl in water). The cleaned lysate was then applied onto a copper TEM grid (200 mesh, SPI) with a carbon-coated ultrathin formvar film. A 10-µl droplet of phage suspension was placed on the grid, left undisturbed for 30 s and then dried using filter paper. Next, a 10-µl droplet of uranyl acetate–water solution (1% w/v) was added to the grid surface and left for 20 s. After drying the grid with filter paper, it was examined under TEM (JEM-1400 Plus, Jeol), operating at an accelerating voltage of 80 kV.

### Phage adaptation
Exponentially growing bacteria and natural (ancestor) phage were mixed at final concentrations of ~$2 × 10^6$ c.f.u.s ml$^{-1}$ and $2 × 10^5$ p.f.u.s ml$^{-1}$ (MOI 0.1), respectively, in a 96-well microtitre plate with a total volume of 200 µl per well. Incubation followed at 37 °C, with shaking at 190 r.p.m. for 24 h. Then, we added 0.1 volumes of chloroform to each well, followed by incubation for 10 min with occasional mixing to terminate bacterial activity. We centrifuged the microtitre plate at 4,000$g$ for 10 min to pellet the chloroform and bacterial debris. We then transferred the phage-containing supernatant to the corresponding well of a second microtitre plate containing evolutionarily naïve exponentially growing bacteria. We repeated this process daily for up to 10 days.

Bacterial growth kinetics curves (bacterial lysis by phage) with ancestral or evolved phages were obtained from sequential optical density ($OD_{600}$) readings using a microtitre plate reader (Epoch Microplate

spectrophotometer, BioTek). We added bacterial cultures at exponential growth to individual wells of a 96-well microtitre plate and mixed them with phage lysate at an MOI of 1. The plate was incubated at 37 °C in a microtitre plate reader with continuous shaking and data were recorded every 15 min for up to 24 h. The phage score was calculated using the bacterial areas under the growth curve (AUC) with and without phage[51,52].

$$\text{Phage score} = \frac{(\text{Average AUC}_{\text{without phage}} - \text{Average AUC}_{\text{with phage}})}{\text{Average AUC}_{\text{without phage}}} \quad (1)$$

After choosing the focal day for each evolved phage, we purified a single phage genotype from the population by double single-plaque purification.

For the 5-day in vitro cocktail resistance evolution experiment, each bacterial host from which the 5 phages of the Entelli-02 were isolated was mixed with the Entelli-02 at an MOI of 1 in a 96-well microtitre plate with a total volume of 200 µl per well, followed by sequential $OD_{600}$ readings as described above. Following a 24-h incubation, 20 µl of each cocktail–bacteria mixture was transferred to fresh medium containing ~$10^6$ p.f.u.s ml$^{-1}$ of the Entelli-02. This process was repeated for five sequential passages, with phage scores from growth kinetics determined daily for each passage.

### Phage stability
To examine the stability of phages on long-term storage, we stored clean phage lysate in LB in 500 µl volumes in 1.7 ml Axygen polypropylene microcentrifuge tubes (Corning). Clean lysates were lysates purified by centrifugation, filtration and chloroform treatment as mentioned above[81]. These tubes were stored at ambient room temperature (~22 °C), 37 °C and 4 °C. To track any change in stability, phage count was performed at the interval of 1, 3, 6 and 12 months, spanning up to 30 months.

### Bacterial and phage genomics and bioinformatics analysis
To extract bacterial DNA, we revived the bacteria from frozen stock and prepared overnight cultures. A 1.5 ml overnight culture was used for DNA extraction using the GenElute Bacterial Genomic DNA kit (Sigma-Aldrich) following manufacturer protocol. For phages, we treated 1 ml of phage lysate with a concentration of >$10^8$ p.f.u.s ml$^{-1}$ with DNase (1 mg ml$^{-1}$) and RNase A (12.5 mg ml$^{-1}$) for 2 h at 37 °C, followed by an inactivation for 5 min at 75 °C. Phage DNA extraction was performed using the Norgen phage DNA kit (Norgen Biotek) following manufacturer instructions. For the cocktail (Entelli-02) genome, we extracted the genome from the final packed product following the same protocol mentioned above, excluding DNase and RNase treatment. We sent bacterial and phage DNA to Azenta Life Sciences in Suzhou, China for indexing and sequencing on an Illumina HiSeq platform.

The raw sequencing reads were trimmed using Trimmomatic (v.0.39)[83], followed by assembly using Unicycler (v.0.4)[84], annotation using Prokka (v.1.13.1)[85], NCBI PGAP[86] and the RAST annotation server[87]. Snippy v.4.0 was used to identify nucleotide variation and SNPs. Phage genomes were assembled into complete genomes and aligned through pairwise sequence alignment using Pyani (https://huttonics.github.io/pyani/). Complete nucleotide sequences were rearranged to match the start gene position of the phage genome. Phages were assigned to the same genus if they had >95% average nucleotide identity (ANI) with other known phages listed in the International Committee on Taxonomy of Viruses (ICTV) database. Phage genomes were examined against antibiotic resistance and virulence factor databases using the ABRicate tool (v.1.0.1), and integrase genes in phage annotations were investigated. To analyse prophage contamination in Entelli-02, prophage sequences on host genomes were predicted using VirSorter2 (ref. 88) and verified manually to exclude falsely categorized or low-quality prophages with <5,000 bp. The raw reads (fastq) of Entelli-02 were

mapped with component phages' genomes, respective production hosts and predicted prophages using Bowtie2 (ref. [89]) and Samtools[90] to obtain the ratios of read coverage and relative abundance (normalized coverage per million reads) as an indicator of prophage and host genome contamination.

## Complementation assay

We designed primers with suitable cut sites for the restriction enzymes targeting genes of interest (Supplementary Table 1). These primers were used to PCR-amplify the respective genes. Next, we performed two overnight double-digestion reactions at 37 °C, employing the restriction enzymes, for each gene of interest and the vector pBBRMCS[79]. The digested products were purified through gel electrophoresis on a 0.7% agarose gel, followed by excision and ligation for 2 h at room temperature using the Instant Sticky-end Ligase Master Mix (NEB) in accordance with manufacturer protocol. The ligated product was then chemically transformed into *E. coli* DH5-α competent cells (NEB) following company protocol. The transformed colonies were confirmed on antibiotic selection-LB plates and subsequently validated by Sanger sequencing. The plasmid containing the desired complement was extracted from the confirmed colonies using GenElute Plasmid Miniprep kit (Sigma-Aldrich). The plasmid containing the required complement was then electroporated into the phage-resistant mutants. To verify the restoration of phage infectivity, we conducted top agar and adsorption assays, with the anticipation of observing plaque formation and virion adsorption.

## In vivo assays (murine phage therapy)

Our mouse model was based on an established protocol using intraperitoneal injection[91,92], which is highly effective for establishing infection, with bacterial colonization reported as early as 20 min post infection[93]. We employed ECC isolate APO57 to establish a mouse infection model and determine the optimal bacterial density required to induce severe septicaemia. For the experiment, the bacterial inocula were prepared from an 18-h culture, washed with PBS and adjusted to a concentration of $1 \times 10^8$ c.f.u.s ml$^{-1}$ (~0.2 OD$_{600}$). We used female 8-week-old BALB/c mice (18–20 g) for these experiments, with each mouse receiving either $1 \times 10^6$ or $5 \times 10^6$ bacterial cells, mixed with 6% porcine stomach mucin in PBS (1:1), for a total volume of 200 µl. Mice (*n* = 3 per group) were injected intraperitoneally and closely monitored for up to 12 h. When the humane endpoint or 12 hpi, whichever occurred earlier, was reached, blood was collected via cardiac puncture, and a laparotomy was performed to obtain a liver section, right kidney and spleen. The organs were weighed, homogenized in PBS, and plated to enumerate bacterial and phage counts, which were then normalized by organ weight or blood volume.

Following the establishment of the infection model, we proceeded with the phage treatment experiment. In this phase, each mouse group received a dose of 200 µl of bacterial suspension at $5 \times 10^6$ c.f.u.s ml$^{-1}$. At 1 hpi, the treatment group (*n* = 6) received 200 µl of PBS containing $1 \times 10^8$ p.f.u.s ml$^{-1}$ of phage, whereas the control group (*n* = 6) received an equivalent volume of sterile PBS. At the established endpoint, blood and organs were collected for bacterial and phage count.

## Phage production

Phage production was completed in two stages. In stage 1, each phage in the cocktail was amplified individually with its respective host in a 1,000-ml batch. Here, 50 ml of an overnight culture of the host bacteria was added to 1,000 ml LB broth supplemented with 1 mM CaCl$_2$ and MgCl$_2$. LB contains tryptone (Oxoid) which is an animal pancreatic digest of casein. This mixture was allowed to grow for ~2 h at 37 °C with shaking until reaching an OD$_{600}$ of ~0.2. At this point, phages were added to the mixture at an MOI of 0.1 and allowed to propagate overnight. Next, the lysate was clarified using sequential depth filters with retention rating of 0.5–5 µm (SUPRAcap 50 Pall Depth filtration capsule SCO50PDH4, Pall, 10423832) and 0.2–3.5 µm (SCO50PDE2, Pall, 10439722) to remove bacterial aggregates. Pressure was monitored to ensure it did not exceed 2 bar

to maintain filter specifications (with manufacturer's maximum limit rating of 3 bar). The filtered lysate was then sterilized into sealed glass bottles within a biosafety cabinet through a sterilization-grade filter with a 0.2-µm removal rating (Supor Mini Kleenpak capsules, KA02EKVP2G, Pall) to ensure complete sterilization of the lysate. In stage 2, we purified and packaged the phages in a separate lab environment that was free from bacterial culturing. We used TFF with buffer exchange using the ÄKTA Flux 6 system (29038438, Cytiva) equipped with a 300-kDa nominal molecular weight cut-off (NMWC) microfiltration hollow fibre cartridge (UFP-300-E-4MA, Cytiva). Briefly, the sterile lysate (~900–1,000 ml) was diluted up to 10-fold with 1× PBS and processed through TFF. This process was continued until the lysate lost its yellowish colour, indicating replacement of the bacterial media with the buffer, and the lysate was concentrated down to ~150 ml final volume. The lysate was then collected and resterilized using a 0.2-µm filter as mentioned above. After clean-up and verification of the titre, the lysate of each phage was mixed, or diluted in 1× PBS as needed, to obtain a cocktail with a target final phage concentration of >$10^8$ p.f.u.s ml$^{-1}$ for each phage component. To clean endotoxins from the cocktail, we initially applied an EndoTrap HD affinity column (LET0035, Lionex) following manufacturer instruction. However, the endotoxin level remained above the acceptable threshold. We therefore performed a 1-octanol purification, followed by removal of any residual octanol using two rounds of ultracentrifugation at 4,000$g$ (ref. [81]). The endotoxin concentration at different stages of the purification process was assessed using the EndoZyme II (22599, Hyglos) and Endosafe PTS cartridge (PTS20F, Charles River) kits. Once a satisfactory endotoxin level was achieved, the preparation was again filter sterilized and packed in 35-ml batches in sealed sterile glass vials (FILL-EASE SV-50C02, Huayi Isotopes) with syringe access. The vials were labelled and stored at 4 °C for clinical use. Third-party independent validation of sterility (US Pharmacopeia, USP 71) and endotoxin (USP 85) for each production batch was performed by Eurofins BioPharma Product Testing (Sydney, Australia). Phage titration was performed to verify the phage recovery throughout the process and product stability during storage.

## Phage–antibiotic interactions

We selected 7 clinically relevant antibiotics from 4 different classes: β-lactams (ceftazidime [third-generation cephalosporin], cefepime [fourth-generation cephalosporin], imipenem [first-generation carbapenem] and meropenem [second-generation carbapenem]), an aminoglycoside (amikacin), a polymyxin (colistin), and a fluoroquinolone (ciprofloxacin). All antibiotics were obtained from Sigma-Aldrich, except cefepime, which was sourced from the European Pharmacopoeia. MICs were determined using the broth microdilution method following the Clinical and Laboratory Standards Institute (CLSI) protocol[94]. Phage–antibiotic–bacteria growth kinetics were analysed by measuring bacterial growth (OD$_{600}$) as described above. Each wild-type strain and its two phage-resistant mutants (R1 and R2) were assessed under three conditions: (1) phage cocktail Entelli-02 only at an MOI of 0.1, (2) antibiotics only (at 0.5× MIC) and (3) Entelli-02 (0.1 MOI)–antibiotic (0.5× MIC) combinations. Experiments were performed in 96-well microtitre plates with a final volume of 200 µl. Bacterial cells were inoculated at a concentration of $5 \times 10^5$ c.f.u.s ml$^{-1}$ per well following the CLSI guideline for antibiotic susceptibility tests. Interactions were classified as positive when the combination resulted in greater bacterial suppression than either treatment alone as measured by area under the curve. No interaction meant that the combination performed similarly to the single treatments, while negative interactions indicated that the combination performed reduced suppression compared with the single treatments. A limitation of the experiment was that when Entelli-02 achieved near-complete suppression, it masked potential positive effects from the combination.

**Statistical analysis.** Statistical analysis was performed using R 4.2.3 and GraphPad Prism v.9.1. Mean and standard error of the mean, median, percentage and area under the curve were calculated as appropriate,

and statistical tests were performed accordingly and reported in figures where appropriate. Two-sided *p* values were calculated and reported where applicable. The cut-off for the phage score was calculated using maximum-likelihood estimation based on the receiver operating characteristic curve implemented online[56]. Graphs were drawn using ggplot2 in R 4.2.3 and figures were drawn using Inkscape 1.3.2 and BioRender.

**Ethics statement.** All protocols involving animals were reviewed and approved by the Monash University Animal Ethics Committee (Project ID: 2022-27681-80733), and the animals were housed at the Monash Animal Research Platform, Monash University, under standard laboratory conditions: 20–22 °C temperature, 30–70% relative humidity and 12:12 h light/dark cycle. We received all the bacterial isolates from the Department of Infectious Diseases at The Alfred Hospital without identifiable patient data. We conducted all procedures in strict adherence to institutional guidelines.

### Reporting summary
Further information on research design is available in the Nature Portfolio Reporting Summary linked to this article.

## Data availability
All data (supplementary data and figures) and source data that were generated during this study are deposited in GitHub at https://github.com/ECCphage (ref. 95). Genome sequences generated during this study are deposited at the NCBI database under project ID PRJNA629076. Genomes of ECC isolates of The Alfred Hospital are available at the NCBI database under project ID PRJNA924056. Source data are provided with this paper.

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

## Acknowledgements

This work, including the efforts of J.J.B., was funded by the National Health and Medical Research Council (NHMRC: 1156588 and 2026130). We thank J. Li (Monash University) for providing plasmids used in this study; D. McCarthy for raw sewage samples for phage isolation; and L. Kan and N. Smith for help in blind interpretation of spot assay results. We acknowledge the Monash Animal Research Facility (MARP) for providing the necessary facilities for animal experiments; the MASSIVE HPC facility (www.massive.org.au) for providing us with cluster time for data analysis; and the Monash Ramaciotti Cryo-EM platform for the use of facilities.

## Author contributions

D.S., A.Y.P. and J.J.B. conceived of the study. D.S., F.G.A. and A.P. performed phage isolation. D.S. performed phenotypic and genotypic characterization of phages, including all bioinformatics. D.S. and R.P. planned and performed complementation experiments. D.S. and R.D. performed phage evolution experiments. D.S. performed the animal experiment, with help from F.G.A. (pilot study) and X.K. (pilot study). F.G.A. and L.B. performed spot assay and growth kinetics at The Alfred Hospital. N.M. provided isolate information. D.K. performed TEM. J.J.B. and A.Y.P. supervised the project. J.J.B. funded the project. D.S. and J.J.B. wrote the original draft. All authors were involved in reviewing and editing the final version of the paper.

## Funding

## Competing interests

The authors declare no competing interests.

## Additional information

**Extended data** is available for this paper at https://doi.org/10.1038/s41564-025-02130-4.

**Correspondence and requests for materials** should be addressed to Dinesh Subedi or Jeremy J. Barr.

**Extended Data Table 1 | Volume and phage titre recovery during phage production**

| Cocktail phages | Volume processed (phage titre recovery efficiency %) | | | | | Product |
| | Stage 1: Phage lysate preparation | | Stage 2: Purification and Packaging | | | |
| | Raw lysate (i) | Bacteria Free lysate (ii) | Pure lysate (iii) | Cocktail (iv)* | Endotoxin free lysate (v) | Titre PFU/mL |
|---|---|---|---|---|---|---|
| øEnA02 | 1000 mL (NA) | 980 mL (100%) | 150 mL (76.5%) | 75 mL (NA) | 50 mL (39.3%) | $1.8 \times 10^8$ |
| øEnC07 | 1000 mL (NA) | 950 mL (100%) | 130 mL (52.1%) | 75 mL (NA) | 50 mL (49.0%) | $4.3 \times 10^8$ |
| øEnC15 | 1000 mL (NA) | 950 mL (100%) | 150 mL (24%) | 75 mL (NA) | 50 mL (30%) | $4.2 \times 10^8$ |
| øNando | 1000 mL (NA) | 950 mL (70%) | 160 mL (56%) | 75 mL (NA) | 50 mL (56.3%) | $1.2 \times 10^9$ |
| øTaquito | 1000 mL (NA) | 950 mL (100%) | 140 mL (32.2%) | 75 mL (NA) | 50 mL (66%) | $2.5 \times 10^8$ |

NA= not applicable * = mixture of 15 mL pure lysate from each component.

**Extended Data Table 2 | Genome and prophage analysis of *Entelli-02***

| Total # of reads | # Matched with cocktail phages | # Matched with host bacterial genome | # Matched with predicted prophage region |
|---|---|---|---|
| 14,623,072 | 14,508,309 (99.21%) | 39,992 (0.27%) | 12,996 (0.08%) |

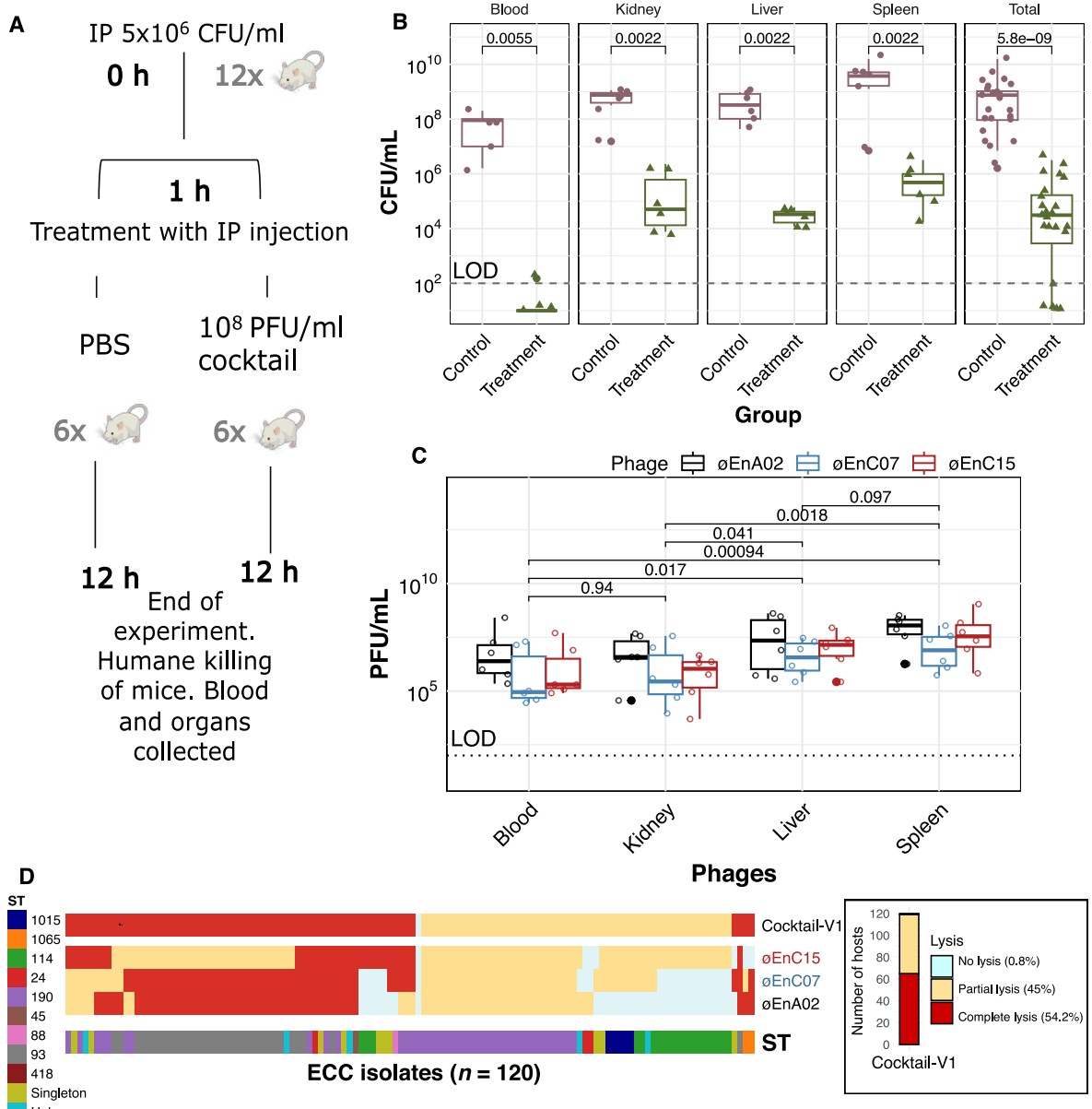

**Extended Data Fig. 1 | Efficacy of the phage cocktail-V1 against mouse infections and its host coverage across a range of clinical isolates.**
(**A**) Experimental scheme for the animal experiment. Grey numbers represent number of mice, black numbers represent hours post-infection. The illustration was created using Inkscape. (**B**) The reduction of bacterial load in vital tissues of control mice compared to the treatment group. Each data point represents an individual mouse ($n = 6$). Box plots show median (centre line), interquartile range (box edges: 25th-75th percentiles), whiskers extending to the most extreme values within 1.5× interquartile range, and outliers as individual points beyond whiskers. Total represents data from all samples ($n = 24$) combined. Statistical analysis was performed using the Mann-Whitney U test (two-sided) with exact

$p$-values to compare medians of bacterial count between groups. (**C**) The propagation of three different phages in various tissues of treated mice. The data ($n = 6$) are shown in box plots as above. Statistical analysis was performed using the Mann-Whitney U test as above. (**D**) Host range map of cocktail-V1 (øEnA02, øEnC07, and øEnC15) and its individual components tested against 120 *Enterobacter cloacae* complex (ECC) isolates from The Alfred Hospital. Each column represents an ECC isolate classified based on their Sequence Type (ST) and is color-coded accordingly. Red boxes indicate complete phage lysis of the respective ECC isolate, yellow boxes represent partial lysis, and light blue boxes indicate no observed lysis.

**A.** øEnC07 and adaptation host AH17D011

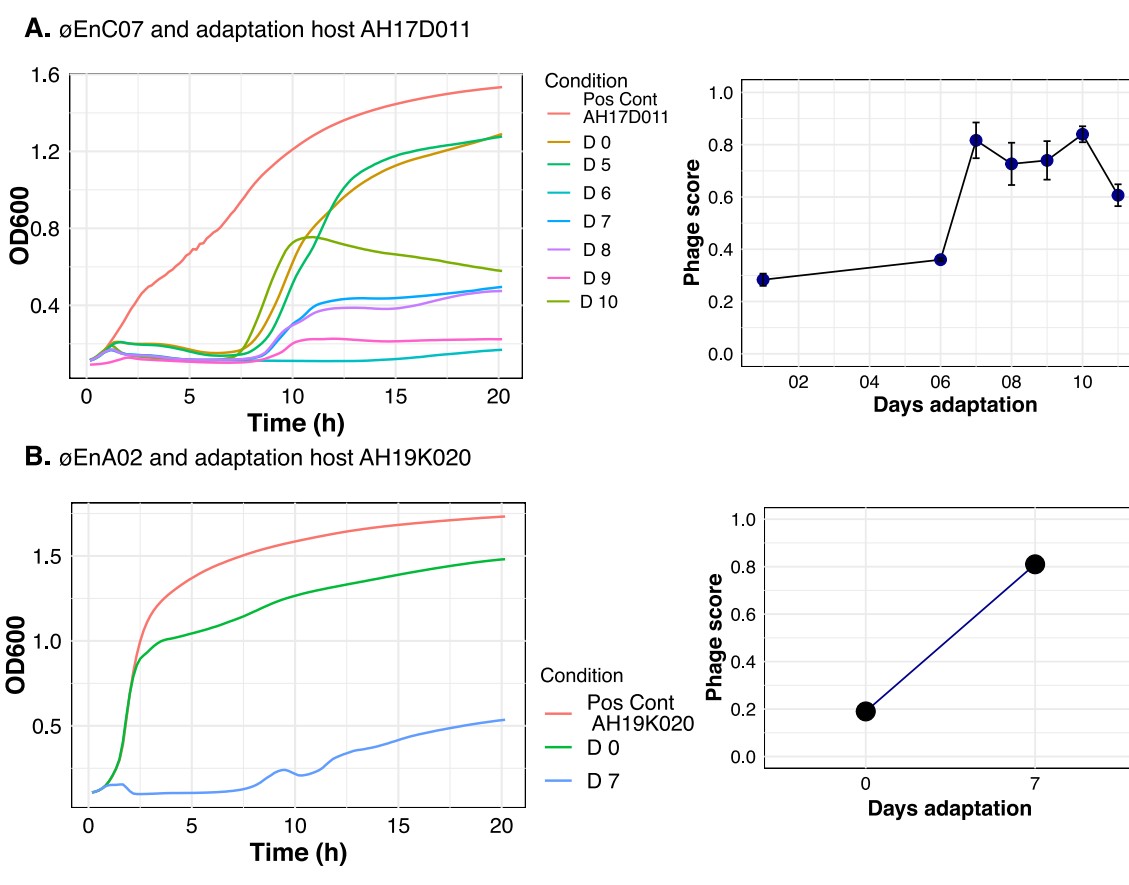

**B.** øEnA02 and adaptation host AH19K020

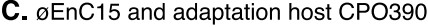

**C.** øEnC15 and adaptation host CPO390

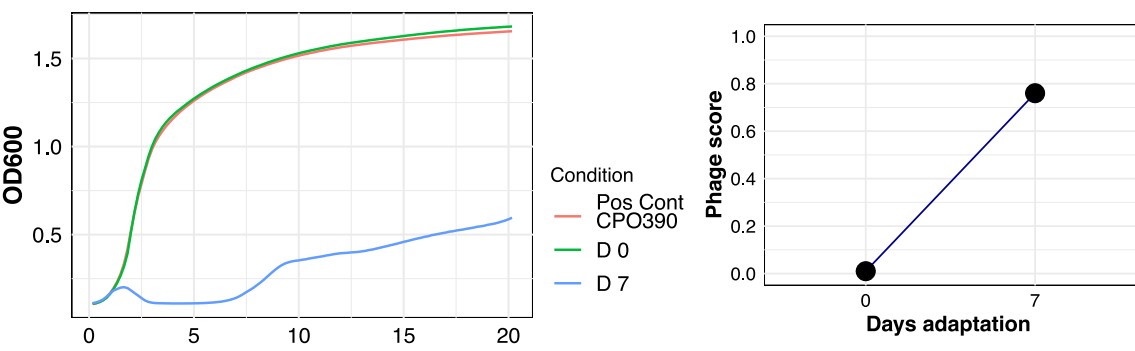

**Extended Data Fig. 2 | Growth kinetics assay (left) and the corresponding phage score plot (right).** The data presented in the figure reflects the performance of the adapted phage in days (D1-D10). D0 represents unadapted (ancestor phage) and Pos Cont and the host names indicate conditions where bacterial growth was observed without the phage treatment. (**A**) øEnC07 and host AH17D011 (pilot study) and corresponding phage score ($n = 3$); error bars represent standard error of mean. (**B**) øEnA02 and host AH19K022 and corresponding phage score ($n = 1$). (**C**) øEnC15 and host CPO390 and corresponding phage score ($n = 1$).

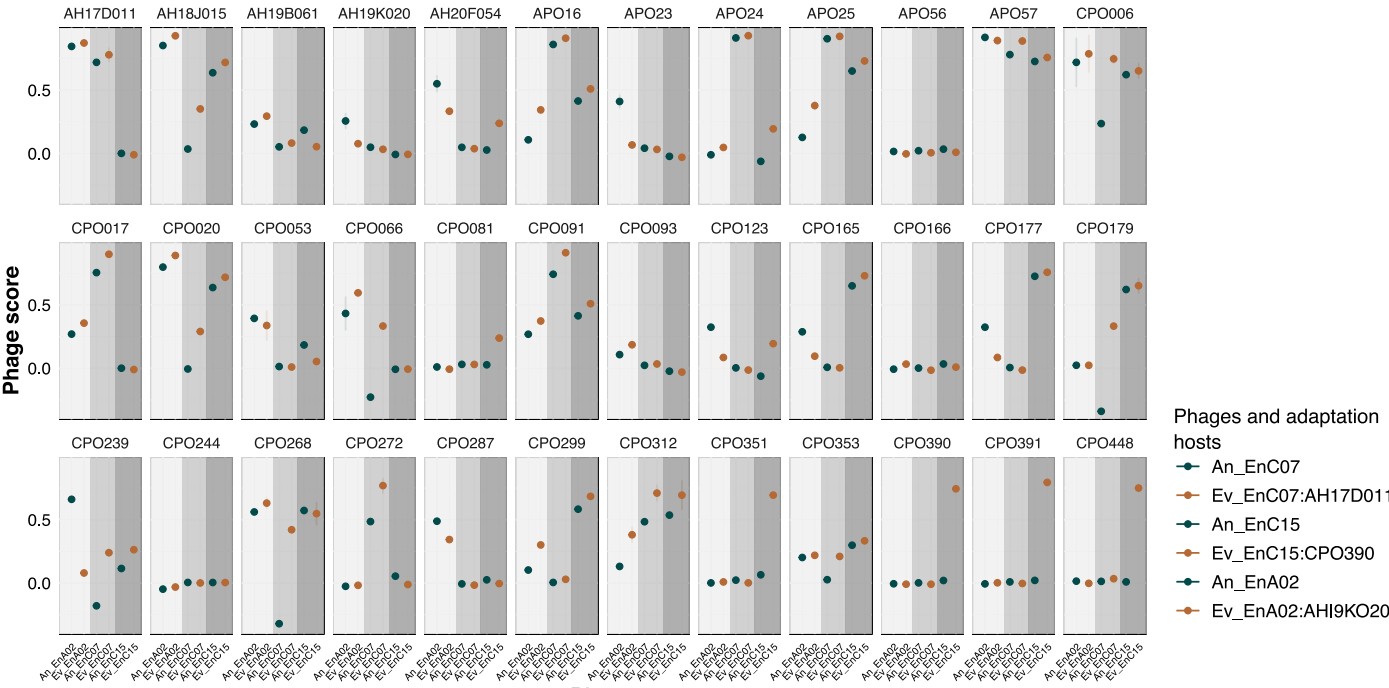

**Extended Data Fig. 3 | Phage scores comparing ancestral (An) and evolved phages (Ev) against *Enterobacter cloacae* complex isolates (*n* = 36).** Each dot represents the mean phage score from four technical replicates; error bars represent standard error of mean. Higher scores indicate greater phage susceptibility. The legend shows the specific phage-host adaptation pairs tested: ancestral phages An_EnC07, An_EnC15, and An_EnA02, and their corresponding evolved derivatives Ev_EnC07:AH17D011, Ev_EnC15:CPO390, and Ev_EnA02:AH19KO20.

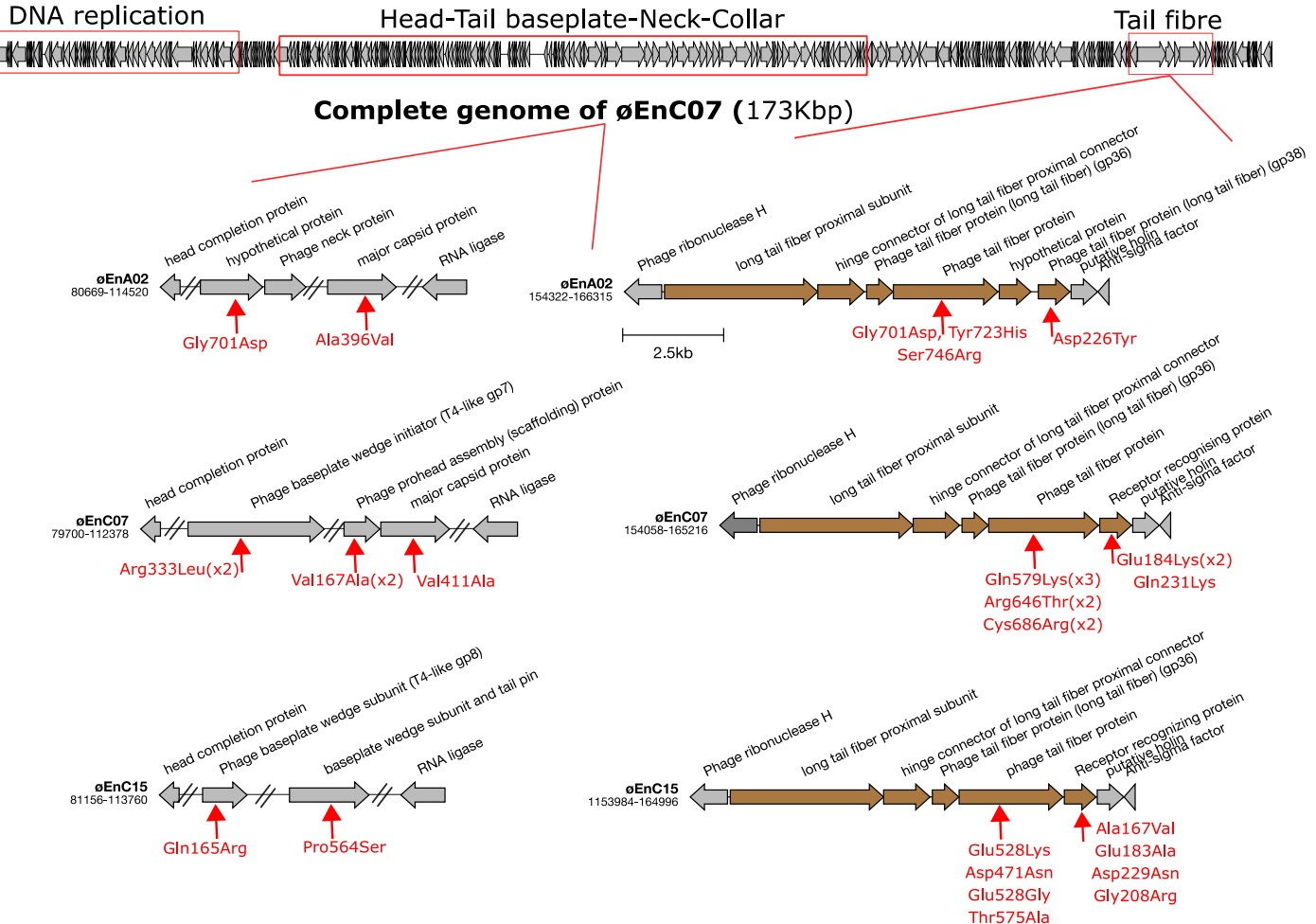

**Extended Data Fig. 4 | Single nucleotide polymorphisms (SNPs) observed within various genotypes of evolved (adapted) phages.** The genome map of øEnC07 is displayed as a reference genome to indicate complete genome, with distinct phage-specific gene regions indicated (DNA replication, head-tail baseplate-neck-collar, and tail fiber regions). Red arrows mark positions of SNPs) identified in evolved phage derivatives øEnA02, øEnC07, and øEnC15 following host adaptation. Each SNP is labelled with the amino acid substitution and genomic position. The brown-shaded regions highlight tail fiber genes where the majority of adaptive mutations occurred. Scale bar represents 2.5 kb.

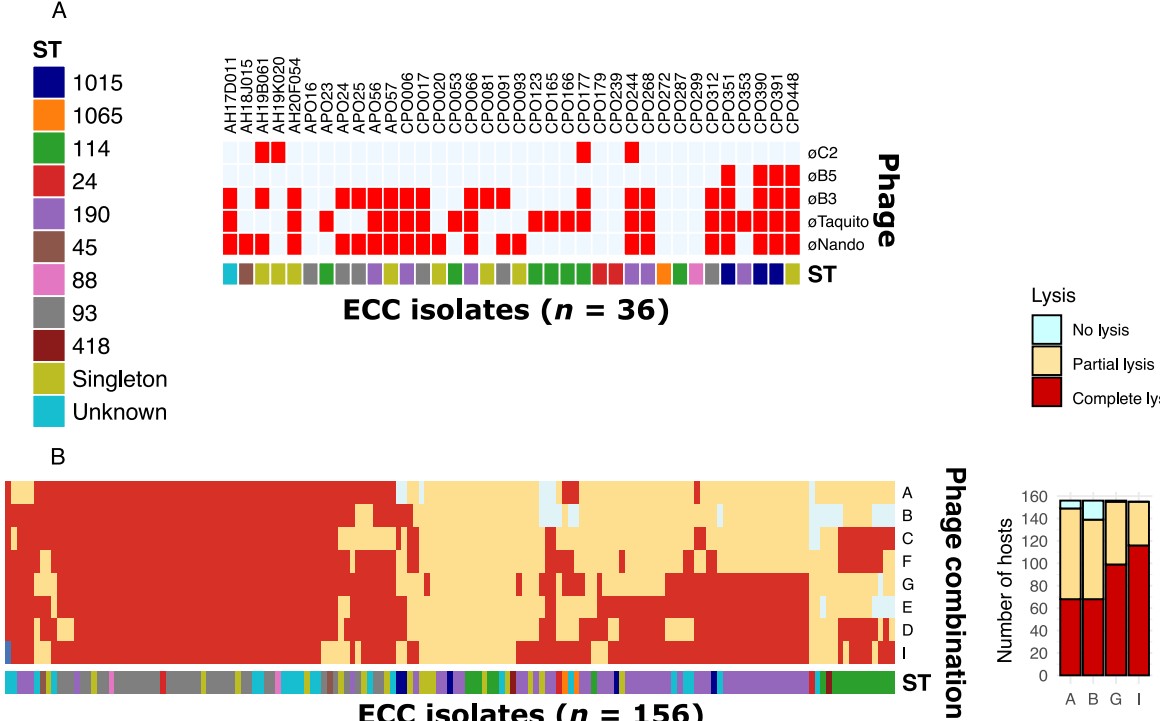

**Extended Data Fig. 5 | Host range analysis of individual phages and phage cocktails against *Enterobacter cloacae* complex isolates.** (**A**) Host range map of five newly isolated phages tested against 36 *Enterobacter cloacae* complex (ECC) isolates. Each row represents an ECC isolate, classified based on Sequence Type (ST), and is colour-coded accordingly. Singletons are STs that only occurred once. (**B**) Host range map of eight different phage combinations tested against a larger panel of 156 ECC isolates. Each row represents a phage cocktail (**A**-I). The bar chart quantifies the number of hosts lysed by selected phage combinations. The ev_ prefix denotes evolved/host-adapted phage variants. Combinations are ordered by their lytic activities. [Phage cocktail compositions: A = øEnA02 + øEnC07 + øEnC15, B = ev_øEnA02 + ev_øEnC07 + ev_øEnC15, C = øEnA02 + øEnC07 + øEnC15 + øTaquito, D = øEnA02 + øEnC07 + øEnC15 + øNando, E = ev_øEnA02 + ev_øEnC07 + ev_øEnC15 + øTaquito, F = ev_øEnA02 + ev_øEnC07 + ev_øEnC15 + øNando, G = ev_øEnA02 + ev_øEnC07 + ev_øEnC15 + øTaquito + øNando, I = øEnA02 + øEnC07 + øEnC15 + øTaquito + øNando].

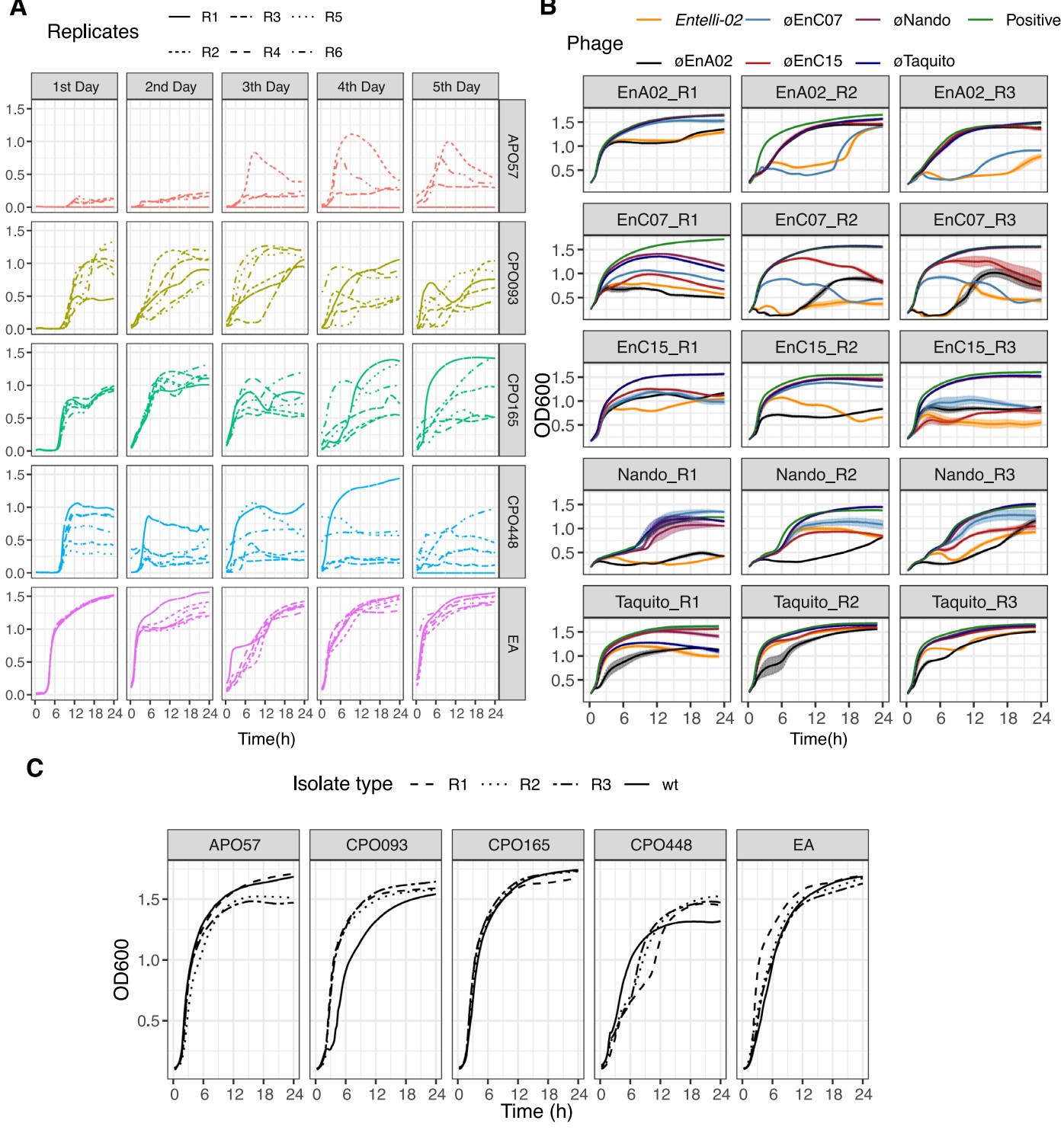

**Extended Data Fig. 6 | Growth kinetics of phage-resistant mutants.** (**A**) Serial passage evolution of each bacterial isolate (APO57, CPO093, CPO165, CPO448, EA) over five days, showing growth dynamics across six biological replicates (R1-R6). (**B**) Growth curves of phage-resistant mutants when challenged with individual phage components and *Entelli-02* full cocktail. Each panel shows independent resistant mutants (R1, R2, R3). Positive control denotes bacteria only growth. (**C**) Comparative growth kinetics of phage-resistant mutants (R1, R2, R3) relative to their corresponding wild-type (WT) parental strains (solid black lines) for five bacterial isolates. Growth is measured as optical density at 600 nm (OD600) over 24 h.

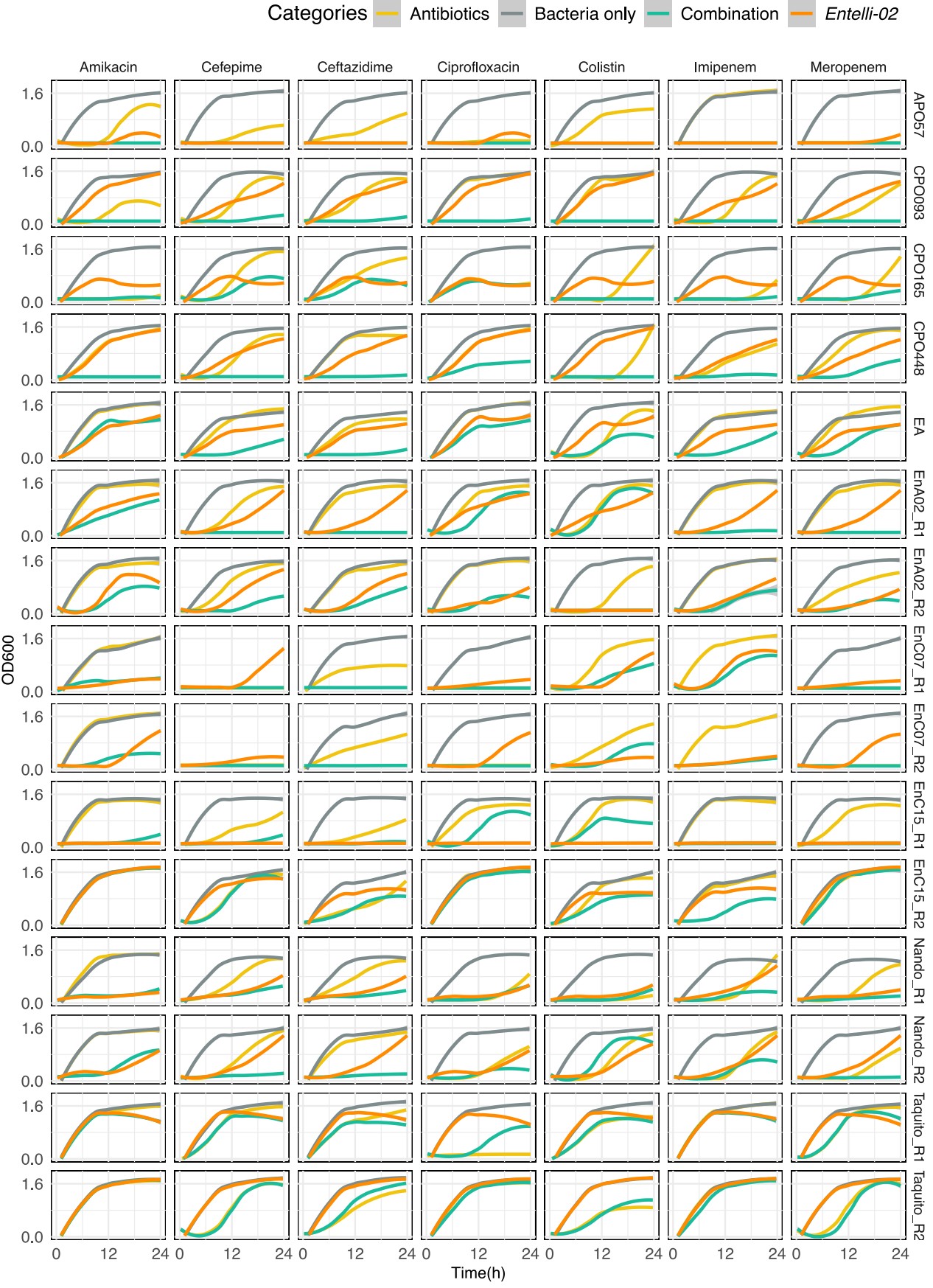

**Extended Data Fig. 7 | See next page for caption.**

**Extended Data Fig. 7 | Growth curves showing the interaction between** *Entelli-02* **phage cocktail and seven different antibiotics across wild type isolates and phage-resistant mutants.** Each panel represents a different isolate-treatment combination, with four treatment categories: antibiotics alone (yellow), bacteria only control (grey), *Entelli-02* -antibiotic combination (teal), and *Entelli-02* (orange). Bacterial strains tested include wild-type isolates (APO57, CPO093, CPO165, CPO448, EA) and their corresponding phage-resistant mutants (two clones, R1 and R2) derived from each component *Entelli-02* phages. Growth is measured as optical density at 600 nm (OD600) over 24 h.

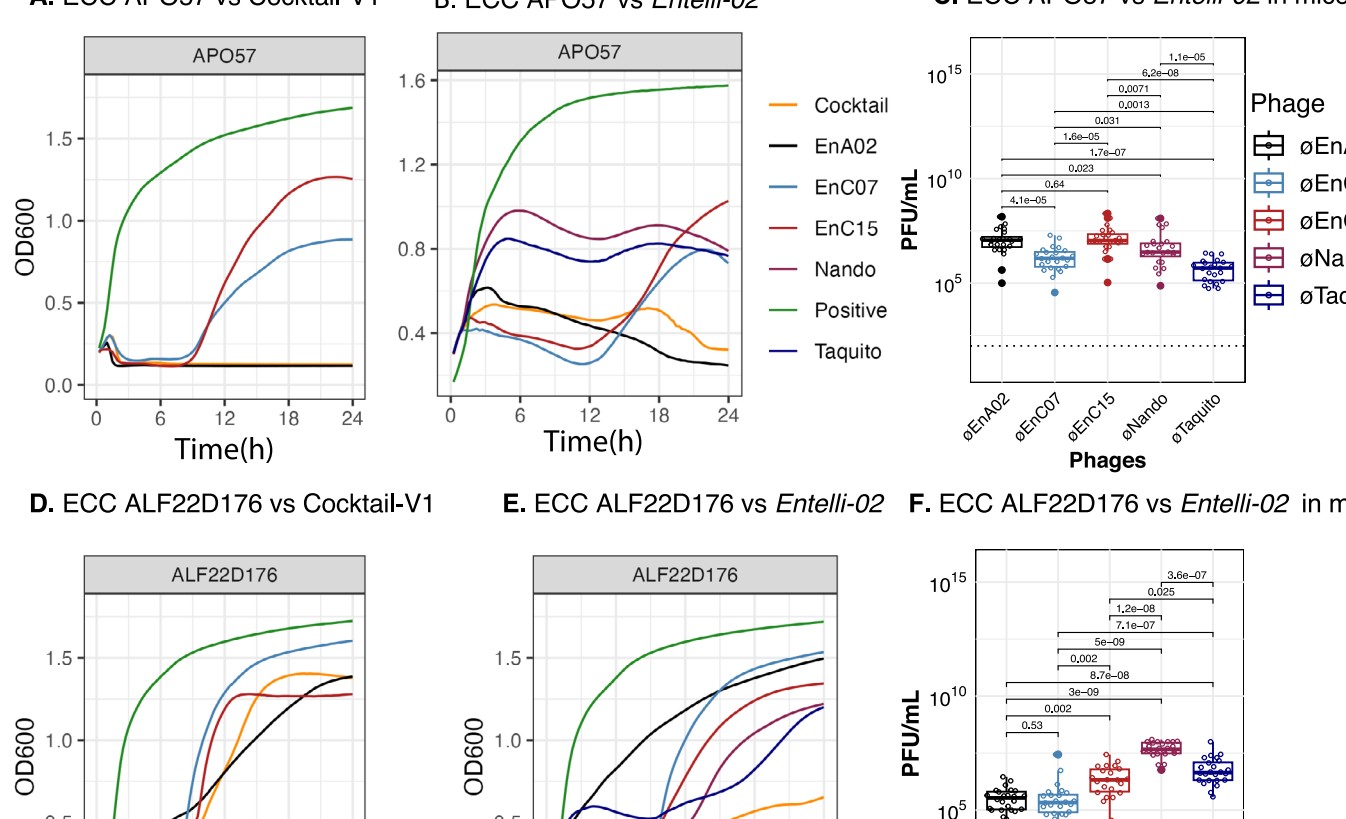

**Extended Data Fig. 8 | Growth inhibition and *in-vivo* efficacy of phages against two ECC isolates (A-F). (A)** Growth kinetics (OD600) of ECC isolate APO57 over 24 h following treatment with cocktail-V1 and its phage components. Positive control represents bacterial growth without phage treatment. **(B)** Growth kinetics (OD600) of APO57 treated with *Entelli-02* and its phage components. **(C)** *In-vivo* phage titres of different phages recovered from tissues of mice infected with APO57 and treated with *Entelli-02*. Data points represent individual mice (*n* = 6/group × 4 organs or tissue). Box plots show median (centre line), interquartile range (box edges: 25th-75th percentiles), whiskers extending to the most extreme values within 1.5× interquartile range, and outliers as individual points beyond whiskers. Statistical analysis was performed using the Mann-Whitney U test (two-sided) with exact *p*-values to compare medians of phage count between groups. **(D)** Growth kinetics (OD600) of ECC isolate ALF22D176 challenged with cocktail-V1 and its phage components. **(E)** Growth kinetics (OD600) of ECC isolate ALF22D176 challenged with *Entelli-02* and its phage components. **(F)** *In-vivo* phage titres recovered from tissues of mice infected with ALF22D176. Box plots show phage titres in four organs/tissues, with medians, interquartile ranges, and outliers indicated as above. Data points represent individual mice (*n* = 6/group × 4 organs or tissue). Statistical analysis was performed as above. Dotted lines in panels **C** and **F** represent detection limits.

# Reporting Summary

## Statistics

For all statistical analyses, confirm that the following items are present in the figure legend, table legend, main text, or Methods section.

| n/a | Confirmed | |
|---|---|---|
| ☐ | ☒ | The exact sample size (*n*) for each experimental group/condition, given as a discrete number and unit of measurement |
| ☐ | ☒ | A statement on whether measurements were taken from distinct samples or whether the same sample was measured repeatedly |
| ☐ | ☒ | The statistical test(s) used AND whether they are one- or two-sided<br>*Only common tests should be described solely by name; describe more complex techniques in the Methods section.* |
| ☐ | ☒ | A description of all covariates tested |
| ☐ | ☒ | A description of any assumptions or corrections, such as tests of normality and adjustment for multiple comparisons |
| ☐ | ☒ | A full description of the statistical parameters including central tendency (e.g. means) or other basic estimates (e.g. regression coefficient) AND variation (e.g. standard deviation) or associated estimates of uncertainty (e.g. confidence intervals) |
| ☐ | ☒ | For null hypothesis testing, the test statistic (e.g. *F*, *t*, *r*) with confidence intervals, effect sizes, degrees of freedom and *P* value noted<br>*Give P values as exact values whenever suitable.* |
| ☒ | ☐ | For Bayesian analysis, information on the choice of priors and Markov chain Monte Carlo settings |
| ☒ | ☐ | For hierarchical and complex designs, identification of the appropriate level for tests and full reporting of outcomes |
| ☒ | ☐ | Estimates of effect sizes (e.g. Cohen's *d*, Pearson's *r*), indicating how they were calculated |

*Our web collection on statistics for biologists contains articles on many of the points above.*

## Software and code

Policy information about availability of computer code

| Data collection | The data for this study was collected using lab experiments. For the comparison and analysis of the bacteriophage genome, we utilised the NCBI genome database and BLAST search. Information on bacteriophages classification was gathered from the International Committee on Taxonomy of Viruses. No special software was used to collect the data. |
|---|---|
| Data analysis | GraphPad Prism 10 and R 4.2.3 with RStudio 2024.04.1 using recent versions of publicly available packages. |

For manuscripts utilizing custom algorithms or software that are central to the research but not yet described in published literature, software must be made available to editors and reviewers. We strongly encourage code deposition in a community repository (e.g. GitHub). See the Nature Portfolio guidelines for submitting code & software for further information.

## Data

Policy information about availability of data

All manuscripts must include a data availability statement. This statement should provide the following information, where applicable:
- Accession codes, unique identifiers, or web links for publicly available datasets
- A description of any restrictions on data availability
- For clinical datasets or third party data, please ensure that the statement adheres to our policy

All raw data, supplementary materials, and analysis data generated during this study are publicly available at https://github.com/ECCphage. Additional supporting

data, figures, and tables are provided within the Article and its Extended Data and Supplementary Information. Genome sequences have been deposited in the NCBI database under BioProject accession number PRJNA629076. All data are freely accessible for research purposes.

# Research involving human participants, their data, or biological material

Policy information about studies with human participants or human data. See also policy information about sex, gender (identity/presentation), and sexual orientation and race, ethnicity and racism.

| | |
|---|---|
| Reporting on sex and gender | This information has not been collected. Not applicable for this study. |
| Reporting on race, ethnicity, or other socially relevant groupings | This information has not been collected. Not applicable for this study. |
| Population characteristics | This information has not been collected. Not applicable for this study. |
| Recruitment | This information has not been collected. Not applicable for this study. |
| Ethics oversight | This information has not been collected. Not applicable for this study. |

Note that full information on the approval of the study protocol must also be provided in the manuscript.

# Field-specific reporting

Please select the one below that is the best fit for your research. If you are not sure, read the appropriate sections before making your selection.

☒ Life sciences  ☐ Behavioural & social sciences  ☐ Ecological, evolutionary & environmental sciences

For a reference copy of the document with all sections, see nature.com/documents/nr-reporting-summary-flat.pdf

# Life sciences study design

All studies must disclose on these points even when the disclosure is negative.

| | |
|---|---|
| Sample size | No proper sample size was calculated for the bacterial isolates. We used the entire population of isolates available at the study site (The Alfred Hospital in Melbourne, Australia). A subset of these isolates, selected based on their genetic diversity, was used in various experiments as detailed in the manuscript. For the animal study, a sample size of at least six animals per group (treatment and control) was calculated to achieve a power of 90% with a significance level of 0.05. |
| Data exclusions | No data were excluded from the study. |
| Replication | The experiments were performed in replicates on at least three different occasions, all attempts at replication were successful, and all data are presented with error bars or whiskers in box plots where applicable. |
| Randomization | Animals were randomly allocated into control and treatment groups. |
| Blinding | Blinding was not relevant to the animal study as bacterial and phage counts were objectively measured after each experiment. Therefore outcomes were not influenced by human judgment. For qualitative assessments of the spot assay by phages, evaluations were conducted by at least two blinded researchers. |

# Reporting for specific materials, systems and methods

We require information from authors about some types of materials, experimental systems and methods used in many studies. Here, indicate whether each material, system or method listed is relevant to your study. If you are not sure if a list item applies to your research, read the appropriate section before selecting a response.

## Materials & experimental systems

| n/a | Involved in the study |
|---|---|
| ☒ | ☐ Antibodies |
| ☒ | ☐ Eukaryotic cell lines |
| ☒ | ☐ Palaeontology and archaeology |
| ☐ | ☒ Animals and other organisms |
| ☒ | ☐ Clinical data |
| ☒ | ☐ Dual use research of concern |
| ☒ | ☐ Plants |

## Methods

| n/a | Involved in the study |
|---|---|
| ☒ | ☐ ChIP-seq |
| ☒ | ☐ Flow cytometry |
| ☒ | ☐ MRI-based neuroimaging |

# Animals and other research organisms

Policy information about studies involving animals; ARRIVE guidelines recommended for reporting animal research, and Sex and Gender in Research

| | |
|---|---|
| Laboratory animals | BALB/c mice |
| Wild animals | The study did not involve wild animal. |
| Reporting on sex | The choice to use female mice in septicemia models was based on our previous study, which showed that they provide consistent data, are easier to handle, and exhibit reduced aggression. |
| Field-collected samples | This Study didn't involve field collected samples. |
| Ethics oversight | All protocols involving animals were reviewed and approved by the Monash University Animal Ethics Committee (Project ID: 2022-27681-80733). The animals were housed at the Monash Animal Research Facility, Monash University. |

Note that full information on the approval of the study protocol must also be provided in the manuscript.

# Plants

| | |
|---|---|
| Seed stocks | Not applicable |
| Novel plant genotypes | Not applicable |
| Authentication | Not applicable |

