## [Peer Review File · Nature Microbiology]

Rational design of a hospital-specific phage cocktail to treat *Enterobacter cloacae* Complex infections

Corresponding Author: Professor Jeremy Barr

Version 0:

Reviewer comments:

Reviewer #1

(Remarks to the Author)

General comments

One of the purported benefits of phage cocktails relates to valency of the coverage. The authors documented very well the process by which they selected the phages for their "improved" cocktail (Entelli-02) which seemed to be driven more by seeking to broaden coverage for the ECC in their MRC collection. In the end, they achieved lytic coverage of ~92% of their expanded ECC collection with their cocktail. How many of the ECC in the collection were covered by more than one phage, more than two phages, etc.? Would it be possible to construct a histogram that demonstrates the depth of the coverage?

Specific comments

Line

69 Reference 16 has recently advanced from a preprint to:

Pirnay JP, Djebara S, Steurs G, Griselain J, Cochez C, De Soir S, Glonti T, Spiessens A, Vanden Berghe E, Green S, Wagemans J, Lood C, Schrevens E, Chanishvili N, Kutateladze M, de Jode M, Ceysens PJ, Draye JP, Verbeken G, De Vos D, Rose T, Onsea J, Van Nieuwenhuysse B; Bacteriophage Therapy Providers; Bacteriophage Donors; Soentjens P, Lavigne R, Merabishvili M. Personalized bacteriophage therapy outcomes for 100 consecutive cases: a multicentre, multinational, retrospective observational study. *Nat Microbiol.* 2024 Jun;9(6):1434-1453. doi: 10.1038/s41564-024-01705-x. Epub 2024 Jun 4. PMID: 38834776; PMCID: PMC11153159.

464 Figure 6H (rather than 6G) is the panel that displays phage activity over time.

Reviewer #2

(Remarks to the Author)

In the present manuscript, Dinesh Subedi and colleagues describe the design of a phage cocktail, Entelli-02, containing five phages targeting *Enterobacter cloacae* complex (ECC). The five phages were chosen to target the majority of 156 ECC strains isolated at The Alfred Hospital in Melbourne (Australia). The main purpose of this "institution specific" cocktail, which would be available on-demand and could be used as a frontline treatment, is to bridge the divide between broad spectrum phage cocktails, which until today didn't show convincing efficiency in randomized clinical trials, and the seemingly more successful personalized phage therapy approaches.

Since ECC is an important nosocomial pathogen harboring a variety of antibiotic resistance genes, a phage cocktail that targets most ECC strains prevalent in a certain institute would indeed be a valuable addition to that institute's ant-bacterial arsenal.

The manuscript is well written and well organized and represents a substantial amount of work.

COMMENTS

1) Phage choice. The authors state that the choice of five phages with five different receptors reduces the chance of resistance. On closer inspection, all five phages use a receptor that is part of the lipopolysaccharide (LPS) of ECC. Therefore, it seems possible that the considerable selective pressure for LPS mutations could result in ECC LPS mutants that are resistant against

all five phages, especially when the same phage cocktail is used frequently in the same institute. For example, each of the observed mutations that cause resistance to phages PhiEnCO7 or PhiTaquito, and results in the loss a large part of LPS (the O-antigen, outer core and a part of the inner core), could result in resistance to all five phages of the Entelli-02 phage cocktail. Adding phages with completely different receptors, for example located in the cell wall of the bacteria, would be an improvement in my opinion. It should also be said that even when three completely different receptors were targeted, it has been shown (in *Pseudomonas aeruginosa*, admittedly) that strains can be selected in the patient, during treatment, that are resistant to all three phages and have mutations in all three receptors, at the same time.

Are there no *Enterobacter* phages that target non-LPS receptors, or were the five phages mainly chosen based on their lytic activity and host range and did all of them happen to have LPS as a receptor? Did the authors check for the in vitro selection of bacterial mutants that are resistant to all five phages?

2) Phage resistance. It has been proposed that when using natural phages in therapy, the selection of bacterial phage resistance is virtually inevitable, and thus the focus should be on selecting phages for which bacterial resistance would incur a fitness cost or re-sensitization to antibiotics (which would be applied concomitantly), and which would thus positively impact the clinical outcome. Do the authors have an idea of the fitness cost the bacterial phage resistance mutations would entail and of possible synergistic (or antagonistic) effects of their phages with standard of care anti-ECC antibiotics? Colistin, one of the last-resort anti-ECC antibiotics, for instance, is known to impair LPS production and could thus (theoretically) elicit an antagonistic effect with phages targeting LPS. Do the authors foresee the use of their phage cocktail in combination with antibiotics?

3) Mouse model. The mouse model used to demonstrate a certain in vivo therapeutic efficacy of the cocktail evaluates three phages that all efficiently target [efficiency of plating (EOP) = 1] the ECC isolate with which the mice are inoculated. This is an ideal setting, which will rarely occur in real-world applications of the phage cocktail. The mouse model is said to be an infection model, but the phages are administered one hour post-inoculation (the authors state one hour post-infection) of the animals with the ECC isolate. How certain can one be that the mice were actually infected with ECC prior to phage application, and not merely contaminated? For example, it could be that the phages prevented the contamination from developing into an infection in the phage-treated mice (prophylaxis), while the mice that did not receive phages developed a full-fledged infection. How sure are the authors that the phages actually resolved an infection? Why was the mouse model, which one would consider to be the final step in the evaluation process, not performed using the final 5-component cocktail? Why perform the animal study before an analysis that shows that the 3 initially selected phages do not have a sufficiently broad host range?

4) Therapeutic grade. The authors state that they produced a therapeutic grade Entelli-02 phage product. Does this mean that the Australian competent authorities agreed that the used production and quality control processes result in a phage product they will allow for generic therapeutic use? If the proposed Entelli-02 phage cocktail cannot be used ad hoc, without prior patient-based competent authority and/or ethical committee approval, and/or informed consent, then its advantage over personalized approaches might be largely lost.

5) Advantages. The authors state “this ‘institutional cocktail’ offers a streamlined alternative to the extensive processes typical of personalized phage therapies, which leverages advantages from both single-patient compassionate use and broad-range phage cocktails”. However, most personalized phage therapy cases are not performed in a compassionate use setting (compassionate use is a program offered by pharmaceutical companies to provide a medicine free of charge for indications that are not already included in a funded scheme. Many of them are performed instance under the umbrella of article 37 of the Declaration of Helsinki (unproven interventions in clinical practice), or in routine without prior competent authority or ethical committee approval, for any infection type, caused by antibiotic-resistant or antibiotic-susceptible bacteria alike (e.g., in Georgia). In personalized phage therapy, typically phages (adapted or not) are selected (based on a “phagogram”) from a collection of pre-produced and quality-controlled phages that has been set up to target the bacteria that are prevalent in the concerned hospital. When there is no time to select the active phages only, it is possible to select and mix compatible phages that are most likely to be active into a cocktail (an ad hoc institutional cocktail as it were). Assuming that a pre-prepared institutional cocktail consists of the phages that are supposed to be active in a certain setting and have been produced separately, and would need to be tested prior – or at least during application (to confirm its activity) – the differences between the ‘institutional cocktail’ and personalized approaches are minimal (the time of mixing phages). Differences in costs are also mainly related to the ad hoc mixing of phages, but on the other hand, individual phages are more stable than ready-to-use cocktails. As observed in previous studies, such as the PhagoBurn study, it seems that also in this case, stocks of individual phages are more stable (> 18 months – Fig. 1J and 5H) than ready to use phage cocktails (6 months with one of the phages showing a 1 log titer reduction – Fig. 6G).

In addition, pre-prepared cocktails will not prevent the need for developing individual phages to cater for the predicted 8% of ECC infected patients that cannot be helped by the Entelli-02 cocktail, and to allow for a regular update of the phage cocktail. Personalized approaches (only using a phage cocktail when necessary) will allow for a more sustainable use of phages, as in non-urgent cases (e.g., chronic infections) only active phages will be applied. In the absence of more information on possible bacterial phage resistance issues, I feel we should avoid as much as possible the systematic and empirical use of the same phage cocktails, thus minimizing the chances of making the same mistakes (the misuse of antibiotics).

6) Adaptation. The authors use the terms “training” and “evolution” when referring to the (pre-)adaptation of phages to their hosts. However, the term “training”, which is frequently used, refers to phenotypic changes only (e.g., one can train to become better at baseball), while the authors clearly show that their adaptation process results in the selection of genetic changes. I suggest to either use the term “adaptation” or to put the term training between brackets.

It is assumed that adapting phages to a bacterial isolate can lead to increased phage lytic activity, but also to delayed bacterial phage resistance evolution. In the present study, the authors seem to have focused on increasing the lytic activity of the phages and not on delaying the emergence of phage-resistant bacteria. Do the authors have information on the effect of their adaptation process on bacterial phage resistance?

The authors decided to standardize the phage adaptation process, determining an ‘optimum phage training duration’. However, the duration of a phage adaptation procedure depends on how much the phage needs to adapt to reach a desired activity level (the required genetic drift). The optimum duration of 7 days is valid for the considered phage/bacterial isolate combination, but

cannot be generalized. The start condition (prior to adaptation) is an EOP of 0.79, which is considered to be a moderate activity by the authors. However, an EOPs of 0.79 is usually considered to be sufficient for therapy and can even be observed in biological replicates (same phage and bacterial isolates) of phage/bacterium couples that can obtain an EOP of 1. In other words, in the presented situation the required adaptation process would be limited (e.g., from EOP 0.79 to EOP 1.0), and no adaptation would even be acceptable for therapeutic use.

Lines 291-292. It is more than intriguing that a phage, which initially was not capable of productive infection in the training host, adapts to propagate in that same host without additional incitation (e.g., without addition of other helper phages that do propagate in the training host). What process would be at the basis of this phenomenon? How sure are the authors that there was no phage infection at all prior to the adaptation process?

7) Phage lytic activity. The terms “productive infection”, “activity” and “hazy spots/lysis” need to be better defined and seem to be mixed up throughout the text and in figures. For instance, lines 260-262: “hazy spots were not confirmed for productive infection, but likely represent a combination of low EOP), phage resistance and bactericidal activity resulting from “killing from without”. A low EOP implies a productive infection, hazy spots due to phage resistance also imply a productive infection, followed by the emergence of colonies of phage resistant bacterial mutants, while “killing from without” does not imply a productive infection. It would have been more informative and clearer if all phages/bacterial isolate combinations would have been compared based on the same tests instead of a mix of spot test, double agar overlay (DAO) method (optional), and growth kinetics (optional). For instance, conclusions with regard to phage activity variations are sometimes solely based on DOA, sometimes on growth kinetics.

8) Phage production. The lysate was diluted up to tenfold in 1 x PBS. Did the PBS contain Magnesium and Calcium? Is the final product compatible with intravenous application?

Were animal component free ingredients (e.g., culture media) used?

Were used solvents (e.g., 1-Octanol) removed from and/or quantified in the final product?

MINOR COMMENTS

Line 664: “..., we purified the resistant bacterial colonies...”. Since phage-resistance need to be confirmed later, I suggest “..., we purified the supposedly resistant bacterial colonies...”.

The terms isolate and strain need to be better defined and are regularly mixed up throughout the manuscript.

Reviewer #3

(Remarks to the Author)

The manuscript “Rational Design of Frontline Institutional Phage Cocktail for the Treatment of Nosocomial Enterobacter cloacae Complex Infections” from D. Subedi, J. J. Barr and co-workers describes an approach to generate a phage-cocktail (Entelli-02) to be used against infections by MDR (MultiDrug-Resistant) Enterobacter cloacae complex (ECC). Based on an iterative method of phage training to increase host range, their strategy succeeds to develop a phage-cocktail of expanded host range within E. cloacae species.

The work developed by the authors is timely, as indeed current phage-treatment options based on highly-personalized treatments or rather general cocktails covers two extremes of all possibilities. A midway solution, which is stable for pharmaceutical handling and usage, while retaining good antibacterial activity against a given species rather than a strain is certainly of the highest translational value. The manuscript is clearly written and in general the figures are of good quality. However, I have a few major points that I believe should be clarified:

1. While I appreciate the strong translation aspects highlighted in the manuscript, I think the emphasis on biological insight and novelty are either absent or not addressed. The experiments and data generated (experimental evolution & sequencing (full genome?)) offer plenty of opportunity to investigate potential novel resistance mechanisms from the bacterial side, and improved infection traits on the phage side. However, the only aspects and mutations highlighted (or even commented) in the manuscript are bound to phage receptors and phage tail proteins (=adsorption). All other mutations (if any) in the genomes potentially also influencing the phenotypic traits of interest are not highlighted, or even more sticking, not mentioned, listed or acknowledged. For instance, no information is provided on how many mutations (if any) other than phage receptors each resistance each mutant had, which can potentially contribute to resistance. Perhaps the most prominent evidence that this could be worth looking is the adsorption experiment using complementation: in no example the complementation completely phenocopies the wild-type. This is a strong indicator that there might be some other mutations supporting the resistance phenotype, which in my view should be acknowledged (even if cannot be totally understood). Without further investigation of these, or from the trained phages, the manuscript brings little biological novelty to the field.

2. The approach deployed to assemble Entelli-02 is very interesting (I don't comment on novelty, non-expert), specially because it probes strain diversity to design a “large” host range viral cocktail within a species. However, the final results show that a cocktail containing the parental phages performs as good as, if not better than, the one containing the trained ones. In addition, there is no sufficient rationale provided for the selected isolate-phage pairs for training or resistance evolution. Same applies for the phages selected for the 8 combinations tested. How do the authors know that the selected isolate-phage pairs are indeed the ones yielding the best phages? The strains used for training will probably have a strong impact on selected phage-traits, so one possibility would be to actually investigate this more in detail in a systematic way (aka more strains). On one hand, it would increase the chance that better phages are indeed obtained (just by trying more). On the other hand, even if no better phages are found, such a systematic study would be very interesting in itself from the evolutionary perspective on which phage traits are selected for across several isolates, or which ones are more isolate specific. It would give a good angle for novel biological insight, I believe.

3. The concept that using more phages with “different” receptors improves efficacy can be very misleading. The fact is that LPS

is the receptor for several phages, and there is no guarantee that “any” LPS mutation would not confer resistance to all these phages (cross-resistance). The authors could either access if this is the case, or ease their claim that this is in fact the case. As it is, this is not tested in the manuscript.

4. Last, while the authors went through large efforts to show clinical importance of their initial cocktail (even mice models), the efficiency of Entelli-02 as clean and purified for pharmacological application is totally absent (or did I miss it? At least not shown in Fig 6). Given that the preparation changes (or might) the titer of each phage, I think it would be absolutely crucial to at least quantify its activity against the strain panel. Without this, the translational prospect of the manuscript is inconclusive.

Minor comments

- The usage of “considerable morbidity and mortality” in the abstract is somewhat vague. What does “considerable” mean? It raises the question on whether this pathogen is really that important. I suggest to either be more precise, or just avoid such vague statement that can open questions on relevance.
- Fig 1, panel I: no indication of how many samples (and variance) were used for EOP estimation.
- Fig 3 panel C: 6 datapoints do not necessarily need a box-plot. I would suggest to just represent directly the six data points.
- Very specific sentence for non-experts “Hazy spots (~45%) were not confirmed for productive infection via the soft overlay (top) agar assay, but likely represented a combination of low EOP, phage resistance and bactericidal activity resulting from killing from without”. What does it mean killing from without?
- Line 276: Can this be seen in Fig 11? it would be great to have an indication (Fig. 11).
- Line 280: this is a vague definition of “phage scores” for the main text. It would be good to have a more clear definition already here to make it easier to follow. At least to state what the phage score reflects. Supp Fig 6 helps only partially. Methods is good, but too late for the non-expert reader.
- Line 291: EOP of strain GPO390 not shown in Fig 1E.
- Fig 4F: Latency lines could also be coloured according to their phage lineage (evolved or ancestor) to facilitate visualization
- Line 347: “To address this, we selected ECC strains from STs that lacked sufficient phage coverage as hosts for targeted phage isolation.” Very vague – which STs? Are they part of the initial smaller set or of the entire collection?
- Line 397: How were the 8 combinations containing 3, 4 or 5 phages from the 8 possible phages selected? Theatrically, there are $56 + 70 + 56$ possibilities to combine 8 phages in groups of 3, 4 or 5...

Decision Letter:

24th September 2024

Dear Professor Barr,

Thank you for your patience while your manuscript "Rational Design of Frontline Institutional Phage Cocktail for the Treatment of Nosocomial Enterobacter cloacae Complex Infections" was under peer-review at Nature Microbiology. It has now been seen by 3 referees, whose expertise and comments you will find at the end of this email. Although they find your work of some potential interest, they have raised a number of concerns that will need to be addressed before we can consider publication of the work in Nature Microbiology.

In particular, the referees have some major concerns regarding the potential emergence of resistance which has been insufficiently tested, they have concerns regarding the in vivo experiments and lack of testing the final phage product (five phages). The referees also feel that the advantage of your approach versus other approaches (more personalized approaches) isn't clear enough. Editorially, we will not require you to provide additional biological insights as suggested by referee #3, however, we will need all other referee comments to be addressed in full, especially testing resistance and addressing concerns related to the mouse work (including testing the five-phage product). If you feel it will not be possible to address these concerns, I would be happy to consult with our editorial colleagues at Nature Communications to see if they can offer an alternative path towards publication.

However, should further experimental data allow you to address these criticisms, we would be happy to look at a revised manuscript.

Please include a data availability statement as a separate section after Methods but before references, under the heading "Data Availability". This section should inform readers about the availability of the data used to support the conclusions of your study. This information includes accession codes to public repositories (data banks for protein, DNA or RNA sequences, microarray, proteomics data etc...), references to source data published alongside the paper, unique identifiers such as URLs to data repository entries, or data set DOIs, and any other statement about data availability. At a minimum, you should include the following statement: "The data that support the findings of this study are available from the corresponding author upon request",

mentioning any restrictions on availability. If DOIs are provided, we also strongly encourage including these in the Reference list (authors, title, publisher (repository name), identifier, year). For more guidance on how to write this section please see: <http://www.nature.com/authors/policies/data/data-availability-statements-data-citations.pdf>

* If you have not done so already we suggest that you begin to revise your manuscript so that it conforms to our Article format instructions at <http://www.nature.com/nmicrobiol/info/final-submission>. Refer also to any guidelines provided in this letter.

When submitting the revised version of your manuscript, please pay close attention to our [href="https://www.nature.com/nature-portfolio/editorial-policies/image-integrity">Digital Image Integrity Guidelines. and to the following points below:](https://www.nature.com/nature-portfolio/editorial-policies/image-integrity)

Link Redacted

Note: This url links to your confidential homepage and associated information about manuscripts you may have submitted or be reviewing for us. If you wish to forward this e-mail to co-authors, please delete this link to your homepage first.

Nature Microbiology is committed to improving transparency in authorship. As part of our efforts in this direction, we are now requesting that all authors identified as 'corresponding author' on published papers create and link their Open Researcher and Contributor Identifier (ORCID) with their account on the Manuscript Tracking System (MTS), prior to acceptance. This applies to primary research papers only. ORCID helps the scientific community achieve unambiguous attribution of all scholarly contributions. You can create and link your ORCID from the home page of the MTS by clicking on 'Modify my Springer Nature account'. For more information please visit www.springernature.com/orcid.

If you wish to submit a suitably revised manuscript we would hope to receive it within 6 months. If you cannot send it within this time, please let us know. We will be happy to consider your revision, even if a similar study has been accepted for publication at Nature Microbiology or published elsewhere (up to a maximum of 6 months).

Yours sincerely,

Reviewer Expertise:

Referee #1: Phage therapy, clinical trials
Referee #2: Phage therapy, clinical trials
Referee #3: Drug-resistant Enterobacter, AMR

Reviewer Comments:

Reviewer #1 (Remarks to the Author):

General comments

One of the purported benefits of phage cocktails relates to valency of the coverage. The authors documented very well the process by which they selected the phages for their "improved" cocktail (Entelli-02) which seemed to be driven more by seeking to broaden coverage for the ECC in their MRC collection. In the end, they achieved lytic coverage of ~92% of their expanded ECC collection with their cocktail. How many of the ECC in the collection were covered by more than one phage, more than two phages, etc.? Would it be possible to construct a histogram that demonstrates the depth of the coverage?

Specific comments

Line

69 Reference 16 has recently advanced from a preprint to:

Pirnay JP, Djebara S, Steurs G, Griselain J, Cochez C, De Soir S, Glonti T, Spiessens A, Vanden Berghe E, Green S, Wagemans J, Lood C, Schrevels E, Chanishvili N, Kutateladze M, de Jode M, Ceysens PJ, Draye JP, Verbeken G, De Vos D, Rose T, Onsea J, Van Nieuwenhuysse B; Bacteriophage Therapy Providers; Bacteriophage Donors; Soentjens P, Lavigne R, Merabishvili M. Personalized bacteriophage therapy outcomes for 100 consecutive cases: a multicentre, multinational, retrospective observational study. *Nat Microbiol.* 2024 Jun;9(6):1434-1453. doi: 10.1038/s41564-024-01705-x. Epub 2024 Jun 4. PMID: 38834776; PMCID: PMC11153159.

464 Figure 6H (rather than 6G) is the panel that displays phage activity over time.

Reviewer #2 (Remarks to the Author):

In the present manuscript, Dinesh Subedi and colleagues describe the design of a phage cocktail, Entelli-02, containing five phages targeting *Enterobacter cloacae* complex (ECC). The five phages were chosen to target the majority of 156 ECC strains isolated at The Alfred Hospital in Melbourne (Australia). The main purpose of this "institution specific" cocktail, which would be available on-demand and could be used as a frontline treatment, is to bridge the divide between broad spectrum phage cocktails, which until today didn't show convincing efficiency in randomized clinical trials, and the seemingly more successful personalized phage therapy approaches.

Since ECC is an important nosocomial pathogen harboring a variety of antibiotic resistance genes, a phage cocktail that targets most ECC strains prevalent in a certain institute would indeed be a valuable addition to that institute's ant-bacterial arsenal.

The manuscript is well written and well organized and represents a substantial amount of work.

COMMENTS

1) Phage choice. The authors state that the choice of five phages with five different receptors reduces the chance of resistance. On closer inspection, all five phages use a receptor that is part of the lipopolysaccharide (LPS) of ECC. Therefore, it seems possible that the considerable selective pressure for LPS mutations could result in ECC LPS mutants that are resistant against all five phages, especially when the same phage cocktail is used frequently in the same institute. For example, each of the observed mutations that cause resistance to phages PhiEnCO7 or PhiTaquito, and results in the loss a large part of LPS (the O-antigen, outer core and a part of the inner core), could result in resistance to all five phages of the Entelli-02 phage cocktail. Adding phages with completely different receptors, for example located in the cell wall of the bacteria, would be an improvement in my opinion. It should also be said that even when three completely different receptors were targeted, it has been shown (in *Pseudomonas aeruginosa*, admittedly) that strains can be selected in the patient, during treatment, that are resistant to all three phages and have mutations in all three receptors, at the same time. Are there no *Enterobacter* phages that target non-LPS receptors, or were the five phages mainly chosen based on their lytic activity and host range and did all of them happen to have LPS as a receptor? Did the authors check for the in vitro selection of bacterial mutants that are resistant to all five phages?

2) Phage resistance. It has been proposed that when using natural phages in therapy, the selection of bacterial phage resistance is virtually inevitable, and thus the focus should be on selecting phages for which bacterial resistance would incur a fitness cost or re-sensitization to antibiotics (which would be applied concomitantly), and which would thus positively impact the clinical outcome. Do the authors have an idea of the fitness cost the bacterial phage resistance mutations would entail and of possible synergistic (or antagonistic) effects of their phages with standard of care anti-ECC antibiotics? Colistin, one of the last-resort anti-ECC antibiotics, for instance, is known to impair LPS production and could thus (theoretically) elicit an antagonistic effect with phages targeting LPS. Do the authors foresee the use of their phage cocktail in combination with antibiotics?

3) Mouse model. The mouse model used to demonstrate a certain in vivo therapeutic efficacy of the cocktail evaluates three phages that all efficiently target [efficiency of plating (EOP) = 1] the ECC isolate with which the mice are inoculated. This is an ideal setting, which will rarely occur in real-world applications of the phage cocktail. The mouse model is said to be an infection model, but the phages are administered one hour post-inoculation (the authors state one hour post-infection) of the animals with the ECC isolate. How certain can one be that the mice were actually infected with ECC prior to phage application, and not merely contaminated? For example, it could be that the phages prevented the contamination from developing into an infection in the phage-treated mice (prophylaxis), while the mice that did not receive phages developed a full-fledged infection. How sure are the authors that the phages actually resolved an infection? Why was the mouse model, which one would consider to be the final step in the evaluation process, not performed using the final 5-component cocktail? Why perform the animal study before an analysis that shows that the 3 initially selected phages do not have a sufficiently broad host range?

4) Therapeutic grade. The authors state that they produced a therapeutic grade Entelli-02 phage product. Does this mean that the Australian competent authorities agreed that the used production and quality control processes result in a phage product

they will allow for generic therapeutic use? If the proposed Entelli-02 phage cocktail cannot be used ad hoc, without prior patient-based competent authority and/or ethical committee approval, and/or informed consent, then its advantage over personalized approaches might be largely lost.

5) Advantages. The authors state “this ‘institutional cocktail’ offers a streamlined alternative to the extensive processes typical of personalized phage therapies, which leverages advantages from both single-patient compassionate use and broad-range phage cocktails”. However, most personalized phage therapy cases are not performed in a compassionate use setting (compassionate use is a program offered by pharmaceutical companies to provide a medicine free of charge for indications that are not already included in a funded scheme. Many of them are performed instance under the umbrella of article 37 of the Declaration of Helsinki (unproven interventions in clinical practice), or in routine without prior competent authority or ethical committee approval, for any infection type, caused by antibiotic-resistant or antibiotic-susceptible bacteria alike (e.g., in Georgia). In personalized phage therapy, typically phages (adapted or not) are selected (based on a “phagogram”) from a collection of pre-produced and quality-controlled phages that has been set up to target the bacteria that are prevalent in the concerned hospital. When there is no time to select the active phages only, it is possible to select and mix compatible phages that are most likely to be active into a cocktail (an ad hoc institutional cocktail as it were). Assuming that a pre-prepared institutional cocktail consists of the phages that are supposed to be active in a certain setting and have been produced separately, and would need to be tested prior – or at least during application (to confirm its activity) – the differences between the ‘institutional cocktail’ and personalized approaches are minimal (the time of mixing phages). Differences in costs are also mainly related to the ad hoc mixing of phages, but on the other hand, individual phages are more stable than ready-to-use cocktails. As observed in previous studies, such as the PhagoBurn study, it seems that also in this case, stocks of individual phages are more stable (> 18 months – Fig. 1J and 5H) than ready to use phage cocktails (6 months with one of the phages showing a 1 log titer reduction – Fig. 6G). In addition, pre-prepared cocktails will not prevent the need for developing individual phages to cater for the predicted 8% of ECC infected patients that cannot be helped by the Entelli-02 cocktail, and to allow for a regular update of the phage cocktail. Personalized approaches (only using a phage cocktail when necessary) will allow for a more sustainable use of phages, as in non-urgent cases (e.g., chronic infections) only active phages will be applied. In the absence of more information on possible bacterial phage resistance issues, I feel we should avoid as much as possible the systematic and empirical use of the same phage cocktails, thus minimizing the chances of making the same mistakes (the misuse of antibiotics).

6) Adaptation. The authors use the terms “training” and “evolution” when referring to the (pre-)adaptation of phages to their hosts. However, the term “training”, which is frequently used, refers to phenotypic changes only (e.g., one can train to become better at baseball), while the authors clearly show that their adaptation process results in the selection of genetic changes. I suggest to either use the term “adaptation” or to put the term training between brackets.

It is assumed that adapting phages to a bacterial isolate can lead to increased phage lytic activity, but also to delayed bacterial phage resistance evolution. In the present study, the authors seem to have focused on increasing the lytic activity of the phages and not on delaying the emergence of phage-resistant bacteria. Do the authors have information on the effect of their adaptation process on bacterial phage resistance?

The authors decided to standardize the phage adaptation process, determining an ‘optimum phage training duration’. However, the duration of a phage adaptation procedure depends on how much the phage needs to adapt to reach a desired activity level (the required genetic drift). The optimum duration of 7 days is valid for the considered phage/bacterial isolate combination, but cannot be generalized. The start condition (prior to adaptation) is an EOP of 0.79, which is considered to be a moderate activity by the authors. However, an EOPs of 0.79 is usually considered to be sufficient for therapy and can even be observed in biological replicates (same phage and bacterial isolates) of phage/bacterium couples that can obtain an EOP of 1. In other words, in the presented situation the required adaptation process would be limited (e.g., from EOP 0.79 to EOP 1.0), and no adaptation would even be acceptable for therapeutic use.

Lines 291-292. It is more than intriguing that a phage, which initially was not capable of productive infection in the training host, adapts to propagate in that same host without additional incitation (e.g., without addition of other helper phages that do propagate in the training host). What process would be at the basis of this phenomenon? How sure are the authors that there was no phage infection at all prior to the adaptation process?

7) Phage lytic activity. The terms “productive infection”, “activity” and “hazy spots/lysis” need to be better defined and seem to be mixed up throughout the text and in figures. For instance, lines 260-262: “hazy spots were not confirmed for productive infection, but likely represent a combination of low EOP), phage resistance and bactericidal activity resulting from “killing from without”. A low EOP implies a productive infection, hazy spots due to phage resistance also imply a productive infection, followed by the emergence of colonies of phage resistant bacterial mutants, while “killing from without” does not imply a productive infection. It would have been more informative and clearer if all phages/bacterial isolate combinations would have been compared based on the same tests instead of a mix of spot test, double agar overlay (DAO) method (optional), and growth kinetics (optional). For instance, conclusions with regard to phage activity variations are sometimes solely based on DOA, sometimes on growth kinetics.

8) Phage production. The lysate was diluted up to tenfold in 1 x PBS. Did the PBS contain Magnesium and Calcium? Is the final product compatible with intravenous application?

Were animal component free ingredients (e.g., culture media) used?

Were used solvents (e.g., 1-Octanol) removed from and/or quantified in the final product?

MINOR COMMENTS

Line 664: “..., we purified the resistant bacterial colonies...”. Since phage-resistance need to be confirmed later, I suggest “..., we purified the supposedly resistant bacterial colonies...”.

The terms isolate and strain need to be better defined and are regularly mixed up throughout the manuscript.

Reviewer #3 (Remarks to the Author):

The manuscript "Rational Design of Frontline Institutional Phage Cocktail for the Treatment of Nosocomial *Enterobacter cloacae* Complex Infections" from D. Subedi, J. J. Barr and co-workers describes an approach to generate a phage-cocktail (Entelli-02) to be used against infections by MDR (MultiDrug-Resistant) *Enterobacter cloacae* complex (ECC). Based on an iterative method of phage training to increase host range, their strategy succeeds to develop a phage-cocktail of expanded host range within *E. cloacae* species.

The work developed by the authors is timely, as indeed current phage-treatment options based on highly-personalized treatments or rather general cocktails covers two extremes of all possibilities. A midway solution, which is stable for pharmaceutical handling and usage, while retaining good antibacterial activity against a given species rather than a strain is certainly of the highest translational value. The manuscript is clearly written and in general the figures are of good quality. However, I have a few major points that I believe should be clarified:

1. While I appreciate the strong translation aspects highlighted in the manuscript, I think the emphasis on biological insight and novelty are either absent or not addressed. The experiments and data generated (experimental evolution & sequencing (full genome?)) offer plenty of opportunity to investigate potential novel resistance mechanisms from the bacterial side, and improved infection traits on the phage side. However, the only aspects and mutations highlighted (or even commented) in the manuscript are bound to phage receptors and phage tail proteins (=adsorption). All other mutations (if any) in the genomes potentially also influencing the phenotypic traits of interest are not highlighted, or even more sticking, not mentioned, listed or acknowledged. For instance, no information is provided on how many mutations (if any) other than phage receptors each resistance each mutant had, which can potentially contribute to resistance. Perhaps the most prominent evidence that this could be worth looking is the adsorption experiment using complementation: in no example the complementation completely phenocopies the wild-type. This is a strong indicator that there might be some other mutations supporting the resistance phenotype, which in my view should be acknowledged (even if cannot be totally understood). Without further investigation of these, or from the trained phages, the manuscript brings little biological novelty to the field.
2. The approach deployed to assemble Entelli-02 is very interesting (I don't comment on novelty, non-expert), specially because it probes strain diversity to design a "large" host range viral cocktail within a species. However, the final results show that a cocktail containing the parental phages performs as good as, if not better than, the one containing the trained ones. In addition, there is no sufficient rationale provided for the selected isolate-phage pairs for training or resistance evolution. Same applies for the phages selected for the 8 combinations tested. How do the authors know that the selected isolate-phage pairs are indeed the ones yielding the best phages? The strains used for training will probably have a strong impact on selected phage-traits, so one possibility would be to actually investigate this more in detail in a systematic way (aka more strains). On one hand, it would increase the chance that better phages are indeed obtained (just by trying more). On the other hand, even if no better phages are found, such a systematic study would be very interesting in itself from the evolutionary perspective on which phage traits are selected for across several isolates, or which ones are more isolate specific. It would give a good angle for novel biological insight, I believe.
3. The concept that using more phages with "different" receptors improves efficacy can be very misleading. The fact is that LPS is the receptor for several phages, and there is no guarantee that "any" LPS mutation would not confer resistance to all these phages (cross-resistance). The authors could either access if this is the case, or ease their claim that this is in fact the case. As it is, this is not tested in the manuscript.
4. Last, while the authors went through large efforts to show clinical importance of their initial cocktail (even mice models), the efficiency of Entelli-02 as clean and purified for pharmacological application is totally absent (or did I miss it? At least not shown in Fig 6). Given that the preparation changes (or might) the titer of each phage, I think it would be absolutely crucial to at least quantify its activity against the strain panel. Without this, the translational prospect of the manuscript is inconclusive.

Minor comments

- The usage of "considerable morbidity and mortality" in the abstract is somewhat vague. What does "considerable" mean? It raises the question on whether this pathogen is really that important. I suggest to either be more precise, or just avoid such vague statement that can open questions on relevance.
- Fig 1, panel I: no indication of how many samples (and variance) were used for EOP estimation.
- Fig 3 panel C: 6 datapoints do not necessarily need a box-plot. I would suggest to just represent directly the six data points.
- Very specific sentence for non-experts "Hazy spots (~45%) were not confirmed for productive infection via the soft overlay (top) agar assay, but likely represented a combination of low EOP, phage resistance and bactericidal activity resulting from killing from without". What does it mean killing from without?
- Line 276: Can this be seen in Fig 1I? it would be great to have an indication (Fig. 1I).
- Line 280: this is a vague definition of "phage scores" for the main text. It would be good to have a more clear definition already here to make it easier to follow. At least to state what the phage score reflects. Supp Fig 6 helps only partially. Methods is good, but too late for the non-expert reader.
- Line 291: EOP of strain CPO390 not shown in Fig 1E.
- Fig 4F: Latency lines could also be coloured according to their phage lineage (evolved or ancestor) to facilitate visualization
- Line 347: "To address this, we selected ECC strains from STs that lacked sufficient phage coverage as hosts for targeted phage isolation." Very vague – which STs? Are they part of the initial smaller set or of the entire collection?
- Line 397: How were the 8 combinations containing 3, 4 or 5 phages from the 8 possible phages selected? Theatrically, there are $56 + 70 + 56$ possibilities to combine 8 phages in groups of 3, 4 or 5...

Version 1:

Reviewer comments:

Reviewer #2

(Remarks to the Author)

I would like to thank the authors for the clarifications and additional experiments, and for the adaptations to the manuscript.

The authors clarified that phage selection was principally based on host coverage, resulting in a limited diversity (within the same bacterial structure - LPS) in phage binding sites.

Designing phage cocktails based mainly on the host range of the phages is an age-old practice. Today, however, considering the current access to phage genome sequencing, and the increasing knowledge regarding the interaction of phages with their bacterial hosts, I expected a more innovative approach to phage cocktail design.

I wonder, especially when considering the large amount of work that went into designing this institutional phage cocktail, why the authors did not decide to opt for a more rational approach, selecting phages that target different bacterial structures, including receptors for which phage binding is predicted (e.g., in in vitro and/or animal experiments) to exert selection for bacteria to evolve phage resistance, while impairing the effectiveness of these receptors, causing reduced pathogenesis (1) or increased sensitivity to certain antibiotic classes (2).

(1) Kortright KE et al. Selection for Phage Resistance Reduces Virulence of *Shigella flexneri*. *Appl Environ Microbiol*. 2022 Jan 25;88(2):e0151421.

(2) Chan BK et al. Phage selection restores antibiotic sensitivity in MDR *Pseudomonas aeruginosa*. *Sci Rep*. 2016 May 26;6:26717.

Regarding the “murine infection model”, the authors state that the demonstration of the presence of bacteria in tissues one hour post inoculation (not infection) is proof of infection. However, I was taught that the presence of bacteria in body tissues demonstrates colonization, while infection required the multiplication of the bacteria in these tissues, causing symptoms, disease, and immune response. I do, however, agree with the authors that initiating anti-bacterial treatment merely one hour after bacterial inoculation is a common practice in animal models and can be found in many publications.

Finally, authors added a summary of the broad benefits and limitations of personalized phage therapy versus broad-spectrum phage cocktails in both the introduction and discussion of the manuscript. They state “In contrast, personalised phage therapy involves the identification and use of a phage with demonstrated activity against a patient’s bacterial infection. This approach is often followed by adapting the phages (also known as “phage training”) to improve their effectiveness, followed by small-scale production and bespoke treatment”. This description refers to the most extreme form of personalized phage therapy. The selection of several phages that are active against the infecting bacteria (a personalized phage cocktail) from a collection of pre-produced and controlled (released) phages is a more common form of personalized phage therapy, which of course, if necessary, can be combined with the procedure described by the authors. The company Adaptive Phage Therapeutics (acquired by BiomX), for instance, developed and produced (under GMP) an extensive phage bank from which phages can be selected to precisely match the patients’ infections.

Reviewer #3

(Remarks to the Author)

I thank the authors for their efforts to answer and clarify all the questions. I also totally understand the challenge that would have been to experimentally address all requests, especially when it comes to simultaneously strengthen mechanistic insights and translational aspects. Therefore, I think the authors invested significant efforts to solidify their findings, namely with assessing (cross-)resistance, effect of antibiotic combinations and in vivo experiments with Entelli-02 – organized in a new section in the manuscript “Host range, phage resistance, and antibiotic synergy of Entelli-02”. In my opinion, this was a good and balanced choice of effort allocation because it improved the quality of the manuscript. I still have a few minor points/questions listed below.

Line 517: Reference to Fig. 7C at the end of the sentence is missing?

Line 518-20: Here it is unclear whether the resistant hosts were obtained from the experiment from 7D or whether resistance evolution was again conducted individually for each phage-host pair. I guess the second?

Figure 7D: I am unsure on how to clearly interpret this panel. As far as I understood, the color reflects phage-scores of infecting parent and resistant mutants with all phages, so this show allow to see resistance towards the phage the mutant was evolved with, but also cross-resistance towards all phages. I think it would be nice if the authors could somehow highlight the evolved host-phage pairs in the figure, since these are almost positive controls – lower phage scores in the resistant mutants are expected (even though APO57 did not seem to evolve that much resistance). As for the rest (cross-resistance) one does not really have an expectation, so all good. I think this would just guide the reader to digest this panel easier.

Line 524: It seems to me that when “one” (rather than “multiple”) phage exhibits strong lytic activity... which is actually good (better?). It somewhat “conflicts” with the next sentence, which is also true, but both are valid.

Figure 7E: What does the last row, separated from the antibiotics, mean?

Line 565: "our Entelli-02 product broadly limited the emergence of phage resistance": Is this actually shown? The statement is quite general (ambiguous). One could readily think that the cocktail slows down resistance development or really limits the emergence of some mutations in direct comparison to using the phages alone, but this was not really tested. I would suggest to rephrase to keep more loyal to the findings – which is that the observed collateral sensitivity patterns among evolved host-phage have the potential to slow down resistance to the cocktail.

Fig 7G (or strain AALF22D176) is not mentioned in the text at all. Is this intentional? Actually here Entelli-2 seems to work better than the initial cocktail, so it can be better after all.

Decision Letter:

Our ref: NMICROBIOL-24082377A

11th July 2025

Dear Jeremy,

Thank you for submitting your revised manuscript "Rational Design of Frontline Institutional Phage Cocktail for the Treatment of Nosocomial *Enterobacter cloacae* Complex Infections" (NMICROBIOL-24082377A). It has now been seen by two of the original referees and their comments are below. The reviewers find that the paper has improved in revision, and therefore we'll be happy in principle to publish it in *Nature Microbiology*, pending minor revisions to satisfy the referees' final requests and to comply with our editorial and formatting guidelines.

Thank you again for your interest in *Nature Microbiology*. Please do not hesitate to contact me if you have any questions.

Sincerely,

Reviewer #2 (Remarks to the Author):

I would like to thank the authors for the clarifications and additional experiments, and for the adaptations to the manuscript.

The authors clarified that phage selection was principally based on host coverage, resulting in a limited diversity (within the same bacterial structure - LPS) in phage binding sites.

Designing phage cocktails based mainly on the host range of the phages is an age-old practice. Today, however, considering the current access to phage genome sequencing, and the increasing knowledge regarding the interaction of phages with their bacterial hosts, I expected a more innovative approach to phage cocktail design.

I wonder, especially when considering the large amount of work that went into designing this institutional phage cocktail, why the authors did not decide to opt for a more rational approach, selecting phages that target different bacterial structures, including receptors for which phage binding is predicted (e.g., in *in vitro* and/or animal experiments) to exert selection for bacteria to evolve phage resistance, while impairing the effectiveness of these receptors, causing reduced pathogenesis (1) or increased sensitivity to certain antibiotic classes (2).

(1) Kortright KE et al. Selection for Phage Resistance Reduces Virulence of *Shigella flexneri*. *Appl Environ Microbiol*. 2022 Jan 25;88(2):e0151421.

(2) Chan BK et al. Phage selection restores antibiotic sensitivity in MDR *Pseudomonas aeruginosa*. *Sci Rep*. 2016 May 26;6:26717.

Regarding the "murine infection model", the authors state that the demonstration of the presence of bacteria in tissues one hour post inoculation (not infection) is proof of infection. However, I was taught that the presence of bacteria in body tissues demonstrates colonization, while infection required the multiplication of the bacteria in these tissues, causing symptoms, disease, and immune response. I do, however, agree with the authors that initiating anti-bacterial treatment merely one hour after bacterial inoculation is a common practice in animal models and can be found in many publications.

Finally, authors added a summary of the broad benefits and limitations of personalized phage therapy versus broad-spectrum phage cocktails in both the introduction and discussion of the manuscript. They state "In contrast, personalised phage therapy involves the identification and use of a phage with demonstrated activity against a patient's bacterial infection. This approach is

often followed by adapting the phages (also known as “phage training”) to improve their effectiveness, followed by small-scale production and bespoke treatment”. This description refers to the most extreme form of personalized phage therapy. The selection of several phages that are active against the infecting bacteria (a personalized phage cocktail) from a collection of pre-produced and controlled (released) phages is a more common form of personalized phage therapy, which of course, if necessary, can be combined with the procedure described by the authors. The company Adaptive Phage Therapeutics (acquired by BiomX), for instance, developed and produced (under GMP) an extensive phage bank from which phages can be selected to precisely match the patients’ infections.

Reviewer #3 (Remarks to the Author):

I thank the authors for their efforts to answer and clarify all the questions. I also totally understand the challenge that would have been to experimentally address all requests, especially when it comes to simultaneously strengthen mechanistic insights and translational aspects. Therefore, I think the authors invested significant efforts to solidify their findings, namely with assessing (cross-)resistance, effect of antibiotic combinations and in vivo experiments with Entelli-02 – organized in a new section in the manuscript “Host range, phage resistance, and antibiotic synergy of Entelli-02”. In my opinion, this was a good and balanced choice of effort allocation because it improved the quality of the manuscript. I still have a few minor points/questions listed below.

Line 517: Reference to Fig. 7C at the end of the sentence is missing?

Line 518-20: Here it is unclear whether the resistant hosts were obtained from the experiment from 7D or whether resistance evolution was again conducted individually for each phage-host pair. I guess the second?

Figure 7D: I am unsure on how to clearly interpret this panel. As far as I understood, the color reflects phage-scores of infecting parent and resistant mutants with all phages, so this should allow to see resistance towards the phage the mutant was evolved with, but also cross-resistance towards all phages. I think it would be nice if the authors could somehow highlight the evolved host-phage pairs in the figure, since these are almost positive controls – lower phage scores in the resistant mutants are expected (even though APO57 did not seem to evolve that much resistance). As for the rest (cross-resistance) one does not really have an expectation, so all good. I think this would just guide the reader to digest this panel easier.

Line 524: It seems to me that when “one” (rather than “multiple”) phage exhibits strong lytic activity... which is actually good (better?). It somewhat “conflicts” with the next sentence, which is also true, but both are valid.

Figure 7E: What does the last row, separated from the antibiotics, mean?

Line 565: “our Entelli-02 product broadly limited the emergence of phage resistance”: Is this actually shown? The statement is quite general (ambiguous). One could readily argue that the cocktail slows down resistance development or really limits the emergence of some mutations in direct comparison to using the phages alone, but this was not really tested. I would suggest to rephrase to keep more loyal to the findings – which is that the observed collateral sensitivity patterns among evolved host-phage have the potential to slow down resistance to the cocktail.

Fig 7G (or strain AALF22D176) is not mentioned in the text at all. Is this intentional? Actually here Entelli-2 seems to work better than the initial cocktail, so it can be better after all.

Version 2:

Decision Letter:

20th August 2025

Dear Jeremy,

I am pleased to accept your Article "Rational design of a hospital-specific phage cocktail to treat *Enterobacter cloacae* Complex infections" for publication in *Nature Microbiology*. Thank you for having chosen to submit your work to us and many congratulations.

Over the next few weeks, your paper will be copyedited to ensure that it conforms to *Nature Microbiology* style. We look particularly carefully at the titles of all papers to ensure that they are relatively brief and understandable.

You may wish to make your media relations office aware of your accepted publication, in case they consider it appropriate to organize some internal or external publicity. Once your paper has been scheduled you will receive an email confirming the

publication details. This is normally 3-4 working days in advance of publication. If you need additional notice of the date and time of publication, please let the production team know when you receive the proof of your article to ensure there is sufficient time to coordinate. Further information on our embargo policies can be found here:

<https://www.nature.com/authors/policies/embargo.html>

Authors may need to take specific actions to achieve compliance with funder and institutional open access mandates. If

your research is supported by a funder that requires immediate open access (e.g. according to [Plan S principles](https://www.springernature.com/gp/open-science/plan-s-compliance) or the [NIH public access policy](https://www.springernature.com/gp/open-science/us-federal-agency-compliance)) then you should select the gold OA route, and we will direct you to the compliant route where possible. Because authors warrant under our subscription licensing terms that they haven't committed to licensing any version of their article under a licence inconsistent with the terms of our agreement – including the applicable embargo period – publication under the subscription model isn't suitable for authors whose funders require no embargo.

Congratulations once again and I look forward to seeing the article published.

With kind regards,

P.S. Click on the following link if you would like to recommend Nature Microbiology to your librarian <http://www.nature.com/subscriptions/recommend.html#forms>

** Visit the Springer Nature Editorial and Publishing website at http://editorial-jobs.springernature.com?utm_source=ejP_NMicro_email&utm_medium=ejP_NMicro_email&utm_campaign=ejp_NMicro for more information about our career opportunities. If you have any questions please click [here](mailto:editorial.publishing.jobs@springernature.com).

Reviewer Comments Nature Microbiology Subedi et al., 2024

Please find below a line-by-line response to all reviewer comments with our responses shown in blue.

Reviewer 1

One of the purported benefits of phage cocktails relates to valency of the coverage. The authors documented very well the process by which they selected the phages for their “improved” cocktail (Entelli-02) which seemed to be driven more by seeking to broaden coverage for the ECC in their MRC collection.

The reviewer is correct in that our design of *Entelli-02* was principally driven by valency of host range coverage across our ECC collection.

In the end, they achieved lytic coverage of ~92% of their expanded ECC collection with their cocktail. How many of the ECC in the collection were covered by more than one phage, more than two phages, etc.? Would it be possible to construct a histogram that demonstrates the depth of the coverage?

To address this comment, we have included a comprehensive analysis of the host range of each of the five phages in *Entelli-02*, including the number of ECC isolates that were targeted by one or more phage in a histogram (Figure 7A), and a heat map showing the Efficiency of Plating (EOP) of each phage across the entire ECC collection (Figure 7B). Please see **lines 498-506** for text changes.

Line-69 Reference 16 has recently advanced from a preprint (Pirnay et al., Nat Microbiol. 2024 Jun;9(6):1434-1453)

Corrected.

Line-464 Figure 6H (rather than 6G) is the panel that displays phage activity over time.

Corrected, (please note that phage activity over time figure is in 6I in the revised version).

Reviewer 2

In the present manuscript, Dinesh Subedi and colleagues describe the design of a phage cocktail, Entelli-02, containing five phages targeting *Enterobacter cloacae* complex (ECC). The five phages were chosen to target the majority of 156 ECC strains isolated at The Alfred Hospital in Melbourne (Australia). The main purpose of this “institution specific” cocktail, which would be available on-demand and could be used as a frontline treatment, is to bridge the divide between broad spectrum phage cocktails, which until today didn’t show convincing efficiency in randomized clinical trials, and the seemingly more successful personalized phage therapy approaches.

Since ECC is an important nosocomial pathogen harboring a variety of antibiotic resistance genes, a phage cocktail that targets most ECC strains prevalent in a certain institute would indeed be a valuable addition to that institute’s ant-bacterial arsenal.

The manuscript is well written and well organized and represents a substantial amount of work. We thank the reviewer for their comments and the review of our work.

COMMENTS

1) Phage choice

The authors state that the choice of five phages with five different receptors reduces the chance of resistance. On closer inspection, all five phages use a receptor that is part of the lipopolysaccharide (LPS) of ECC.

We acknowledge that all five phages in our *Entelli-02* product broadly recognise the bacterial LPS layer, although we have shown greater granularity in the specific sub-components these phages target. As we described in our response to Reviewer 1, our phage selection was principally based on host range coverage, and we identified the receptors of each phage after this point. As such, our phage selection was not driven by diversity of receptors. We have expanded on this point in the main text (see lines 117-119).

Therefore, it seems possible that the considerable selective pressure for LPS mutations could result in ECC LPS mutants that are resistant against all five phages, especially when the same phage cocktail is used frequently in the same institute.

This is an important point. To address this, we completed additional experiments testing for the emergence of phage resistance to our *Entelli-02* cocktail (Figure 7C).

Here, we selected the five hosts of isolation for each respective phage in our cocktail. We then conducted a five-day *in vitro* evolution experiment with daily doses of *Entelli-02* and reported the killing efficiency of the cocktail against each host reported as a phage score. Broadly we saw limited emergence of phage resistance over the five days (Figure 7C). We do note that host Eareo, which was primarily infected by \$\phi\$ EnA02 and only showed low-level infectivity to other phages, showed the greatest reduction in infectivity (presumably driven by the emergence of phage resistance), whilst host APO57, which was susceptible to all five phages had minimal reduction in infectivity over the five days. Please see manuscript lines 507-517 for full text changes.

For example, each of the observed mutations that cause resistance to phages PhiEnCO7 or PhiTaquito, and results in the loss a large part of LPS (the O-antigen, outer core and a part of the inner core), could result in resistance to all five phages of the Entelli-02 phage cocktail.

To address this comment, we isolated three phage-resistant mutants for each phage-host pair (using the host of isolation), followed by screening for cross-infectivity against all phage and

Entelli-02 with infectivity presented as a phage score (Figure 7D). Broadly, we find that phage-resistant mutants were still susceptible to the *Entelli-02* cocktail albeit with varying degrees of efficiency.

For ϕ EnC07, we observed phage resistance emerge as a loss-of-function mutation in the inner LPS core in host APO57 (Figure 2 A&D). Surprisingly, we found high cross infectivity for *Entelli-02* against ϕ EnC07 phage resistant mutants, which was driven by ϕ EnA02 that has a secondary receptor as an outer membrane protein. For ϕ Taquito, similarly phage resistance emerged through loss-of-function mutations in the inner LPS core in host CPO165 (Figure 5E). Here we do find cross-resistance to most phages in *Entelli-02*, although there still remained low-to-moderate infectivity by ϕ EnA02. Please see manuscript lines 507-517 for full text changes.

Adding phages with completely different receptors, for example located in the cell wall of the bacteria, would be an improvement in my opinion. It should also be said that even when three completely different receptors were targeted, it has been shown (in *Pseudomonas aeruginosa*, admittedly) that strains can be selected in the patient, during treatment, that are resistant to all three phages and have mutations in all three receptors, at the same time.

We agree, however as stated above our selection of phages was driven by host range rather than receptor diversity.

Are there no Enterobacter phages that target non-LPS receptors, or were the five phages mainly chosen based on their lytic activity and host range and did all of them happen to have LPS as a receptor?

Please see our responses above for phage selection. Regarding non-LPS receptors, we did find that one of our phages (ϕ EnA02) recognised both the LPS and an outer membrane protein as its receptors (Figure 2B&C).

Did the authors check for the *in vitro* selection of bacterial mutants that are resistant to all five phages?

Please see our responses above for the *in vitro* selection of phage resistant mutants. We did not find evidence of isolates that were completely resistant to all five phages, although there was reduction in phage scores for select isolates (Figure 7D).

2) Phage resistance. It has been proposed that when using natural phages in therapy, the selection of bacterial phage resistance is virtually inevitable, and thus the focus should be on selecting phages for which bacterial resistance would incur a fitness cost or re-sensitization to antibiotics (which would be applied concomitantly), and which would thus positively impact the clinical outcome. Do the authors have an idea of the fitness cost the bacterial phage resistance mutations would entail and of possible synergistic (or antagonistic) effects of their phages with standard of care anti-ECC antibiotics? Colistin, one of the last-resort anti-ECC antibiotics, for instance, is known to impair LPS production and could thus (theoretically) elicit an antagonistic effect with phages targeting LPS. Do the authors foresee the use of their phage cocktail in combination with antibiotics?

To address this comment, we screened the previously described phage-resistant mutants for fitness costs through comparative growth curves compared with wild-type (Extended Data Fig 8C). We did not observe any significant growth defects under standard growth in LB media (see lines 562-564).

We further examined interactions between the wild-type and 10 phage-resistant mutants (taken from Fig 7D) with seven standard of care antibiotics, including Colistin (Fig 7E). Here, each antibiotic and was tested alone at 0.5× MIC, *Entelli-02* alone at an MOI of 0.1, and combinations of both for a total of 315 interactions (Extended Data Fig 9), which we used to identify positive, neutral, and negative phage-antibiotic interactions. Broadly we observe positive interactions between *Entelli-02* and most antibiotics with wild-type ECC isolates, which did trend towards neutral upon the emergence of phage resistance. As the reviewer suggested, we observed an increased number of antagonists interactions between phage-resistant mutants and Colistin, likely due to impaired LPS production. Please see manuscript **lines 537-560** for full text changes.

3) Mouse model. The mouse model used to demonstrate a certain in vivo therapeutic efficacy of the cocktail evaluates three phages that all efficiently target [efficiency of plating (EOP) ~ 1] the ECC isolate with which the mice are inoculated. This is an ideal setting, which will rarely occur in real-world applications of the phage cocktail.

We agree that this murine infection model and the isolate that was selected were an ideal combination. To address this, we conducted additional *in vivo* experiments (see details below) selecting a contemporary ECC clinical isolate from The Alfred Hospital's ongoing outbreak (Figure 7G; strain AALF22D176). Importantly, neither our phages or the *Entelli-02* cocktail had previously been exposed or tested against this isolate. As such, we believe this infection model represents a realistic pre-clinical challenge of our product.

The mouse model is said to be an infection model, but the phages are administered one hour post-inoculation (the authors state one hour post-infection) of the animals with the ECC isolate. How certain can one be that the mice were actually infected with ECC prior to phage application, and not merely contaminated? For example, it could be that the phages prevented the contamination from developing into an infection in the phage-treated mice (prophylaxis), while the mice that did not receive phages developed a full-fledged infection. How sure are the authors that the phages actually resolved an infection?

Infection followed by treatment at one-hour post infection is a widely utilised protocol, which was used previously by us (PMID: 35537278; PMID: 33432151) and across many studies in the scientific literature (PMID: 31586662; PMID: 22729924). For example, Smith et al. (PMID: 21116344) have demonstrated the colonisation of mouse tissues can occur as early as 20 minutes post IP infection. Similarly, *Leptospira*, which have a much slower replication rate compared to ECC (6 hours vs 40 minutes), are capable of colonising mouse tissues within an hour of infection (PMID: 35923802).

Since ECC is not a natural pathogen of mice, high inoculum loads are typically required to establish reproducible infections. This was consistent with our pilot studies where a minimum of 5×10^6 CFU/inoculum was required as a consistent minimal lethal dose (Extended Data Figure 3 B&C). These doses, while effective in reliably initiating infection, result in bacterial burdens in blood and tissues that far exceed typical human levels and can rapidly progress to overwhelming infection. Delaying antimicrobial treatment further could mask therapeutic efficacy due to the rapid and excessive proliferation of bacteria. Importantly, our study demonstrates the presence of bacterial colonisation in tissues of both phage-treated and untreated mice, indicating that phage administration occurred after infection was established, rather than merely preventing the progression of contamination into infection. To justify these, we have added some more information in the methods (see **lines 909-911**).

Why was the mouse model, which one would consider to be the final step in the evaluation process, not performed using the final 5-component cocktail? Why perform the animal study before an analysis that shows that the 3 initially selected phages do not have a sufficiently broad host range?

We acknowledge that the initial animal experiment was conducted using only cocktail-V1 on a bacterial isolate (APO57) with high Efficiency of Plating (EOP \approx 1). As phage cocktail development was an iterative process, this *in vivo* experiment served as a checkpoint to evaluate phage activity prior to further optimisation (see **lines 235-236**).

To address this, we repeated the *in vivo* experiment comparing the antimicrobial activity of both cocktail-V1 and *Entelli-02* with the same ECC isolate APO57 (Figure 7F). Additionally, we obtain a novel, contemporary ECC clinical isolate from The Alfred Hospital that had not been previously used in cocktail formulation or testing and compared the antimicrobial activity of our two cocktails (Figure 7G). Please see manuscript **lines 570-591** for full text changes.

4) Therapeutic grade. The authors state that they produced a therapeutic grade Entelli-02 phage product. Does this mean that the Australian competent authorities agreed that the used production and quality control processes result in a phage product they will allow for generic therapeutic use? If the proposed Entelli-02 phage cocktail cannot be used ad hoc, without prior patient-based competent authority and/or ethical committee approval, and/or informed consent, then its advantage over personalized approaches might be largely lost.

We acknowledge the ambiguity in our use of ‘therapeutic grade’ phage product along with the variation in regulatory approvals for phage therapy between international jurisdictions. Where possible, we have tried to generalise our terminology and regulatory differences but recognise this resulted in our descriptions being incomplete at times. We have now clearly defined ‘therapeutic grade’ in the main text (see lines **431-435**):

“Importantly, we define a therapeutic-grade phage product as having been produced under institutionally approved guidelines with end-point quality control measures for sterility, endotoxins, phage activity, and phage purity, and is suitable for intravenous administration to patients under Australia’s Therapeutic Goods Administrations (TGA) Special Access Scheme (Category A).”

To expand on this point, the Australian governing authority for therapeutic and medical goods is the Therapeutic Goods Administration (TGA). Currently, no bacteriophage products have been approved by the TGA for ad hoc use and regulatory approval must go through either the Special Access Scheme (SAS) or through the clinical trials framework. Our team has obtained regulatory and ethical committee approval from The Alfred Hospital to administer phage therapy on a case-by-case basis through the SAS. This included an extensive review of our phage production pipeline (which we document in this manuscript), our end-point quality control measures, and informed patient consent paperwork, which remains a requirement through the SAS.

Even with these regulatory requirements, we believe the use of a pre-formulated institutional cocktail such as *Entelli-02* offers advantages over personalised phage therapy. We expand on these points below.

5) Advantages. The authors state “this ‘institutional cocktail’ offers a streamlined alternative to the extensive processes typical of personalized phage therapies, which leverages advantages from both single-patient compassionate use and broad-range phage cocktails”. However, most personalized phage therapy cases are not performed in a compassionate use setting

(compassionate use is a program offered by pharmaceutical companies to provide a medicine free of charge for indications that are not already included in a funded scheme. Many of them are performed instance under the umbrella of article 37 of the Declaration of Helsinki (unproven interventions in clinical practice), or in routine without prior competent authority or ethical committee approval, for any infection type, caused by antibiotic-resistant or antibiotic-susceptible bacteria alike (e.g., in Georgia).

We appreciate the reviewer's clarification on the distinction between compassionate use and personalised approaches. To address this, we have removed the term "compassionate use" from the manuscript.

Our phage cocktail is personalised at the institutional level, with the primary goal of improving access and enabling a rapid response to hospital-associated infections—particularly in cases where personalised phage preparation would fall outside the treatment window for critically ill patients. We expand on this point further below.

In personalized phage therapy, typically phages (adapted or not) are selected (based on a "phagogram") from a collection of pre-produced and quality-controlled phages that has been set up to target the bacteria that are prevalent in the concerned hospital. When there is no time to select the active phages only, it is possible to select and mix compatible phages that are most likely to be active into a cocktail (an ad hoc institutional cocktail as it were).

We agree with the reviewer's stance on personalised phage therapy approaches, which were the inspiration behind our design of the *Entelli-02* institutional cocktail. We have summarised the broad benefits and limitations of personalised phage therapy versus broad-spectrum phage cocktails in both our introduction (see **lines 56-64**) and discussion (see **lines 629-638**).

Assuming that a pre-prepared institutional cocktail consists of the phages that are supposed to be active in a certain setting and have been produced separately, and would need to be tested prior – or at least during application (to confirm its activity) – the differences between the 'institutional cocktail' and personalized approaches are minimal (the time of mixing phages). Differences in costs are also mainly related to the ad hoc mixing of phages, but on the other hand, individual phages are more stable than ready-to-use cocktails.

We agree that the differences in time and cost for preparing five individual phage preparations versus a single five-phage cocktail (as we have done with *Entelli-02*) are not extreme. However, there were time and cost savings associated with our production of *Entelli-02* as a combined cocktail rather than five individual phage preparations. These include our endotoxin depletion steps, which were performed on our combine cocktail rather than each phage individually, saving time, resources, and labour. In addition, all downstream packaging and storage steps were less complex and time-consuming once the five phages had been combined. This also reduced our expenditure for third-party sterility and endotoxin testings on a combined product. Further, having consulted with The Alfred Hospital's clinical and pharmacy teams, their preference was for the clinical administration of a singular phage product (i.e., a pre-prepared cocktail) rather than multiple products as this reduced their handling, administration, and lowered the risk of dosing errors. We acknowledge there are also benefits of producing and administering each of these five phages individually, however we made the decision to produce a combine cocktail based on the above points.

As observed in previous studies, such as the PhagoBurn study, it seems that also in this case, stocks of individual phages are more stable (> 18 months – Fig. 1J and 5H) than ready to use phage cocktails (6 months with one of the phages showing a ~ 1 log titer reduction – Fig. 6G).

Regarding phage stability, we have obtained new 18-month data for *Entelli-02* (Figure 6I) showing a plateau in phage stability. However, the stability data we report of individual phages (Figure 1J and 5H) and the stability of each phage within the final *Entelli-02* product (Figure 6I) are not directly comparable as the phages had been produced and stored in different ways.

For our individual phages, these were produced according to our previously published ‘Phage on Tap’ protocol (PMID: 30128988). We then performed phage stability tests using lysates that were purified through centrifugation, filtration and chloroform treatment, but remained in the original LB media used for lysate production, which we have clarified in methods (see **line 857-862**). Comparatively, our *Entelli-02* cocktail was produced through the Monash Phage Foundry (see methods **lines 930-968**) and stored in 1X PBS supplemented with 1 mM CaCl₂. As such, we cannot make the claim as to whether our phages were more or less stable in a cocktail formulation.

In addition, pre-prepared cocktails will not prevent the need for developing individual phages to cater for the predicted 8% of ECC infected patients that cannot be helped by the *Entelli-02* cocktail, and to allow for a regular update of the phage cocktail. Personalized approaches (only using a phage cocktail when necessary) will allow for a more sustainable use of phages, as in non-urgent cases (e.g., chronic infections) only active phages will be applied. In the absence of more information on possible bacterial phage resistance issues, I feel we should avoid as much as possible the systematic and empirical use of the same phage cocktails, thus minimizing the chances of making the same mistakes (the misuse of antibiotics).

We appreciate the reviewer’s points and largely agree. It is not our intent for our institutional phage cocktail to replace personalised approaches, but rather we view *Entelli-02* as an extension of personalised phage therapy approaches with high purported efficacy to ECC infections from The Alfred Hospital (i.e., an institutional cocktail).

We further agree that where possible, only active phages should be administered to patients and support the sustainable use of phages in therapeutic settings. The primary goal and use-case for *Entelli-02* was to improve access and enable a rapid response to hospital-associated infections—particularly septicemia cases of Gram-negative, multidrug-resistant infections where the clinical treatment window is short (often <48 hours). In such cases, a pre-produced and validated institutional cocktail with high antimicrobial efficacy against a target pathogen group would enable the treatment of these high-risk patient cohorts.

To address these comments, we have revised our introduction to emphasize the benefits of personalised phage therapy and how we define *Entelli-02* as an institutional cocktail (**see lines 92-99**) Further, we have expanded our discussion to acknowledge several of these points as limitations and to avoid the ineffective use of phages (**see lines 732-739**)

6) Adaptation. The authors use the terms “training” and “evolution” when referring to the (pre-)adaptation of phages to their hosts. However, the term “training”, which is frequently used, refers to phenotypic changes only (e.g., one can train to become better at baseball), while the authors clearly show that their adaptation process results in the selection of genetic changes. I suggest to either use the term “adaptation” or to put the term training between brackets. We have changed our use of ‘training’ in place ‘adaptation’ and added “(also known as phage training)” between brackets as requested (**see line 65**). We now use the term adaptation throughout the manuscript.

It is assumed that adapting phages to a bacterial isolate can lead to increased phage lytic

activity, but also to delayed bacterial phage resistance evolution. In the present study, the authors seem to have focused on increasing the lytic activity of the phages and not on delaying the emergence of phage-resistant bacteria. Do the authors have information on the effect of their adaptation process on bacterial phage resistance?

The reviewer is correct in that we did not specifically adapt our phages to delay the emergence of bacterial resistance. Rather, our goal was to maximise phage infectivity, which we assessed through growth kinetics, Efficiency of Plating (EOPs), and lytic replication parameters (i.e., one-step growth curve), as presented in Figure 4.

As previously discussed, we have completed additional experiments investigating the effects of phage adaptation on the emergence of phage resistance and their associated fitness trade-offs (see Reviewer 2's points 1 & 2 and our detailed responses above).

The authors decided to standardize the phage adaptation process, determining an 'optimum phage training duration'. However, the duration of a phage adaptation procedure depends on how much the phage needs to adapt to reach a desired activity level (the required genetic drift). The optimum duration of 7 days is valid for the considered phage/bacterial isolate combination, but cannot be generalized.

We agree and acknowledge that the optimal duration of phage adaptation cannot be generalised and have added a statement in the results addressing this (see **lines 328-330**) and discuss further in discussion (see **lines 679-685**).

The start condition (prior to adaptation) is an EOP of 0.79, which is considered to be a moderate activity by the authors. However, an EOPs of 0.79 is usually considered to be sufficient for therapy and can even be observed in biological replicates (same phage and bacterial isolates) of phage/bacterium couples that can obtain an EOP of 1. In other words, in the presented situation the required adaptation process would be limited (e.g., from EOP 0.79 to EOP 1.0), and no adaptation would even be acceptable for therapeutic use.

Regarding this strain selection, we agree with the reviewer and have removed the term "moderate EOPs" to better reflect the observed value of 0.79.

Lines 291-292. It is more than intriguing that a phage, which initially was not capable of productive infection in the training host, adapts to propagate in that same host without additional incitation (e.g., without addition of other helper phages that do propagate in the training host). What process would be at the basis of this phenomenon? How sure are the authors that there was no phage infection at all prior to the adaptation process?

We were similarly surprised to see this phage-host pair adaptation and initially designed this experiment as a negative control. We have since re-examined infection profile using growth kinetics at MOIs ranging from 100 to 0.001 and spot assays across multiple dilutions. We again found no evidence of infection for this wild-type phage-host pair. This adaptation may be the result of an underlying phage defence mechanism and we selected for an escape mutant, but this is just a hypothesis and would require further investigation to disentangle. We have revised the explanation in the manuscript accordingly (see **lines 298-303**).

7) Phage lytic activity. The terms "productive infection", "activity" and "hazy spots/lysis" need to be better defined and seem to be mixed up throughout the text and in figures. For instance, lines 260-262: "hazy spots were not confirmed for productive infection, but likely represent a combination of low EOP), phage resistance and bactericidal activity resulting from "killing from without". A low EOP implies a productive infection, hazy spots due to phage

resistance also imply a productive infection, followed by the emergence of colonies of phage resistant bacterial mutants, while “killing from without” does not imply a productive infection. We have made changes through our manuscript to address this comment. For our spot assays, we now refer to these as either complete, partial, or no lysis throughout the manuscript.

It would have been more informative and clearer if all phages/bacterial isolate combinations would have been compared based on the same tests instead of a mix of spot test, double agar overlay (DAO) method (optional), and growth kinetics (optional). For instance, conclusions with regard to phage activity variations are sometimes solely based on DOA, sometimes on growth kinetics.

We acknowledge there is a large variety in methods used to validate our phage infectivity and host range. Much of this is due to workload constraints and screening infectivity and host range of multiple phages across an expanding ECC host library. Please see our response to Reviewer 1 who raised similar concerns. To address this, we have included a final host range map of our *Entelli-02* cocktail using EOP measures (Figure 7 A&B).

8) Phage production. The lysate was diluted up to tenfold in 1X PBS. Did the PBS contain Magnesium and Calcium? Is the final product compatible with intravenous application? Were animal component free ingredients (e.g., culture media) used? Were used solvents (e.g., 1-Octanol) removed from and/or quantified in the final product? Our final *Entelli-02* product was reconstituted in 1X Phosphate-buffered saline (PBS) supplemented with 1 mM calcium chloride to enhance phage stability, we have added this information to the manuscript (see **lines 449, 933 and 958**). The original culture media (Lysogeny Broth; composition in methods) contained animal-derived components. Residual solvents (i.e., 1-Octanol) were removed from the final product through two rounds of ultracentrifugation at 4000 ×g, although we did not quantify residual solvent in the final product.

MINOR COMMENTS

Line 664: “..., we purified the resistant bacterial colonies...”. Since phage-resistance need to be confirmed later, I suggest “..., we purified the supposedly resistant bacterial colonies...”. Corrected (see **line 814**).

The terms isolate and strain need to be better defined and are regularly mixed up throughout the manuscript.

We have corrected this and used term “isolate(s)” throughout the manuscript.

Reviewer #3

The manuscript “Rational Design of Frontline Institutional Phage Cocktail for the Treatment of Nosocomial Enterobacter cloacae Complex Infections” from D. Subedi, J. J. Barr and co-workers describes an approach to generate a phage-cocktail (Entelli-02) to be used against infections by MDR (MultiDrug-Resistant) Enterobacter cloacae complex (ECC). Based on an iterative method of phage training to increase host range, their strategy succeeds to develop a phage-cocktail of expanded host range within E. cloacae species.

The work developed by the authors is timely, as indeed current phage-treatment options based on highly-personalized treatments or rather general cocktails covers two extremes of all possibilities. A midway solution, which is stable for pharmaceutical handling and usage, while retaining good antibacterial activity against a given species rather than a strain is certainly of the highest translational value. The manuscript is clearly written and in general the figures are of good quality.

We thank the reviewer for the comments and peer review of our manuscript.

However, I have a few major points that I believe should be clarified:

1. While I appreciate the strong translational aspects highlighted in the manuscript, I think the emphasis on biological insight and novelty are either absent or not addressed. The experiments and data generated (experimental evolution & sequencing (full genome?)) offer plenty of opportunity to investigate potential novel resistance mechanisms from the bacterial side, and improved infection traits on the phage side.

We acknowledge the strong translational aspects of our manuscript, and this was one of our goals from the project’s inception. However, we spent considerable time and efforts to investigate and characterise our phages and present biological insights into their function. This includes, but is not limited to, characterising phage infectivity and lifecycles through one-step growth curves, delineation of phage receptors, the adaptation of our phages along with a genomic and phenotypic analysis of changes. We have further completed additional experimental work investigating the emergence of phage resistance and their associated fitness trade-offs in this revision. While we cannot explore all avenues, we hope this revision address this reviewers’ concerns.

However, the only aspects and mutations highlighted (or even commented) in the manuscript are bound to phage receptors and phage tail proteins (=adsorption). All other mutations (if any) in the genomes potentially also influencing the phenotypic traits of interest are not highlighted, or even more sticking, not mentioned, listed or acknowledged.

We focused our analysis of phage-resistant mutants on mutations within genes affecting surface-associated structures, as these mutations are well accepted to be the primary mechanism for the emergence of phage resistance. We have made some slight in-text changes to emphasize this (see **lines 169-170**). However, all the identified SNPs in the genomes of the resistant mutants were included in the Supplementary Data 2 file.

For instance, no information is provided on how many mutations (if any) other than phage receptors each resistance each mutant had, which can potentially contribute to resistance. Perhaps the most prominent evidence that this could be worth looking is the adsorption experiment using complementation: in no example the complementation completely phenocopies the wild-type. This is a strong indicator that there might be some other mutations supporting the resistance phenotype, which in my view should be acknowledged (even if

cannot be totally understood). Without further investigation of these, or from the trained phages, the manuscript brings little biological novelty to the field.

While we cannot rule out the possibility that other mutations contributed to partial restoration or alteration of adsorption, we also note that the receptor genes were complemented using over-expression plasmids. This may have influenced gene expression levels, an aspect that was not assessed in this study.

To address these comments, we have revised the text to further the partial restoration of adsorption and mutations in non-surface associated genes (see **lines 199-204**).

2. The approach deployed to assemble Entelli-02 is very interesting (I don't comment on novelty, non-expert), specially because it probes strain diversity to design a "large" host range viral cocktail within a species. However, the final results show that a cocktail containing the parental phages performs as good as, if not better than, the one containing the trained ones. In addition, there is no sufficient rationale provided for the selected isolate-phage pairs for training or resistance evolution. Same applies for the phages selected for the 8 combinations tested. How do the authors know that the selected isolate-phage pairs are indeed the ones yielding the best phages?

This is a valid point and a major limitation with phage adaptation approaches. Our selection of phage-host pairs to adapt was largely arbitrary and based on selecting a 'low, medium, and high' EOP hosts for our three select phages. Had we selected different host, then likely we would have observed differing results based on selective pressures.

Regarding our selection of the final cocktail across our 8 combinations, this was based firstly on host range, and we show that both combinations G and I could productively infect comparable number of hosts. Second, we consider the lytic replication of each phage and demonstrated that all three of our evolved phages exhibit improved replicative features as assessed through the one-step growth curve (Figure 4, F,G&H).

The strains used for training will probably have a strong impact on selected phage-traits, so one possibility would be to actually investigate this more in detail in a systematic way (aka more strains). On one hand, it would increase the chance that better phages are indeed obtained (just by trying more). On the other hand, even if no better phages are found, such a systematic study would be very interesting in itself from the evolutionary perspective on which phage traits are selected for across several isolates, or which ones are more isolate specific. It would give a good angle for novel biological insight, I believe.

We agree with the reviewer's point, however the proposed work is extensive and outside of the scope of our current manuscript. We have added additional points in our discussion addressing these limitations and the need for future work (see **lines 682-685**).

3. The concept that using more phages with "different" receptors improves efficacy can be very misleading. The fact is that LPS is the receptor for several phages, and there is no guarantee that "any" LPS mutation would not confer resistance to all these phages (cross-resistance). The authors could either access if this is the case, or ease their claim that this is in fact the case. As it is, this is not tested in the manuscript.

Please see our responses to Reviewers 1 and 2 above, along with our inclusion of additional experimental data (Figure 7 C, D&E) addressing this point.

4. Last, while the authors went through large efforts to show clinical importance of their initial cocktail (even mice models), the efficiency of Entelli-02 as clean and purified for

pharmacological application is totally absent (or did I miss it? At least not shown in Fig 6). Given that the preparation changes (or might) the titter of each phage, I think it would be absolutely crucial to at least quantify its activity against the strain panel. Without this, the translational prospect of the manuscript is inconclusive.

The reviewer is correct in that we did not complete an *in vivo* experiment testing the efficacy of our *Entelli-02* product. Please see the response to Reviewers 1 and 2 above, along with the new experimental data (Figure 7 F&G) addressing this point.

Minor comments

- The usage of “considerable morbidity and mortality” in the abstract is somewhat vague. What does “considerable” mean? It raises the question on whether this pathogen is really that important. I suggest to either be more precise, or just avoid such vague statement that can open questions on relevance.

We have removed this statement from the abstract.

- Fig 1, panel I: no indication of how many samples (and variance) were used for EOP estimation.

We have included this in figure legend (Fig 1I) stating $n = 1$.

- Fig 3 panel C: 6 datapoints do not necessarily need a box-plot. I would suggest to just represent directly the six data points.

We have added data points on the graph.

- Very specific sentence for non-experts “Hazy spots (~45%) were not confirmed for productive infection via the soft overlay (top) agar assay, but likely represented a combination of low EOP, phage resistance and bactericidal activity resulting from killing from without”. What does it mean killing from without?

We have reworked this section to improve its readability. We have also removed the term ‘hazy spots’ in place of partial lysis.

The term ‘killing from without’ is a virology term that refers to cellular killing in the absence of viral replication, typically due to excessive viral adsorption events causing host membrane depolarisation. Or simply put, cellular killing without viral replication.

- Line 276: Can this be seen in Fig 1I? it would be great to have an indication (Fig. 1I).

Added indication of Fig 1I (see **Line 280**).

- Line 280: this is a vague definition of “phage scores” for the main text. It would be good to have a more clear definition already here to make it easier to follow. At least to state what the phage score reflects. Supp Fig 6 helps only partially. Methods is good, but too late for the non-expert reader.

We have added a simplified description of the phage score to the main text (see **lines 285-286-427**):

“Phage scores takes growth kinetic data and integrates this into a single value from 0 to 1 (see methods), with higher values representing greater phage fitness and infectivity.”

- Line 291: EOP of strain CPO390 not shown in Fig 1E.

We assume the reviewer is referring to Figure 1I. Yes, this is correct—CPO390 was not infected by the initial three phages, therefore it is not in the EOPs graph.

- Fig 4F: Latency lines could also be coloured according to their phage lineage (evolved or ancestor) to facilitate visualization

We have added a coloured lines for visualisation of the latency period in Figure 4F as requested.

- Line 347: “To address this, we selected ECC strains from STs that lacked sufficient phage coverage as hosts for targeted phage isolation.” Very vague – which STs? Are they part of the initial smaller set or of the entire collection?

The selected isolates for targeted phage isolation were listed in supplementary data 1. To address this point, we have simplified the sentence to improve readability (see **lines 356-357**).

- Line 397: How were the 8 combinations containing 3, 4 or 5 phages from the 8 possible phages selected? Theatrically, there are $56 + 70 + 56$ possibilities to combine 8 phages in groups of 3, 4 or 5...

We have amended this sentence to avoid confusion and improve readability (see **lines 403-405**).

Reviewer Comments Nature Microbiology Subedi et al., 2024

Please find below a line-by-line response to all reviewer comments with our responses shown in blue.

Reviewer 2

I would like to thank the authors for the clarifications and additional experiments, and for the adaptations to the manuscript. The authors clarified that phage selection was principally based on host coverage, resulting in a limited diversity (within the same bacterial structure - LPS) in phage binding sites.

Designing phage cocktails based mainly on the host range of the phages is an age-old practice. Today, however, considering the current access to phage genome sequencing, and the increasing knowledge regarding the interaction of phages with their bacterial hosts, I expected a more innovative approach to phage cocktail design.

I wonder, especially when considering the large amount of work that went into designing this institutional phage cocktail, why the authors did not decide to opt for a more rational approach, selecting phages that target different bacterial structures, including receptors for which phage binding is predicted (e.g., in in vitro and/or animal experiments) to exert selection for bacteria to evolve phage resistance, while impairing the effectiveness of these receptors, causing reduced pathogenesis (1) or increased sensitivity to certain antibiotic classes (2).

(1) Kortright KE et al. Selection for Phage Resistance Reduces Virulence of *Shigella flexneri*. *Appl Environ Microbiol.* 2022 Jan 25;88(2):e0151421.

(2) Chan BK et al. Phage selection restores antibiotic sensitivity in MDR *Pseudomonas aeruginosa*. *Sci Rep.* 2016 May 26;6:26717.

As noted in our earlier revisions, our cocktail design was primarily guided by host range coverage during phage isolation and selection. However, we agree with reviewer's point and have provided suggested information and relevant references as;

“These findings highlight the importance of designing phage cocktails with multiple active agents per host and where possible, including phages with diverse receptors to maximise therapeutic robustness and minimise cross-resistance. The use of receptor-diverse phages may also offer additional benefits, such as reduced bacterial pathogenesis⁵⁹ or increased sensitivity to certain antibiotic classes⁶⁰.”

Regarding the “murine infection model”, the authors state that the demonstration of the presence of bacteria in tissues one hour post inoculation (not infection) is proof of infection. However, I was taught that the presence of bacteria in body tissues demonstrates colonization, while infection required the multiplication of the bacteria in these tissues, causing symptoms, disease, and immune response. I do, however, agree with the authors that initiating anti-bacterial treatment merely one hour after bacterial inoculation is a common practice in animal models and can be found in many publications.

Finally, authors added a summary of the broad benefits and limitations of personalized phage therapy versus broad spectrum phage cocktails in both the introduction and discussion of the manuscript. They state “In contrast, personalised phage therapy involves the identification and use of a phage with demonstrated activity against a patient's bacterial infection. This approach is often followed by adapting the phages (also known as “phage training”) to improve their effectiveness, followed by small-scale production and bespoke treatment”.

This description refers to the most extreme form of personalized phage therapy. The selection of several phages that are active against the infecting bacteria (a personalized phage cocktail) from a collection of pre-produced and controlled (released) phages is a more common form of personalized phage therapy, which of course, if necessary, can be combined with the procedure described by the authors. The company Adaptive Phage Therapeutics (acquired by BiomX), for instance, developed and produced (under GMP) an extensive phage bank from which phages can be selected to precisely match the patients' infections.

We agree with the reviewer's comments and acknowledge that selecting multiple active phages against the infecting bacteria from a collection of pre-produced, quality-controlled phages (i.e., a personalised cocktail) is a valid—and in some cases, potentially preferable—approach to cocktail preparation. However, we wish to re-emphasise that producing *Entelli-02* as a pre-formulated cocktail offered several practical advantages, including streamlined endotoxin removal, simplified packaging and storage, reduced third-party testing costs, and clinical preference for a single-dose product, all of which informed our decision to pursue this approach.

We have addressed reviewer's points in the discussion as;

“Alternatively, a personalised phage therapy approach, which utilises individual phages that are pre-produced in quality-controlled batches, would enable modular treatment strategies with similar benefits to *Entelli-02* in their speed and precision for therapeutic applications.”

Reviewer 3

I thank the authors for their efforts to answer and clarify all the questions. I also totally understand the challenge that would have been to experimentally address all requests, especially when it comes to simultaneously strengthen mechanistic insights and translational aspects. Therefore, I think the authors invested significant efforts to solidify their findings, namely with assessing (cross-)resistance, effect of antibiotic combinations and in vivo experiments with *Entelli-02* – organized in a new section in the manuscript “Host range, phage resistance, and antibiotic synergy of *Entelli-02*”. In my opinion, this was a good and balanced choice of effort allocation because it improved the quality of the manuscript. I still have a few minor points/questions listed below.

Line 517: Reference to Fig. 7C at the end of the sentence is missing?

We have added the reference to the Fig.

Line 518-20: Here it is unclear whether the resistant hosts were obtained from the experiment from 7D or whether resistance evolution was again conducted individually for each phage-host pair. I guess the second?

Resistance evolution was re-conducted individually for each phage-host pair. We have rewritten the associated text to state this point more clearly.

“To further investigate phage resistance impacts on bacterial infectivity and antibiotic interactions, we repeated our phage-resistance evolution experiments to isolate three new and

independent phage-resistant mutants, for each component phage from *Entelli-02*, using their respective hosts of isolation (total of 15 mutants).”

Figure 7D: I am unsure on how to clearly interpret this panel. As far as I understood, the color reflects phage-scores of infecting parent and resistant mutants with all phages, so this show allow to see resistance towards the phage the mutant was evolved with, but also cross-resistance towards all phages. I think it would be nice if the authors could somehow highlight the evolved host-phage pairs in the figure, since these are almost positive controls – lower phage scores in the resistant mutants are expected (even though APO57 did not seem to evolve that much resistance). As for the rest (cross-resistance) one does not really have an expectation, so all good. I think this would just guide the reader to digest this panel easier.

We have highlighted the instances of cross resistance and clarified this within the figure legend.

Line 524: It seems to me that when “one” (rather than “multiple”) phage exhibits strong lytic activity... which is actually good (better?). It somewhat “conflicts” with the next sentence, which is also true, but both are valid.

Our data highlighted that while a single, active phage can confer strong initial efficacy, cocktails where multiple phages exhibit lytic activity against a given host are more robust in the face of emerging cross resistance. We have clarified this as;

“However, in cases where only one phage dominates the lytic activity (e.g., øEnA02, øEnC15, or øTaquito), cross-resistance can reduce *Entelli-02* effectiveness. This suggests that relying on one dominant phage increases the risk that resistance to it may compromise the entire cocktail.”

Figure 7E: What does the last row, separated from the antibiotics, mean?

We have added a legend for this in the Figure.

Line 565: “our *Entelli-02* product broadly limited the emergence of phage resistance”: Is this actually shown? The statement is quite general (ambiguous). One could readily that the cocktail slows down resistance development or really limits the emergence of some mutations in direct comparison to using the phages alone, but this was not really tested. I would suggest to rephrase to keep more loyal to the findings – which is that the observed collateral sensitivity patterns among evolved host-phage have the potential to slow down resistance to the cocktail.

We agree and have changed the word “limited” to “slowed”

Fig 7G (or strain AALF22D176) is not mentioned in the text at all. Is this intentional? Actually here *Entelli-2* seems to work better than the initial cocktail, so it can be better after all.

We have now mentioned the isolate name in the text.